# A SECOND-ORDER PERSPECTIVE ON MODEL COMPOSITIONALITY AND INCREMENTAL LEARNING

**Angelo Porrello**[1], **Lorenzo Bonicelli**[1], **Pietro Buzzega**[1], **Monica Millunzi**[1,2],
**Simone Calderara**[1], **Rita Cucchiara**[1,3]

[1]University of Modena and Reggio Emilia, Italy    [2]AxyonAI, Italy    [3]IIT-CNR, Italy

## ABSTRACT

The fine-tuning of deep pre-trained models has revealed compositional properties, with multiple specialized modules that can be arbitrarily composed into a single, multi-task model. However, identifying the conditions that promote compositionality remains an open issue, with recent efforts concentrating mainly on linearized networks. We conduct a theoretical study that attempts to demystify compositionality in standard non-linear networks through the second-order Taylor approximation of the loss function. The proposed formulation highlights the importance of staying within the pre-training basin to achieve composable modules. Moreover, it provides the basis for two dual incremental training algorithms: the one from the perspective of multiple models trained individually, while the other aims to optimize the composed model as a whole. We probe their application in incremental classification tasks and highlight some valuable skills. In fact, the pool of incrementally learned modules not only supports the creation of an effective multi-task model but also enables unlearning and specialization in certain tasks. Code available at `https://github.com/aimagelab/mammoth`.

## 1 INTRODUCTION

In the last two decades, AI technologies have predominantly been viewed as monoliths, centered around a single model trained on a single (albeit large) dataset. This paradigm has undeniably yielded impressive results and is likely to continue doing so. However, to meet the numerous challenges of business, more modular and flexible approaches, such as *model averaging* (McMahan et al., 2017) and *Mixture Of Experts* (Yadav et al., 2024a), have experienced a renewed interest. In fact, the life cycle of monolithic models demands incremental and intensive updates to accommodate the diversity of markets, platforms, and customer needs. Moreover, many companies lack the financial means to assemble a proficient AI team or access GPU clusters to support the rapid evolution of these factors. In an effort to democratize AI, we advocate for modern paradigms (Li et al., 2023b; Pfeiffer et al., 2023) prioritizing fast adaptability, model customization, and data heterogeneity.

Recent frameworks (Liu & Soatto, 2023; Bowman et al., 2023) support these principles through *model compositionality*, allowing the creation of bespoke models through cheap editing operations. Surprisingly, fine-tuning pre-trained models not only enables efficient transfer with limited data (Zhuang et al., 2020) but also offers insights into compositionality (Ilharco et al., 2023). In fact, it has been shown (Wortsman et al., 2022) that a simple linear combination of individually fine-tuned weights yields meaningful outcomes. The resulting composed model exhibits rich and robust representations (Zhang & Bottou, 2023) without incurring additional inference or memory costs. Such a simple but powerful schema has been primarily employed in *model soups* (Wortsman et al., 2022), where individual models are trained on the same task but with different hyper-parameters. In contrast, *Task arithmetic* (Ilharco et al., 2023; Liu & Soatto, 2023; Ortiz-Jimenez et al., 2024) addresses a cross-dataset setting with each individual model focusing on a distinct task. Such a framework has also been recently explored for language modeling through *ColD fusion* (Don-Yehiya et al., 2023).

In this context, we elaborate on two points. Firstly, considering multiple models trained individually on different tasks, we aim to understand the conditions that allow the successful combination of their weights. This finding has been predominantly explored empirically (Ilharco et al., 2023), with a few notable exceptions (Ortiz-Jimenez et al., 2024) that aim to provide a more theoretical understanding.

However, these approaches only focus on linearized networks and tangent fine-tuning (Liu & Soatto, 2023). We instead aim for a formulation that generalizes to standard non-linear deep networks and diverse fine-tuning strategies (*e.g.*, LoRA (Hu et al., 2022)). To do so, we propose leveraging the second-order Taylor approximation of the loss function around the pre-training weights. This enables us to derive a relationship between the capabilities of individual models and those of their composition, offering a key takeaway. Specifically, for the composed model to be accurate, each component should attain decent performance on examples outside its individual training distribution.

The second point of our study is a derivation of the former and revolves around learning composable models in an incremental fashion, dedicating an individual module to each incoming task. In this respect, previous works (Bowman et al., 2023; Liu & Soatto, 2023) build on the intuitive idea that modularity provides a natural foundation for incremental training. Through model editing and composition, in fact, one can simply introduce novel concepts without having to re-train the model. In this work, we turn the perspective and argue that compositional capabilities require incremental learning skills. Indeed, as mentioned above, preserving out-of-distribution performance is essential for compositionality. Nevertheless, this can be framed as a continual learning problem (Kirkpatrick et al., 2017), where the objective of each individual model is to maintain the pre-training general capabilities on examples that fall outside its specific training distribution.

On this basis, we propose two algorithms that tackle the incremental fine-tuning of composable models. Both use the same second-order formulation but approach the problem from opposed perspectives: optimizing the loss of each model individually *vs.* optimizing the loss of the composed model. Evaluated in the *class-incremental* setting (Van de Ven et al., 2022), our approaches lead to an accurate multi-task model with further editing capabilities, which allow the efficient *specialization* of the model on certain tasks, as well as the ability of removing some others (*unlearning*).

## 2 FRAMEWORK

We consider $f(\cdot; \boldsymbol{\theta}) : \mathcal{X} \to \mathcal{Y}$ as a twice-differentiable deep network with weights $\boldsymbol{\theta} \in \Theta \subseteq \mathbb{R}^m$. It takes inputs $\boldsymbol{x} \in \mathcal{X} \subseteq \mathbb{R}^d$ and yields a conditional distribution $p_\theta(\boldsymbol{y}|\boldsymbol{x})$ over the targets $\boldsymbol{y} \in \mathcal{Y} \subseteq \mathbb{R}^c$. In this paper, we focus on **incremental training**, which progresses sequentially through a series of $T$ classification tasks $\mathcal{T} = \{1, 2, \dots, T\}$. Each task $t \sim \mathcal{T}$ is characterized by a dataset $\mathcal{D}_t$ with $n_t$ training samples $\boldsymbol{x}, \boldsymbol{y} \sim p_t(\boldsymbol{x}, \boldsymbol{y})$ drawn from a distribution varying across tasks. We assume that tasks share the same loss function $\ell(\boldsymbol{\theta}|\boldsymbol{x}, \boldsymbol{y})$, *i.e.*, the negative log-likelihood $-\log p_\theta(\boldsymbol{y}|\boldsymbol{x})$.

In this setting, we maintain a pool of composable networks $\mathcal{P} = \{f(\cdot; \boldsymbol{\theta}_t) \mid \boldsymbol{\theta}_t \triangleq \boldsymbol{\theta}_0 + \boldsymbol{\tau}_t\}_{t \in \mathcal{T}}$, where each model is fine-tuned from a common set of pre-training weights $\boldsymbol{\theta}_0$. The *task vector* (Ilharco et al., 2023) $\boldsymbol{\tau}_t$ indicates the displacement in weight space w.r.t. $\boldsymbol{\theta}_0$ after training on task $t$. We obtain the weights of the composed model $f_\mathcal{P}$ by averaging the weights within the pool:

$$f_\mathcal{P} \triangleq f(\cdot; \boldsymbol{\theta}_\mathcal{P}) \quad \text{s.t.} \quad \boldsymbol{\theta}_\mathcal{P} = \boldsymbol{\theta}_0 + \sum_{t=1}^{T} w_t \boldsymbol{\tau}_t, \quad \sum_{t=1}^{T} w_t = 1 \tag{1}$$

where $w_t$ balances the contribution of the $t$-th learner. While some works (Asadi et al., 2024; Huang et al., 2024) optimize these coefficients, we devise uniform weights $w_t = 1/T$ in our algorithms.

**Scope.** How can we learn multiple disjoint tasks through a pool $\mathcal{P}$ of models, so that the composed model performs well on their union? To answer this question, we introduce the concept of **empirical risk**, *i.e.*, the average loss $\hat{\ell}(\boldsymbol{\theta}|\mathcal{D})$, computed over the union $\mathcal{D} = \bigcup_{t=1}^{T} \mathcal{D}_t$ of all training tasks:

$$\hat{\ell}(\boldsymbol{\theta}|\mathcal{D}) = \frac{1}{\sum_{t=1}^{T} n_t} \sum_{\boldsymbol{x}, \boldsymbol{y} \in \mathcal{D}} \ell(\boldsymbol{\theta}|\boldsymbol{x}, \boldsymbol{y}) \approx \mathbb{E}_{\substack{t \sim \mathcal{T} \\ \boldsymbol{x}, \boldsymbol{y} \sim p_t(\boldsymbol{x}, \boldsymbol{y})}} [\ell(\boldsymbol{\theta}|\boldsymbol{x}, \boldsymbol{y})] \tag{2}$$

To simplify notation, we will henceforth omit the explicit dependence of the loss on the data, denoting the individual loss $\ell(\boldsymbol{\theta}|\boldsymbol{x}, \boldsymbol{y})$ simply as $\ell(\boldsymbol{\theta})$, and the empirical risk $\hat{\ell}(\boldsymbol{\theta}|\mathcal{D})$ as $\hat{\ell}(\boldsymbol{\theta})$. On this basis, the rest of this section will delve into the following research questions.

> *Question i)* Given the empirical risk of each individual model, what can we say about the composed model $f(\cdot; \boldsymbol{\theta}_\mathcal{P})$? *Question ii)* How can we train individual learners $f(\cdot; \boldsymbol{\theta}_t)$ on distinct tasks (***individual training***) to still achieve a reliable composition $f(\cdot; \boldsymbol{\theta}_\mathcal{P})$? *Question iii)* Instead of optimizing each model on its individual loss, could we optimize each model based on the loss of the whole composed model $f(\cdot; \boldsymbol{\theta}_\mathcal{P})$ (***ensemble training***)?

## 2.1 INDIVIDUAL LEARNERS *vs.* THE COMPOSED MODEL: A PRE-TRAINING PERSPECTIVE

We now relate the composed model $f_{\mathcal{P}}$ to the individual components $f(\cdot; \boldsymbol{\theta}_t)$ of the pool. To do so, we introduce the **second-order** Taylor approximation $\ell_{\mathrm{cur}}(\boldsymbol{\theta})$ of the loss around the pre-trained weights $\boldsymbol{\theta}_0$:

$$\ell(\boldsymbol{\theta}) = \ell_{\mathrm{cur}}(\boldsymbol{\theta}) + \mathcal{O}(\|\boldsymbol{\theta} - \boldsymbol{\theta}_0\|^3) \tag{3}$$

$$\text{where} \quad \ell_{\mathrm{cur}}(\boldsymbol{\theta}) = \ell(\boldsymbol{\theta}_0) + (\boldsymbol{\theta} - \boldsymbol{\theta}_0)^{\mathrm{T}} \nabla \ell(\boldsymbol{\theta}_0) + \frac{1}{2}(\boldsymbol{\theta} - \boldsymbol{\theta}_0)^{\mathrm{T}} \mathbf{H}_\ell(\boldsymbol{\theta}_0)(\boldsymbol{\theta} - \boldsymbol{\theta}_0). \tag{4}$$

$\nabla \ell(\boldsymbol{\theta}_0) \triangleq \nabla_{\boldsymbol{\theta}} \ell(\boldsymbol{\theta}_0 | \boldsymbol{x}, \boldsymbol{y})$ and $\mathbf{H}_\ell(\boldsymbol{\theta}_0) \triangleq \nabla_{\boldsymbol{\theta}}^2 \ell(\boldsymbol{\theta}_0 | \boldsymbol{x}, \boldsymbol{y})$ are the gradient and the Hessian around $\boldsymbol{\theta}_0$. Similarly, we can define the second-order approximation $\hat{\ell}_{\mathrm{cur}}(\boldsymbol{\theta}) \approx \hat{\ell}(\boldsymbol{\theta})$ of the empirical risk, which corresponds to averaging the approximated loss across examples from all tasks.

**Assumption.** We now assume that $\boldsymbol{\theta} = \boldsymbol{\theta}_0$ is a point of **local minimum** of the empirical risk $\hat{\ell}(\boldsymbol{\theta})$[1] across all tasks (Eq. 2). Under this hypothesis, the Hessian of the empirical risk $\mathbf{H}_{\hat{\ell}}(\boldsymbol{\theta}_0) \succeq 0$ is positive semidefinite. In light of this and the quadratic nature of $\hat{\ell}_{\mathrm{cur}}(\boldsymbol{\theta})$, we can state that the second-order approximation of the empirical risk $\hat{\ell}_{\mathrm{cur}}(\boldsymbol{\theta})$ is **convex**. Therefore, we apply the **Jensen's inequality** to derive a relation between the composed model and the individual components:

$$\hat{\ell}_{\mathrm{cur}}(\boldsymbol{\theta}_{\mathcal{P}} = \boldsymbol{\theta}_0 + \textstyle\sum_{t=1}^{T} w_t \boldsymbol{\tau}_t) \leq \textstyle\sum_{t=1}^{T} w_t \, \hat{\ell}_{\mathrm{cur}}(\boldsymbol{\theta}_t = \boldsymbol{\theta}_0 + \boldsymbol{\tau}_t). \tag{5}$$

The relation states that, under the second-order approximation, the empirical risk $\hat{\ell}_{\mathrm{cur}}(\boldsymbol{\theta}_{\mathcal{P}})$ of the composed model is upper-bounded by the convex combination of its individuals. In other words, if each individual model is trained to the optimum with near-zero loss value, there are some guarantees on the loss function attained by the composed model. Notably, this relation could help reduce the computational footprint during inference, as it enables the reduction of forward passes, from multiple (one for each individual) to a singular pass (performed on the composed model).

At a first glance, the result of Eq. 5 appears similar to the statement of Eq. 2 in (Liu & Soatto, 2023):

$$\hat{\ell}(\boldsymbol{\theta}_{\mathcal{P}} = \boldsymbol{\theta}_0 + \textstyle\sum_{t=1}^{T} w_t \boldsymbol{\tau}_t) \leq \textstyle\sum_{t=1}^{T} w_t \hat{\ell}(\boldsymbol{\theta}_t = \boldsymbol{\theta}_0 + \boldsymbol{\tau}_t) \tag{6}$$

$$\text{given that } f_t(\cdot; \boldsymbol{\theta}_t) \triangleq f_{\mathrm{lin}}(\cdot; \boldsymbol{\theta}_t) = f(\cdot; \boldsymbol{\theta}_0) + (\boldsymbol{\theta}_t - \boldsymbol{\theta}_0)^{\mathrm{T}} \nabla f(\cdot; \boldsymbol{\theta}_0) \quad (\textit{tangentness}) \tag{7}$$

However, some notable distinctions remain. Their inequality applies to the exact risk $\hat{\ell}$ but is valid only for linearized models (*i.e.*, fine-tuned in the tangent space of pre-training weights). In contrast, our result pertains to the *second-order* approximation $\hat{\ell}_{\mathrm{cur}}$ of the risk and applies to *any* fine-tuning strategy (*e.g.*, LoRA, adapters, etc.). Intuitively, our inequality provides more flexibility to the training of individual learners, as long as: *i)* the learners remain in the pre-training basin, such that $\mathcal{O}(\|\boldsymbol{\theta} - \boldsymbol{\theta}_0\|^3) \to 0$ and $\ell_{\mathrm{cur}}$ can be considered a good proxy of $\ell$; *ii)* $\boldsymbol{\theta}_0$ is a local minimum of $\ell$.

## 2.2 ENABLING INDIVIDUAL TRAINING IN INCREMENTAL SCENARIOS

As mentioned above, a possible application of Eq. 5 is to devote each learner to a distinct task and optimize them in isolation (*i.e.*, **individual training**). Indeed, the upper bound in Eq. 5 describes a sort of worst-case scenario for the risk of the composed model: at worst, it collapses to that given by the upper bound. Nonetheless, if every individual model were *accurate* on all tasks, the right-side term of Eq. 5 would yield an appealing upper-bound to the risk of the composed model.

**Limits of Eq. 5.** Under the constraints of individual training, each individual learner $f(\cdot; \boldsymbol{\theta}_t)$ can be optimized only on the current distribution $p_t(\boldsymbol{x}, \boldsymbol{y})$. Therefore, when considering examples from other data distributions $p_{t' \neq t}(\boldsymbol{x}, \boldsymbol{y})$, the loss $\ell_{\mathrm{cur}}(\boldsymbol{\theta}_t | \boldsymbol{x}, \boldsymbol{y})$ of the $t$-th individual learner is likely to be much higher for examples outside its training distribution $p_t(\boldsymbol{x}, \boldsymbol{y})$. As a consequence, the upper bound delivered by the right-side of Eq. 5 is likely to increase one task after the other, and we cannot rely on it to recover a reliable composed model $f(\cdot; \boldsymbol{\theta}_{\mathcal{P}})$.

In this respect, our proposal is to tighten the upper bound of Eq. 5 through explicit regularization, which we devise during the optimization of each individual learner. In practice, each model is

---

[1]As shown in Ostapenko et al. (2022), techniques like linear probing, latent replay, and incremental linear discriminant analysis can be employed to enforce the optimality of the base model $\boldsymbol{\theta}_0$, along with Instruction Tuning for language modeling (Yadav et al., 2024c). See Sec. 3 for additional implementation details.

provided with a learning objective that extends beyond minimizing the loss on the assigned data distribution $p_t(\boldsymbol{x}, \boldsymbol{y})$. Specifically, to ensure decent performance on external distributions $p_{t' \neq t}(\boldsymbol{x}, \boldsymbol{y})$, we anchor the model to the pre-training knowledge for out-of-distribution examples. We do so by encouraging the predictions of $f(\cdot; \boldsymbol{\theta}_t)$ to be close to those generated by the pre-trained model $f(\cdot; \boldsymbol{\theta}_0)$, given examples $\boldsymbol{x}, \boldsymbol{y} \sim p_{t' \neq t}(\boldsymbol{x}, \boldsymbol{y})$. Denoting the pre-training posterior distribution as $p_{\boldsymbol{\theta}_0}(\boldsymbol{y}|\boldsymbol{x})$, we have:

$$\underset{\boldsymbol{\theta}_t}{\text{minimize}} \quad \mathbb{E}_{\boldsymbol{x}, \boldsymbol{y} \sim p_t(\boldsymbol{x}, \boldsymbol{y})} \left[ \ell_{\text{cur}}(\boldsymbol{\theta}_t | \boldsymbol{x}, \boldsymbol{y}) \right] + \mathcal{D}_{\text{KL}}(p_{\boldsymbol{\theta}_0}(\boldsymbol{y}|\boldsymbol{x}) || p_{\boldsymbol{\theta}_t}(\boldsymbol{y}|\boldsymbol{x})). \quad (8)$$

It is noted that the computation of the KL term $\mathcal{D}_{\text{KL}}(\cdot)$ requires sampling from the external distributions $p_{t' \neq t}(\boldsymbol{x}, \boldsymbol{y})$. Theoretically, this clashes with the constraints of individual training. Thankfully, if $\boldsymbol{\tau}_t \to 0$, the KL term can be approximated (Chaudhry et al., 2018) with the *distance* between $\boldsymbol{\theta}_t$ and pre-training weights $\boldsymbol{\theta}_0$:

$$\mathcal{D}_{\text{KL}}(p_{\boldsymbol{\theta}_0}(\boldsymbol{y}|\boldsymbol{x}) \,||\, p_{\boldsymbol{\theta}_t}(\boldsymbol{y}|\boldsymbol{x})) \approx \tfrac{1}{2}(\boldsymbol{\theta}_t - \boldsymbol{\theta}_0)^{\text{T}} \text{F}_{\boldsymbol{\theta}_0}(\boldsymbol{\theta}_t - \boldsymbol{\theta}_0) = \tfrac{1}{2}\|\boldsymbol{\tau}_t\|_{\text{F}_{\boldsymbol{\theta}_0}}^2. \quad (9)$$

The distance is computed in the Riemannian manifold (Lee, 2006) induced by the Fisher Information Matrix (FIM) (Huszár, 2018); namely, a $|\theta| \times |\theta|$ positive semi-definite matrix given by:

$$\text{F}_{\boldsymbol{\theta}_0} = \mathbb{E}_{\substack{t \sim \mathcal{T} \\ \boldsymbol{x} \sim p_t(\boldsymbol{x}, \boldsymbol{y})}} \left[ \mathbb{E}_{\boldsymbol{y} \sim p_{\boldsymbol{\theta}_0}(\boldsymbol{y}|\boldsymbol{x})} \left[ \nabla \log p_{\boldsymbol{\theta}_0}(\boldsymbol{y}|\boldsymbol{x}) \nabla \log p_{\boldsymbol{\theta}_0}(\boldsymbol{y}|\boldsymbol{x})^\top \right] \right]. \quad (10)$$

Under the local minimum hypothesis on pre-training weights, the FIM at $\boldsymbol{\theta}_0$ equals the expected[2] Hessian of the negative log-likelihood (Martens, 2020): $\text{F}_{\boldsymbol{\theta}_0} = \mathbb{E}_{\boldsymbol{x}} \left[ \mathbf{H}_\ell(\boldsymbol{\theta}_0) \right]$. Due to this connection, the FIM yields insights on the sensitivity of each parameter to changes in the data distribution.

**Application to incremental learning.** Given that optimizing the regularization above does not necessitate data from external tasks, it can be readily adapted to incremental learning. Following Chaudhry et al. (2018); Schwarz et al. (2018), we introduce a few additional approximations. Firstly, we limit to estimate the diagonal $\hat{\text{F}}_{\boldsymbol{\theta}_0}$ of the FIM, thus avoiding the prohibitive footprint required to treat a $|\theta| \times |\theta|$ matrix. Basically, the diagonal $\hat{\text{F}}_{\boldsymbol{\theta}_0}$ consists of a Monte Carlo estimate of the (squared) gradients of the log-likelihood. Secondly, we note that the expectation $\mathbb{E}_{\boldsymbol{x}} \left[ \mathbf{H}_\ell(\boldsymbol{\theta}_0) \right]$ in the FIM cannot be directly estimated, as examples from all tasks are not available simultaneously but rather sequentially. We hence compute the expectation incrementally (Chaudhry et al., 2018; Schwarz et al., 2018) one task at a time. As each new task becomes available, we calculate a *local* Fisher matrix on the data of the new task and then accumulate it into a *global* running Fisher estimate. As outlined by Alg. 1, the accumulation can be thought as a simple summation, net of re-normalizing operations reflecting the number of samples of each task (see App. E).

Given Eq. 9 and the diagonal FIM, the augmented optimization problem for the $t$-th learner becomes:

$$\underset{\boldsymbol{\theta}_t}{\text{minimize}} \quad \mathbb{E}_{\boldsymbol{x}, \boldsymbol{y} \sim p_t(\boldsymbol{x}, \boldsymbol{y})} \left[ \ell_{\text{cur}}(\boldsymbol{\theta}_t | \boldsymbol{x}, \boldsymbol{y}) \right] + \frac{\alpha}{2} \text{EWC}_{\boldsymbol{\theta}_0}(\boldsymbol{\theta}_t) \quad (11)$$

$$\text{where} \quad \text{EWC}_{\boldsymbol{\theta}_0}(\boldsymbol{\theta}_t) = \sum_i^{|\theta|} \hat{\text{F}}_{\boldsymbol{\theta}_0}^{(i)} (\boldsymbol{\theta}_t^{(i)} - \boldsymbol{\theta}_0^{(i)})^2. \quad (12)$$

where $\alpha \geq 0$ is an hyper-parameter and $\text{EWC}_{\boldsymbol{\theta}_0}(\cdot)$ indicates the Riemannian distance from the pre-training weights $\boldsymbol{\theta}_0$. The acronym $\text{EWC}_{\boldsymbol{\theta}_0}$ highlights the strong analogy with Elastic Weight Consolidation (Kirkpatrick et al., 2017), a well-established approach against *catastrophic forgetting* (McCloskey & Cohen, 1989). In a sense, our term prevents forgetting pre-training knowledge; however, while our anchor is fixed at $\boldsymbol{\theta}_0$, the anchor of EWC instead shifts and focuses on the weights learned during the preceding task.

## 2.3 JOINT TRAINING OF THE COMPOSED MODEL IN INCREMENTAL SCENARIOS

Individual training profitably aligns with *decentralized learning* (Bowman et al., 2023), emphasizing minimal interactions between learners and privacy preservation. Nevertheless, it might be inefficient when these constraints are not of interest and the goal is simply to create an accurate model. In fact, individual training prevents a learner from leveraging the knowledge of other ensemble members, eliminating the potential for beneficial mutual transfer. For these reasons, we adopt the dual perspective, in which each model is directly optimized using the loss of the composed model $f(\cdot; \boldsymbol{\theta}_\mathcal{P})$ (**ensemble training**). Since the inequality in Eq. 5 does not provide much help for the explicit optimization of $\ell_{\text{cur}}(\boldsymbol{\theta}_\mathcal{P})$, we quantify the exact gap between the two sides of the Jensen's inequality:

---

[2]Precisely, we take the expectation w.r.t. data from other tasks $t' \neq t$ to reflect the regularization in Eq. 8.

**Theorem 1.** *Let us assume a pool $\mathcal{P}$ with $T \geq 2$ models, with the $t$-th model parameterized by $\boldsymbol{\theta}_t = \boldsymbol{\theta}_0 + \boldsymbol{\tau}_t$. If we compose them through coefficients $w_1, \ldots, w_T$ s.t. $w_t \in [0, 1]$ and $\sum_{t=1}^{T} w_t = 1$, the 2nd order approximation $\ell_{\mathrm{cur}}(\boldsymbol{\theta}_{\mathcal{P}})$ evaluated on composed weights $\boldsymbol{\theta}_{\mathcal{P}} = \boldsymbol{\theta}_0 + \sum_{t=1}^{T} w_t \boldsymbol{\tau}_t$ is:*

$$\ell_{\mathrm{cur}}(\boldsymbol{\theta}_{\mathcal{P}}) + \Omega(\boldsymbol{\theta}_1, \ldots, \boldsymbol{\theta}_T) = \sum_{t=1}^{T} w_t \ell_{\mathrm{cur}}(\boldsymbol{\theta}_t) \tag{13}$$

$$\textbf{where} \quad \Omega(\boldsymbol{\theta}_1, \ldots, \boldsymbol{\theta}_T) = \frac{1}{2} \sum_{t=1}^{T} \sum_{t' < t} w_t w_{t'} (\boldsymbol{\tau}_t - \boldsymbol{\tau}_{t'})^{\mathrm{T}} \mathbf{H}_\ell(\boldsymbol{\theta}_0)(\boldsymbol{\tau}_t - \boldsymbol{\tau}_{t'}). \tag{14}$$

*Proof in App. B.* The equality introduces a term $\Omega(\cdot)$ that is non-negative (due to $\mathbf{H}_\ell(\boldsymbol{\theta}_0) \succeq 0$) and depends on weights $\boldsymbol{\theta}_1, \ldots, \boldsymbol{\theta}_T$. Therefore, $\Omega(\cdot)$ is proportional to the cumulative *distance* between every pair of learners, within the Riemannian manifold induced by the Hessian $\mathbf{H}_\ell(\boldsymbol{\theta}_0)$ (Lee, 2006). Notably, Eq. 13 permits to draw the following insights: *i)* optimizing both the loss of the composed model $\ell_{\mathrm{cur}}(\boldsymbol{\theta}_{\mathcal{P}})$ and the term $\Omega(\cdot)$ is equivalent to individual training; *ii)* during *inference*, if $\Omega(\cdot)$ tends towards zero, performing a prediction with the composed model is akin to conducting multiple forward passes, each corresponding to an individual learner. Based on that interpretation, we now transition to a setting that considers the explicit auxiliary minimization of $\Omega(\cdot)$, as follows:

$$\underset{\boldsymbol{\theta}_1, \ldots, \boldsymbol{\theta}_T}{\mathrm{minimize}} \quad \underset{\substack{t \sim \mathcal{T} \\ \boldsymbol{x}, \boldsymbol{y} \sim p_t(\boldsymbol{x}, \boldsymbol{y})}}{\mathbb{E}} [\ell_{\mathrm{cur}}(\boldsymbol{\theta}_{\mathcal{P}} | \boldsymbol{x}, \boldsymbol{y}) + \beta \, \Omega(\boldsymbol{\theta}_1, \ldots, \boldsymbol{\theta}_T)] \tag{15}$$

where $\beta \geq 0$ is an hyper-parameter. It is noted that minimizing $\Omega(\cdot)$ encourages alignment among task vectors, especially for those weights that are sensitive/important in the pre-training loss landscape.

Similarly to Jeffares et al. (2024), the objective of Eq. 15 can be interpreted as a smooth transition between individual and ensemble training. Indeed, given that $\Omega(\cdot) = \sum_t w_t \ell_{\mathrm{cur}}(\boldsymbol{\theta}_t) - \ell_{\mathrm{cur}}(\boldsymbol{\theta}_{\mathcal{P}})$, Eq. 15 can be restated:

$$\ell_{\mathrm{cur}}(\boldsymbol{\theta}_{\mathcal{P}}) + \beta \, \Omega(\boldsymbol{\theta}_1, \ldots, \boldsymbol{\theta}_T) \overset{\mathrm{Eq.\ 13}}{=} (1 - \beta) \, \ell_{\mathrm{cur}}(\boldsymbol{\theta}_{\mathcal{P}}) + \beta \sum_{t=1}^{T} w_t \ell_{\mathrm{cur}}(\boldsymbol{\theta}_t) \tag{16}$$

This suggests that by minimizing both the loss $\ell_{\mathrm{cur}}(\boldsymbol{\theta}_{\mathcal{P}})$ of the joint composed model and $\Omega(\cdot)$, we also implicitly optimize the individual models. Notably, this result not only paves the way for a well-performing ensemble but also for components that are reliable when considered individually.

**Incremental ensembles.** We now refine the problem in Eq. 15 to account for incremental settings. We firstly bring the expectation inside $\Omega(\cdot)$ (Eq. 14) and replace the Hessian $\mathbf{H}_\ell(\boldsymbol{\theta}_0)$ with its expectation, taken across data points $\boldsymbol{x}, \boldsymbol{y} \sim p_t(\boldsymbol{x}, \boldsymbol{y})$ from all tasks up to the current one. Afterwards, we capitalize on a property mentioned in Sec. 2.2, which states that, for a point of maximum likelihood like $\boldsymbol{\theta} = \boldsymbol{\theta}_0$, the expected Hessian of the negative log-likelihood coincides with the Fisher $\mathrm{F}_{\boldsymbol{\theta}_0}$. As a result, we can approximate $\mathbb{E}_{\boldsymbol{x}}[\mathbf{H}_\ell(\boldsymbol{\theta}_0)]$ with the tractable diagonal Fisher $\hat{\mathrm{F}}_{\boldsymbol{\theta}_0}$. Based on $\mathbf{H}_\ell(\boldsymbol{\theta}_0) \approx \hat{\mathrm{F}}_{\boldsymbol{\theta}_0}$ and further steps (see App. B), we can rearrange $\Omega(\cdot)$ as:

$$\Omega(\cdot) \approx \Omega_{\hat{\mathrm{F}}}(\boldsymbol{\theta}_1, \ldots, \boldsymbol{\theta}_T) = \frac{1}{2} \sum_{t=1}^{T} w_t (1 - w_t) \underbrace{\mathrm{EWC}_{\boldsymbol{\theta}_0}(\boldsymbol{\theta}_t)}_{\text{see Eq. 12}} - \sum_{t=1}^{T} \sum_{t' < t} w_t w_{t'} \boldsymbol{\tau}_t^{\mathrm{T}} \hat{\mathrm{F}}_{\boldsymbol{\theta}_0} \boldsymbol{\tau}_{t'}. \tag{17}$$

Intuitively, $\Omega_{\hat{\mathrm{F}}}(\cdot)$ aims to preserve pre-training knowledge embodied by $\boldsymbol{\theta}_0$ through the first term; through the second one, instead, it encourages pairwise alignment between task vectors. Also, we highlight that the summations depend on the number of models $T$. Therefore, the *more* components we manage within the ensemble, the *more* crucial it becomes to minimize $\Omega_{\hat{\mathrm{F}}}(\cdot)$ within Eq. 15.

Finally, we plug the approximated barrier term $\Omega_{\hat{\mathrm{F}}}(\cdot)$ into the optimization problem outlined in Eq. 15. As learning proceeds in subsequent tasks, we can optimize the composed model only one task at a time. To prevent biasing the previously learned components of the pool toward current data, at each round $t$ we optimize only the weights $\boldsymbol{\theta}_t \triangleq \boldsymbol{\theta}_0 + \boldsymbol{\tau}_t$ of the corresponding $t$-th learner, while freezing the others components of the pool. This yields the following form for the $t$-th learning task:

$$\underset{\boldsymbol{\theta}_t}{\mathrm{minimize}} \quad \mathbb{E}_{\boldsymbol{x}, \boldsymbol{y} \sim p_t(\boldsymbol{x}, \boldsymbol{y})} [\ell_{\mathrm{cur}}(\boldsymbol{\theta}_{\mathcal{P}} | \boldsymbol{x}, \boldsymbol{y})] + \beta \, \Omega_{\hat{\mathrm{F}}}(\boldsymbol{\theta}_1, \ldots, \boldsymbol{\theta}_t). \tag{18}$$

Finally, if we optimize only one $\boldsymbol{\theta}_t$ at a time and $w_t = \frac{1}{t}$, then several terms in Eq. 17 can be ignored, such as $\mathrm{EWC}_{\boldsymbol{\theta}_0}(\boldsymbol{\theta}_{t'})$ for $t' < t$. Hence, minimizing Eq. 18 is equivalent to minimize:

$$\mathbb{E}_{\boldsymbol{x}, \boldsymbol{y} \sim p_t(\boldsymbol{x}, \boldsymbol{y})} [\ell_{\mathrm{cur}}(\boldsymbol{\theta}_{\mathcal{P}} | \boldsymbol{x}, \boldsymbol{y})] + \frac{\beta}{t} \left[ (1 - \frac{1}{t}) \frac{1}{2} \mathrm{EWC}_{\boldsymbol{\theta}_0}(\boldsymbol{\theta}_t) - \frac{1}{t} \sum_{t' < t} \boldsymbol{\tau}_t^{\mathrm{T}} \hat{\mathrm{F}}_{\boldsymbol{\theta}_0} \boldsymbol{\tau}_{t'} \right]. \tag{19}$$

As shown in App. C for both full and LoRA fine-tuning, the gradients of the regularization term in Eq. 19 can be computed in **closed form**. Importantly, this derivation enables a lightweight optimization, as the analytical gradients maintain **constant complexity** w.r.t. the number of tasks.

---

**Algorithm 1** Incremental Task Arithmetic (ITA) *vs.* Incremental Ensemble Learning (IEL)

---

**Input:** $T$ disjoint classification tasks $D_t = \{(\boldsymbol{x}, \boldsymbol{y})\}_{n_t}$, a pre-trained DNN $f(\cdot; \boldsymbol{\theta}_0)$, learning rate $\texttt{lr}$, hyper-parameters $\alpha$ and $\beta$, initialized pool $\mathcal{P} = \emptyset$ and the diagonal Fisher $\hat{F}_{\boldsymbol{\theta}_0} = \mathbf{0}$.

**for** each task $t \in \{1, 2, \ldots, T\}$ **do**

  $\boldsymbol{h}_0^{(t)} \leftarrow$ Linear Probing on $D_t$ with the pre-trained $f(\cdot; \boldsymbol{\theta}_0)$                 Task pre-consolidation

  $\boldsymbol{\theta}_0 \leftarrow \boldsymbol{\theta}_0 \cup \boldsymbol{h}_0^{(t)}$                 ▷ add the pre-training classification head

  $\hat{F}_{\boldsymbol{\theta}_0} \leftarrow \hat{F}_{\boldsymbol{\theta}_0} + \hat{F}_{\boldsymbol{\theta}_0}^{(t)}$       ▷ update the global Fisher with the local Fisher $\hat{F}_{\boldsymbol{\theta}_0}^{(t)}$ estimated on $D_t$

  $\boldsymbol{\tau}_t \leftarrow \mathcal{N}(\mu_{\text{init}}, \sigma_{\text{init}})$                                   Fine-tuning

  $\mathcal{P} \leftarrow \mathcal{P} \cup \{f(\cdot; \boldsymbol{\theta}_t = \boldsymbol{\theta}_0 + \boldsymbol{\tau}_t)\}$          ▷ extend $\mathcal{P}$ with the weights of the $t$-th learner

  **for** each example $(\boldsymbol{x}, \boldsymbol{y})$ **in** $D_t$ **do**

    $p_{\boldsymbol{\theta}}(\boldsymbol{y}|\boldsymbol{x}) \leftarrow f(\boldsymbol{x}; \boldsymbol{\theta}_t)$               ▷ predict with the $t$-th learner $\boldsymbol{\theta}_t \leftarrow \boldsymbol{\theta}_0 + \boldsymbol{\tau}_t$

    $p_{\boldsymbol{\theta}}(\boldsymbol{y}|\boldsymbol{x}) \leftarrow f(\boldsymbol{x}; \boldsymbol{\theta}_{\mathcal{P}})$     ▷ predict with the composed model $\boldsymbol{\theta}_{\mathcal{P}} \leftarrow \boldsymbol{\theta}_0 + \frac{1}{t} \sum_{t'=1}^{t} \boldsymbol{\tau}_{t'}$

    $\ell \leftarrow -\log p_{\boldsymbol{\theta}}(\boldsymbol{y}|\boldsymbol{x})$

    $\boldsymbol{\tau}_t \leftarrow \boldsymbol{\tau}_t - \texttt{lr} \cdot \nabla_{\boldsymbol{\tau}_t}[\ell + \frac{\alpha}{2} \text{EWC}_{\boldsymbol{\theta}_0}(\boldsymbol{\theta}_t)]$       ▷ arithmetic-oriented regularization (Eq. 11)

    $\boldsymbol{\tau}_t \leftarrow \boldsymbol{\tau}_t - \texttt{lr} \cdot \nabla_{\boldsymbol{\tau}_t}[\ell + \beta \Omega_{\hat{F}}(\mathcal{P})]$       ▷ ensemble-oriented regularization (Eqs. 18 and 19)

---

## 3   ALGORITHM(S)

We present **Incremental Task Arithmetic** (**ITA**) and **Incremental Ensemble Learning** (**IEL**), two **distinct** algorithms for *individual* and *ensemble* incremental learning respectively. As shown in Alg. 1, they divide each task into the *pre-consolidation* and *fine-tuning* stages, with differences occurring in the latter. We direct the reader to App. E for comprehensive implementation guidelines.

**Task pre-consolidation.** Due to the pivotal role of the local optimality of $\boldsymbol{\theta}_0$ (Sec. 2.1), we now follow up on this aspect and consider *closed vocabulary* models. Unlike models like CLIP (Radford et al., 2021), closed vocabulary models require the addition of a tailored classification head to handle novel classes. In line with Ortiz-Jimenez et al. (2024), we denote the process of fine-tuning only the added classification head as **linear probing** (**LP**): specifically, LP keeps the rest of the layers fixed to the pre-training $\boldsymbol{\theta}_0$. In our work, we basically exploit linear probing to enforce the optimality of pre-training weights $\boldsymbol{\theta}_0$. Namely, during the pre-consolidation phase, we conduct a few preliminary training epochs on the $t$-th incoming task, with the sole purpose of fine-tuning the new classification head. From that point onward, the fine-tuned head $\boldsymbol{h}_0^t$ is regarded as a part of pre-training weights $\boldsymbol{\theta}_0$. Finally, the pre-consolidation stage concludes with the update of the diagonal Fisher matrix $\hat{F}_{\boldsymbol{\theta}_0}$.

**Fine-tuning.** While the pre-consolidation step is identical for both ITA and IEL, they differ during the fine-tuning phase. The shared goal is to learn a task vector $\boldsymbol{\tau}_t$ s.t. $\boldsymbol{\theta}_t \equiv \boldsymbol{\theta}_0 + \boldsymbol{\tau}_t$ for the current task. However, ITA treats $\boldsymbol{\tau}_t$ as the weights of an individual model, whereas IEL interprets it as a new learnable component of the composition. In other words, ITA computes predictions through the $t$-th learner $f(\boldsymbol{x}; \boldsymbol{\theta}_t)$, while IEL leverages the composed function $f(\boldsymbol{x}; \boldsymbol{\theta}_{\mathcal{P}})$. Moreover, their regularizing objectives differ: ITA builds upon Eq. 11 (*i.e.*, the additional EWC-like term computed w.r.t. $\boldsymbol{\theta}_0$), while IEL exploit Eqs. 18 and 19 to train the composed model. Notably, both approaches can be applied to any fine-tuning strategy in the form of $\boldsymbol{\theta}_0 + \Delta\boldsymbol{\theta}$. We conduct experiments on Full Fine-Tuning (*i.e.*, $\boldsymbol{\tau}_t \in \mathbb{R}^{F_{OUT} \times F_{IN}}$), Low-Rank Adaptation (LoRA) (Hu et al., 2022) (*i.e.*, $\boldsymbol{\tau}_t = B_t A_t$), and on $(\text{IA})^3$ (Liu et al., 2022). About the latter, let $h$ represent the hidden dimension and $l \in \mathbb{R}^h$ denote the $(\text{IA})^3$ task-specific vector: it can be shown that $(\text{IA})^3$ is equivalent to $\boldsymbol{\theta}_t \equiv \boldsymbol{\theta}_0 + \boldsymbol{\tau}_t$ with $\boldsymbol{\tau}_t = \boldsymbol{\theta}_0 \odot ((l - \mathbf{1}_h) \otimes \mathbf{1}_h)$, where $\mathbf{1}_h$ is the all-ones vector. For each fine-tuning strategy, we initialize their parameters so that the resulting task vector $\boldsymbol{\tau}_t$ starts as a null vector at the beginning.

**Computational analysis.** As outlined in App. D, by treating the composed model as a cumulative average of individual models, both training/inference stages of ITA/IEL maintain constant complexity $\mathcal{O}(1)$ with respect to the number of tasks $T$. Indeed, a single forward pass is sufficient to compute the output of the composed model (**constant time**). Moreover, we do not need to store a separate set of weights for each task (**constant memory**), provided we are not interested in more complex forms of composition than the simplest uniform average (as required for specialization and unlearning).

## 4 RELATION WITH EXISTING WORKS

**Task arithmetic.** Standard and Parameter-Efficient (PEFT) fine-tuning have been shown to support addition/subtraction of task vectors. However, while the evidence in (Zhang et al., 2024; Ilharco et al., 2023) is primarily empirical, we derive theoretical insights about the pre-conditions for task arithmetic, emphasizing the importance of staying close to the pre-training basin. In this respect, our derivations ground previous findings regarding the efficacy of low learning rates (Ilharco et al., 2023; Ortiz-Jimenez et al., 2024). Remarkably, staying within the pre-training basin has also been proved beneficial in (Sadrtdinov et al., 2024) for ensemble learning. The conditions for compositionality are also studied by Ortiz-Jimenez et al. (2024) on *linearized* networks (Eq. 7). Albeit considering this work inspirational, we see the pros of task arithmetic in the non-linear regime. Firstly, non-linear models surpass their linearized counterparts in single-task accuracy, making them more attractive. To reduce the gap, linearization-aware fine-tuning has to be used (Liu & Soatto, 2023), contrarily to our approach that is compatible with the prevalent fine-tuning techniques. Secondly, linearized inference requires the demanding Jacobian-vector product (three times slower than a forward pass).

**Ensemble learning.** While original model soups (Wortsman et al., 2022) combine multiple weights fine-tuned on the same dataset, we herein managed to unlock *running model soups* in cross-dataset incremental settings, with possible returns in terms of forward transfer (Lopez-Paz & Ranzato, 2017). The optimization of the whole deep ensemble is also discussed in Jeffares et al. (2024) for standard ensembles, *i.e.*, averaging the outputs of different models. Their derivation is similar to ours and regards the decomposition of the ensemble loss into the strength of the individual learners and their diversity. However, Jeffares et al. (2024) use this result to elucidate the shortcomings of jointly trained deep ensembles, whereas we leverage it to provide effective regularization for model soups in incremental scenarios. Similar to our IEL, several works (Li et al., 2022; 2023a; Schmidt et al., 2023) build an ensemble through a cumulative mean of intermediate checkpoints sampled along the training trajectory, with benefits in terms of generalization and preservation of zero-shot pre-training capabilities (Zheng et al., 2023). Differently, Jolicoeur-Martineau et al. (2024) maintain a population of models trained with varying configurations (*e.g.*, data augmentations). They also gradually push each weight toward the population average, thus encouraging alignment across individual models. Notably, such a behaviour is also positively rewarded by our second-order formulation (see Eq. 17).

**Incremental learning.** Adding new knowledge in deep networks often degrades the ability on earlier tasks. Approaches against forgetting can be divided into three groups: those involving *regularization* (Kirkpatrick et al., 2017; Zenke et al., 2017), those retaining old data for *rehearsal* (Lopez-Paz & Ranzato, 2017; Aljundi et al., 2019), and those allocating new modules (Mallya & Lazebnik, 2018; Abati et al., 2020). Among the latter, SEED (Rypeść et al., 2024) manages an ensemble of expert networks learned incrementally, sharing similarities with our IEL. However, SEED stores separate models and combines their outputs, whereas our approach maintains a cumulative weight average of past experts, minimizing memory and computational overhead. This running average can also be achieved by InfLoRA (Liang & Li, 2024), which addresses interference among LoRA modules through tailored initialization. In contrast, our method addresses this issue through explicit regularization. In addition, a recent trend capitalizes on prompt-tuning (Wang et al., 2022a;b; Smith et al., 2023), devising a pool of learnable prompts. Notably, the extent to which these models support compositionality is investigated in Perera et al. (2023). Bowman et al. (2023) propose À-la-carte Prompt Tuning (APT), an attention mechanism that enables the creation of bespoke models by composing arbitrary prompts. Their setting called *à-la-carte* aligns with the scope of our ITA; however, our work extends to a broader range of PEFT modules, unlike APT that is limited to soft prompts.

**NTK-based Incremental learning.** The authors of Tangent Model Composition (TMC) (Liu & Soatto, 2023) build on the foundational work of Ortiz-Jimenez et al. (2024) to address incremental learning. They enforce task arithmetic across subsequent tasks through linearization-aware fine-tuning, which entails a first-order Taylor approximation of the output function around $\boldsymbol{\theta}_0$. In this context, each task of TMC is effectively equivalent to training a kernel predictor using the **Neural Tangent Kernel** (**NTK**) (Jacot et al., 2018), defined as $k_{\mathrm{NTK}}(x, x') = \nabla_{\boldsymbol{\theta}} f(x; \boldsymbol{\theta}_0)^\top \nabla_{\boldsymbol{\theta}} f(x'; \boldsymbol{\theta}_0)$. In contrast, our approach employs a higher-order approximation of the empirical risk, thereby focusing on the geometry of the loss landscape (Chaudhry et al., 2018). Notably, other recent works (Liu et al., 2024) have adopted the NTK framework to tackle incremental learning: *e.g.*, TKIL (Xiang & Shlizerman, 2023) exploit the NTK formulation to align representations for current and past tasks.

Table 1: Comparison with SOTA (Final Accuracy [↑]). Best results in **bold**, second-best underlined. EWC, LwF-MC, DER++ (buffer size of $1,000$ examples), SEED, and TMC rely on full fine-tuning; L2P, CODA, and APT utilize prompt-based learning. Finally, InfLoRA adopts LoRA fine-tuning.

| Model | IN-R | C-100 | CUB | Caltech | MIT | RESISC | CropDis. |
|---|---|---|---|---|---|---|---|
| Joint | 86.08 | 91.74 | 87.12 | 93.21 | 87.19 | 96.79 | 99.71 |
| Finetune | 22.31 | 21.57 | 29.35 | 44.03 | 18.85 | 12.90 | 20.54 |
| EWC | 58.64 | 73.49 | 40.33 | 73.23 | 64.44 | 58.80 | 70.33 |
| LWF-MC | 50.93 | 72.16 | 21.88 | 79.73 | 67.04 | 68.09 | 78.28 |
| DER++ | 60.53 | 83.02 | 76.10 | 86.11 | 68.88 | 67.23 | **98.77** |
| L2P | 67.17 | 87.32 | 78.95 | 91.22 | 83.17 | 63.47 | 75.18 |
| CODA | 74.12 | 86.48 | 78.54 | 90.57 | 77.73 | 69.50 | 74.65 |
| SEED | 55.87 | 83.39 | 85.35 | 90.04 | 86.34 | 74.81 | 92.77 |
| APT | 65.32 | 86.19 | 69.51 | 87.71 | 75.83 | 49.99 | 59.37 |
| InfLoRA | 76.97 | 87.17 | 79.14 | 90.53 | 79.14 | 79.92 | 89.05 |
| TMC | 60.01 | 78.42 | 71.72 | 82.30 | 68.66 | 60.66 | 66.56 |
| ITA-FFT | 76.43 | 89.38 | 84.80 | 92.32 | 85.35 | 80.50 | 91.81 |
| ITA-LoRA | 77.79 | 89.96 | 85.55 | 92.65 | **86.60** | 82.00 | 95.85 |
| ITA-$(\text{IA})^3$ | 77.04 | **90.66** | **85.67** | 92.67 | 84.74 | **83.73** | 95.41 |
| IEL-FFT | **80.09** | 89.38 | 84.89 | 92.23 | 82.79 | 81.42 | 95.83 |
| IEL-LoRA | 79.93 | 89.53 | 84.95 | 92.19 | 84.49 | 82.53 | 95.88 |
| IEL-$(\text{IA})^3$ | 77.86 | 89.72 | 84.57 | **92.70** | 85.54 | 81.50 | 95.68 |

## 5 EXPERIMENTS

**Datasets.** Following affine works (Wang et al., 2022b; Bowman et al., 2023; Liu & Soatto, 2023), we evaluate on these **class-incremental** benchmarks: Split ImageNet-R (Hendrycks et al., 2021) (10 tasks × 20 classes each), Split CIFAR-100 (Krizhevsky et al., 2009) (10 tasks × 10 classes), Split CUB-200 (Wah et al., 2011) (10 tasks × 20 classes), Split Caltech-256 (Griffin et al., 2007) (10 tasks, as in (Liu & Soatto, 2023)), and Split MIT-67 (Quattoni & Torralba, 2009) (10 tasks, as in (Liu & Soatto, 2023)). We conduct further tests on the **aerial** and **medical** domains using Split RESISC45 (Cheng et al., 2017) (9 tasks × 5 classes) and Split CropDiseases (Hughes et al., 2015) (7 tasks × 5 classes). They provide a challenging benchmark for our proposals due to their **low domain similarity** with the ImageNet pre-training (Oh et al., 2022). Further details can be found in App. G.

**Benchmarking.** We compare against recognized incremental methods as EWC (Kirkpatrick et al., 2017), LwF-MC (Rebuffi et al., 2017), DER++ (Buzzega et al., 2020b), L2P (Wang et al., 2022b), and CODA (Smith et al., 2023). We also asses SEED (Rypeść et al., 2024), InfLoRA (Liang & Li, 2024), APT (Bowman et al., 2023), and TMC (Liu & Soatto, 2023), four approaches featuring compositionality and ensemble learning. Their description and the main differences from our approaches are provided in App. F. All methods, including ours, utilize the **same backbone** – a ViT-B/16 (Dosovitskiy et al., 2021) with supervised pre-training on ImageNet21K (Ridnik et al., 2021) – and the same batch size (128). We compute the accuracy on all classes at the end of the final task (Final Accuracy, FA). Following Buzzega et al. (2020b), the **hyperparameters** are chosen through a grid search on a validation set (*i.e.*, 10% of the training set). This procedure was repeated for each method, ensuring careful tuning of the search space. See the supplementary for *1)* the **standard deviation** of the FA (the results are averaged over three runs, see App. H.2); *2)* the results expressed as Final Forgetting (Chaudhry et al., 2018) (App. H.1); *3)* the selected hyper-parameters (App. I).

**Comparison with SOTA.** As reported in Tab. 1, ITA and IEL outperform existing approaches on all datasets except MIT-67 and CropDisease. At times, they match SEED; however, its reliance on Mixture of Experts makes its inference computationally demanding. Our results far exceed those of TMC, showcasing the potential of the non-linear regime over linearization. Considering the good results on RESISC and CropDis., ITA and IEL do not seem affected by large domain shifts, indicating that our formulation remains effective even when the pre-training optimality is challenged. Finally, given the comparable results of ITA and IEL, we recommend ITA as the preferred starting point for future research, in light of the greater flexibility offered by the individual training paradigm.

Table 2: For ITA, analysis of the impact of the proposed regularization loss (FA [↑]).

| Model | IN-R | C-100 | CUB | Caltech | MIT | RESISC | CropDis. |
|---|---|---|---|---|---|---|---|
| **ITA-FFT** *(reg)* | **76.43** | **89.38** | **84.80** | **92.32** | **85.35** | **80.50** | **91.81** |
| *without Eq.* 12 *reg.* | 8.61 | 17.59 | 10.47 | 12.76 | 12.01 | 17.17 | 20.64 |
| *Eq.* 12 *only on* CLS | 76.00 | 87.60 | 83.54 | 91.04 | 81.45 | 75.26 | 77.00 |
| **ITA-LoRA** *(reg)* | **77.79** | 89.96 | **85.55** | **92.65** | **86.60** | **82.00** | 95.85 |
| *without Eq.* 12 *reg.* | 50.17 | 66.58 | 60.58 | 74.87 | 52.74 | 37.59 | 55.86 |
| *Eq.* 12 *only on* CLS | 77.33 | **90.03** | **85.55** | 92.59 | 84.86 | 80.64 | **96.22** |

**On the impact of regularization.** Tab. 2 reports the results of ITA when the regularization term in Eq. 12 is removed (the same analysis for ITA-`(IA)`³ and IEL is available in App. H). To evaluate the effect on distinct layers, we additionally assess the model with only the last classification layer regularized (see *Eq.* 12 *only on* CLS in Tab. 2). As observed: *i)* applying Eq. 12 is beneficial for all examined fine-tuning strategies; *ii)* although regularizing all layers is the most consistent approach, applying Eq. 12 only on the classification head already yields good accuracy. This suggests that compositionality in closed-set models is largely dependent on the learning dynamics of the last layer. Finally, full fine-tuning (FFT) struggles the most when no regularization or partial regularization is applied, in contrast to PEFT modules like LoRA and `(IA)`³ that still manage to achieve decent results. We ascribe this evidence to the tendency of PEFT modules to forget less of the pre-trained knowledge (Biderman et al., 2024), a feature we identify as beneficial for model compositionality.

To gain insights into our regularization, we revisit the foundations of ITA: specifically the inequality in Eq. 5, which states that the risk of the composed model $\hat{\ell}_{\mathrm{cur}}(\boldsymbol{\theta}_\mathcal{P})$ is upper bounded by the weighted risk of the individual models $\sum w_t \hat{\ell}_{\mathrm{cur}}(\boldsymbol{\theta}_t)$. In practice, there is no guarantee that this upper bound is tight: as these individual models are trained on disjoint tasks, their risk is likely high for examples outside their training distribution. We hence expect the upper bound to be loose: this is shown in Fig. 1, where we measure the effect of our regularization on the empirical risk of the composed model ⬜ and on the upper bound ▨.

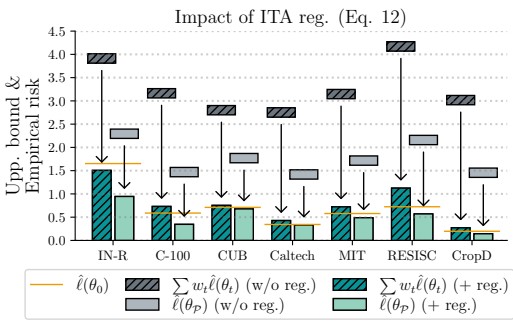

Figure 1: Effect of ITA. Best viewed in color.

From this lens, ITA significantly tightens the upper bound, with a remarkable reduction of the risk 🟩 associated with the composed model $\hat{\ell}_{\mathrm{cur}}(\boldsymbol{\theta}_\mathcal{P})$. Notably, the latter consistently surpasses the pre-trained model $\hat{\ell}_{\mathrm{cur}}(\boldsymbol{\theta}_0)$ ── (obtained with linear probing). This indicates that there is a good margin to find a point in parameter space that improves pre-training. Vice versa, the upper bound $\sum w_t \hat{\ell}_{\mathrm{cur}}(\boldsymbol{\theta}_t)$ does not exceed pre-training, even with regularization applied, raising the theoretical possibility of *regret* between the composed model and pre-training. This work provides empirical evidence to address it but a theoretical analysis is needed, which we leave for future works.

**On the compositional skills of ITA and IEL.** We herein investigate *specialization* and *unlearning* capabilities of our approaches. Specifically, given the task vectors trained on the sequence of tasks, we freeze and use them to edit the composed model. As these adaptations involve simple additions and subtractions of task vectors, we can perform *specialization* and *unlearning* in a zero-shot fashion requiring no examples. In particular, through the former test on specialization, we examine whether composing only a subset of the task vectors can boost the composed model on the respective selected tasks. In doing so, we focus on three tasks of the incremental sequence *i.e.*, the first, the central, and the last one. The performance is then measured considering the average FA across the tasks we wish to specialize in (FA$_{\mathrm{TGT}}$) and the average FA across the others (FA$_{\mathrm{CTRL}}$). In the second test, instead, our goal is to eliminate knowledge of a specific task without degrading performance on other tasks (*unlearning*). Specifically, to unlearn a given task, we subtract its associated task vector from the composition, which still includes the weights of the tasks that need to be preserved. Afterwards: *i)* we measure the accuracy for the unlearned task (FA$_{\mathrm{TGT}}$) and the others (FA$_{\mathrm{CTRL}}$); *ii)* the procedure is then repeated for all tasks, with the performance averaged across them.

Table 3: Analysis of compositional capabilities – see App. H for results on other datasets. In parentheses, we report the gain (or loss) in accuracy on the target task.

| Dataset | Model | zero-shot specialization | | zero-shot unlearning | |
|---|---|---|---|---|---|
| | | $FA_{TGT}$ [↑] | $FA_{CTRL}$ | $FA_{TGT}$ [↓] | $FA_{CTRL}$ |
| **IN-R** | ITA-LoRA | 80.83 (+11.40) | 50.52 | 22.77 (−55.02) | 52.72 (−25.07) |
| | IEL-LoRA | 73.46 (−06.68) | 38.46 | 18.55 (−61.38) | 41.97 (−37.96) |
| | TMC | 69.93 (+08.36) | 34.08 | 45.77 (−14.24) | 54.37 (−05.64) |
| **C-100** | ITA-LoRA | 92.80 (+01.63) | 60.06 | 28.67 (−61.29) | 71.96 (−17.99) |
| | IEL-LoRA | 77.77 (−13.22) | 37.90 | 19.48 (−70.05) | 56.52 (−33.01) |
| | TMC | 87.53 (+06.49) | 45.75 | 55.63 (−22.79) | 71.83 (−06.59) |

The results are in Tab. 3 and Tab. 7 of App. H, where we compare with TMC by Liu & Soatto (2023), *i.e.*, model linearization and tangent fine-tuning. Regarding **specialization**, both ITA and TMC improve the accuracy on the target tasks. IEL, instead, leads to a severe drop during specialization. This yields an interesting finding: while Tab. 1 highlights its performance as satisfying, we observe that the ensemble struggles when any of its members are removed. In an era where ensemble models are experiencing renewed attention (Liang & Li, 2024; Marouf et al., 2024), this evidence should be sobering. On the other hand, we underline the specialization capabilities of ITA, with the best absolute performance on the target tasks. We ascribe it to the devised regularization objective and the natural advantages of the non-linear regime. When focusing on **unlearning**, ITA shows mixed results. Notably, both ITA and TMC consistently reduce the performance of the task to be forgotten. While this also affects the tasks that should be preserved, ITA exhibits a greater distinction between the target and control tasks, indicating a more effective disentanglement of knowledge across tasks.

## 6 DISCUSSION OF LIMITATIONS AND FUTURE DIRECTIONS

Second-order Taylor approximations are valid tools to make the theory more tractable. For instance, a similar approach was used by Mirzadeh et al. (2020) to support the importance of wider local minima against forgetting. Nonetheless, these approximations often rely on certain concessions, the first regarding their validity during training. Since our approximation is performed at the pre-training weights, it may become inaccurate as parameters drift. While importance-based terms like ours may partially mitigate the drift, other techniques could be used (*e.g.*, a L2-norm penalty on task vectors or low learning rate). Even without explicit countermeasures, existing studies suggest that deep networks often fall into a regime where the loss tends to be almost convex in a neighborhood around their local minima (Goodfellow et al., 2015; Lucas et al., 2021; Yunis et al., 2022). Also, under some conditions, their training dynamics can enter the lazy regime (Chizat et al., 2019; Jacot et al., 2018), where these models rapidly achieve near-zero loss with minimal changes to their weights.

Moreover, two other concessions warrant a discussion: *approx. i)* we implicitly model the weight distribution with a Gaussian (Chaudhry et al., 2018) (Eq. 9); *approx. ii)* the Fisher matrix is approximated by its diagonal (Eq. 11). Regarding *approx. i)*, while non-Gaussian posteriors are often cumbersome to manage, Farquhar et al. (2020) challenge the common belief that mean-field approximations are overly restrictive for deep networks. Moreover, although *approx. ii)* may seem crude, the diagonal approximation is efficient, with a low memory footprint. Still, our approach could profit from more accurate estimations of the Hessian, like the Kronecker factored approximation (Martens & Grosse, 2015), which considers the interactions between weights of the same layer.

**Future work.** Through theoretical and empirical analyses, we support the importance of remaining within the pre-training basin to achieve composable deep networks. However, there is still much to explore along this path. We mainly focus on closed-set classification models but it would be noteworthy to extend our analysis to self-supervised pre-training and open-vocabulary models like CLIP. Indeed, recent works (Zheng et al., 2023) have shown their tendency to forget zero-shot pre-training capabilities while fine-tuning. In this respect, our second-order regularization could significantly aid compositionality in CLIP-based models. Finally, we believe that staying within the pre-train basin is only one aspect to consider; future research should emphasize the **exploration** of the pre-train basin, to achieve composable modules with a higher degree of specialization on their respective tasks.

ACKNOWLEDGMENTS

This work has been supported by the PNRR-M4C2 (PE00000013) project "FAIR - Future Artificial Intelligence Research", funded by the European Commission, and by the Italian Ministerial grant PRIN 2020 "LEGO.AI: LEarning the Geometry of knOwledge in AI systems", n. 2020TA3K9N. We acknowledge the CINECA award under the ISCRA initiative, for the availability of high-performance computing resources and support.

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

## A  APPENDIX / SUPPLEMENTAL MATERIAL

The appendix is organized via the following contributions:

- App. B provides the proofs for Theorem 1 and Eq. 17.
- App. C illustrates that gradients of Eq. 19 can be derived in closed-form.
- App. D reports the computational analysis for our approaches Incremental Task Arithmetic (ITA) and Incremental Ensemble Learning (IEL).
- App. E describes the essential details required to reproduce the methods described in the main paper.
- Apps. F and G supplements the main paper with additional information regarding the relation with existing works and the adopted benchmarks.
- App. H provides additional experimental results: the forgetting metric, the standard deviation measured in each experiment, an additional ablation study regarding IEL, an extensive evaluation of model compositionality across multiple datasets, an analysis of the similarities between task vectors learned by ITA and IEL, and a timing analysis.
- App. I outlines the hyperparameter configurations adopted in our experiments.

## B  PROOFS

### B.1  PROOF OF THEOREM 1

*Proof.* For the sake of notation, we will use the shortcuts $\ell_0 \triangleq \ell(\boldsymbol{\theta}_0)$, $\mathbf{J}_\ell \triangleq \nabla\ell(\boldsymbol{\theta}_0)$, and $\mathbf{H}_\ell \triangleq \mathbf{H}_\ell(\boldsymbol{\theta}_0)$. Given a setting with only two learners $A$ and $B$, the second-order approximation of the loss function of the composed model $\mathcal{P}$ is:

$$\ell_{\text{cur}}(\boldsymbol{\theta}_\mathcal{P}) = \ell_0 + (\boldsymbol{\theta}_\mathcal{P} - \boldsymbol{\theta}_0)^\mathrm{T}\mathbf{J}_\ell + \frac{1}{2}(\boldsymbol{\theta}_\mathcal{P} - \boldsymbol{\theta}_0)^\mathrm{T}\mathbf{H}_\ell(\boldsymbol{\theta} - \boldsymbol{\theta}_0)$$

$$= \underbrace{(w_a + w_b)}_{=1}\ell_0 + (\underbrace{\boldsymbol{\theta}_\mathcal{P} - \boldsymbol{\theta}_0}_{w_a\boldsymbol{\tau}_a + w_b\boldsymbol{\tau}_b})^\mathrm{T}\mathbf{J}_\ell + \frac{1}{2}(\boldsymbol{\theta}_\mathcal{P} - \boldsymbol{\theta}_0)^\mathrm{T}\mathbf{H}_\ell(\boldsymbol{\theta} - \boldsymbol{\theta}_0)$$

$$= w_a\ell_0 + w_b\ell_0 + (w_a\boldsymbol{\tau}_a + w_b\boldsymbol{\tau}_b)^\mathrm{T}\mathbf{J}_\ell + \frac{1}{2}(w_a\boldsymbol{\tau}_a + w_b\boldsymbol{\tau}_b)^\mathrm{T}\mathbf{H}_\ell(w_a\boldsymbol{\tau}_a + w_b\boldsymbol{\tau}_b)$$

$$= w_a\ell_0 + w_b\ell_0 + w_a\boldsymbol{\tau}_a^\mathrm{T}\mathbf{J}_\ell + w_b\boldsymbol{\tau}_b^\mathrm{T}\mathbf{J}_\ell + \frac{1}{2}(w_a\boldsymbol{\tau}_a^\mathrm{T} + w_b\boldsymbol{\tau}_b^\mathrm{T})\mathbf{H}_\ell(w_a\boldsymbol{\tau}_a + w_b\boldsymbol{\tau}_b)$$

$$= w_a[\ell_0 + \boldsymbol{\tau}_a^\mathrm{T}\mathbf{J}_\ell] + w_b[\ell_0 + \boldsymbol{\tau}_b^\mathrm{T}\mathbf{J}_\ell] + \frac{1}{2}(w_a\boldsymbol{\tau}_a^\mathrm{T}\mathbf{H}_\ell + w_b\boldsymbol{\tau}_b^\mathrm{T}\mathbf{H}_\ell)(w_a\boldsymbol{\tau}_a + w_b\boldsymbol{\tau}_b)$$

$$= w_a[\ell_0 + \boldsymbol{\tau}_a^\mathrm{T}\mathbf{J}_\ell] + w_b[\ell_0 + \boldsymbol{\tau}_b^\mathrm{T}\mathbf{J}_\ell] + \frac{1}{2}(w_a^2\boldsymbol{\tau}_a^\mathrm{T}\mathbf{H}_\ell\boldsymbol{\tau}_a + w_b^2\boldsymbol{\tau}_b^\mathrm{T}\mathbf{H}_\ell\boldsymbol{\tau}_b +$$
$$+ \; w_aw_b\boldsymbol{\tau}_a^\mathrm{T}\mathbf{H}_\ell\boldsymbol{\tau}_b + w_aw_b\boldsymbol{\tau}_b^\mathrm{T}\mathbf{H}_\ell\boldsymbol{\tau}_a)$$

Given that $\boldsymbol{\tau}_a^\mathrm{T}\mathbf{H}_\ell\boldsymbol{\tau}_b$ is a scalar and $(ABC)^\mathrm{T} = (C^\mathrm{T}B^\mathrm{T}A^\mathrm{T})$
$$\rightarrow \boldsymbol{\tau}_a^\mathrm{T}\mathbf{H}_\ell\boldsymbol{\tau}_b = (\boldsymbol{\tau}_a^\mathrm{T}\mathbf{H}_\ell\boldsymbol{\tau}_b)^\mathrm{T} = (\boldsymbol{\tau}_b^\mathrm{T}\underbrace{\mathbf{H}_\ell}_{\text{symm.}}{}^\mathrm{T}\boldsymbol{\tau}_a) = \boldsymbol{\tau}_b^\mathrm{T}\mathbf{H}_\ell\boldsymbol{\tau}_a$$

$$\ell_{\text{cur}}(\boldsymbol{\theta}_\mathcal{P}) = w_a[\ell_0 + \boldsymbol{\tau}_a^\mathrm{T}\mathbf{J}_\ell] + w_b[\ell_0 + \boldsymbol{\tau}_b^\mathrm{T}\mathbf{J}_\ell] + \frac{1}{2}(w_a^2\boldsymbol{\tau}_a^\mathrm{T}\mathbf{H}_\ell\boldsymbol{\tau}_a + w_b^2\boldsymbol{\tau}_b^\mathrm{T}\mathbf{H}_\ell\boldsymbol{\tau}_b + 2w_aw_b\boldsymbol{\tau}_a^\mathrm{T}\mathbf{H}_\ell\boldsymbol{\tau}_b)$$

$$= w_a[\ell_0 + \boldsymbol{\tau}_a^\mathrm{T}\mathbf{J}_\ell] + w_b[\ell_0 + \boldsymbol{\tau}_b^\mathrm{T}\mathbf{J}_\ell] + \frac{1}{2}(\underbrace{w_a^2\boldsymbol{\tau}_a^\mathrm{T}\mathbf{H}_\ell\boldsymbol{\tau}_a}_{\substack{w_a^2 = w_a(1-w_b) \\ = w_a - w_aw_b}} + w_b^2\boldsymbol{\tau}_b^\mathrm{T}\mathbf{H}_\ell\boldsymbol{\tau}_b + 2w_aw_b\boldsymbol{\tau}_a^\mathrm{T}\mathbf{H}_\ell\boldsymbol{\tau}_b)$$

$$= w_a[\ell_0 + \boldsymbol{\tau}_a^\mathrm{T}\mathbf{J}_\ell + \frac{1}{2}\boldsymbol{\tau}_a^\mathrm{T}\mathbf{H}_\ell\boldsymbol{\tau}_a] + w_b[\ell_0 + \boldsymbol{\tau}_b^\mathrm{T}\mathbf{J}_\ell] + \frac{1}{2}(w_a\boldsymbol{\tau}_a^\mathrm{T}\mathbf{H}_\ell\boldsymbol{\tau}_a + w_b\boldsymbol{\tau}_b^\mathrm{T}\mathbf{H}_\ell\boldsymbol{\tau}_b +$$
$$- w_aw_b\boldsymbol{\tau}_a^\mathrm{T}\mathbf{H}_\ell\boldsymbol{\tau}_a - w_aw_b\boldsymbol{\tau}_b^\mathrm{T}\mathbf{H}_\ell\boldsymbol{\tau}_b + 2w_aw_b\boldsymbol{\tau}_a^\mathrm{T}\mathbf{H}_\ell\boldsymbol{\tau}_b)$$

$$
\begin{aligned}
&= w_a[\ell_0 + \boldsymbol{\tau}_a^{\mathrm{T}}\mathbf{J}_\ell + \frac{1}{2}\boldsymbol{\tau}_a^{\mathrm{T}}\mathbf{H}_\ell\boldsymbol{\tau}_a] + w_b[\ell_0 + \boldsymbol{\tau}_b^{\mathrm{T}}\mathbf{J}_\ell + \frac{1}{2}\boldsymbol{\tau}_b^{\mathrm{T}}\mathbf{H}_\ell\boldsymbol{\tau}_b] + \\
&\quad - \frac{1}{2}w_a w_b(\boldsymbol{\tau}_a^{\mathrm{T}}\mathbf{H}_\ell\boldsymbol{\tau}_a + \boldsymbol{\tau}_b^{\mathrm{T}}\mathbf{H}_\ell\boldsymbol{\tau}_b - 2\boldsymbol{\tau}_a^{\mathrm{T}}\mathbf{H}_\ell\boldsymbol{\tau}_b) \\
&= w_a\ell_{\mathrm{cur}}(\boldsymbol{\theta}_A) + w_b\ell_{\mathrm{cur}}(\boldsymbol{\theta}_B) - \frac{1}{2}w_a w_b(\boldsymbol{\tau}_a^{\mathrm{T}}\mathbf{H}_\ell\boldsymbol{\tau}_a + \boldsymbol{\tau}_b^{\mathrm{T}}\mathbf{H}_\ell\boldsymbol{\tau}_b - 2\boldsymbol{\tau}_a^{\mathrm{T}}\mathbf{H}_\ell\boldsymbol{\tau}_b) \\
&= w_a\ell_{\mathrm{cur}}(\boldsymbol{\theta}_A) + w_b\ell_{\mathrm{cur}}(\boldsymbol{\theta}_B) - \frac{1}{2}w_a w_b(\boldsymbol{\tau}_a - \boldsymbol{\tau}_b)^{\mathrm{T}}\mathbf{H}_\ell(\boldsymbol{\tau}_a - \boldsymbol{\tau}_b).
\end{aligned}
$$

In the multiple learning setting, we have that $\boldsymbol{\theta}_{\mathcal{P}} = \boldsymbol{\theta}_0 + \boldsymbol{\tau}_{\mathcal{P}} = \boldsymbol{\theta}_0 + \sum_{t=1}^{T} w_t\boldsymbol{\tau}_t$ with $\sum_{t=1}^{T} w_t = 1$. Therefore:

$$
\begin{aligned}
\ell_{\mathrm{cur}}(\boldsymbol{\theta}_{\mathcal{P}}) &= \ell_0 + (\boldsymbol{\theta}_{\mathcal{P}} - \boldsymbol{\theta}_0)^{\mathrm{T}}\mathbf{J}_\ell + \frac{1}{2}(\boldsymbol{\theta}_{\mathcal{P}} - \boldsymbol{\theta}_0)^{\mathrm{T}}\mathbf{H}_\ell(\boldsymbol{\theta} - \boldsymbol{\theta}_0) \\
&= \underbrace{\sum_{t=1}^{T} w_t\,\ell_0}_{=1} + \boldsymbol{\tau}_{\mathcal{P}}^{\mathrm{T}}\mathbf{J}_\ell + \frac{1}{2}\boldsymbol{\tau}_{\mathcal{P}}^{\mathrm{T}}\mathbf{H}_\ell\boldsymbol{\tau}_{\mathcal{P}} = \sum_{t=1}^{T} w_t\ell_0 + (\sum_{t=1}^{T} w_t\boldsymbol{\tau}_t)^{\mathrm{T}}\mathbf{J}_\ell + \frac{1}{2}\boldsymbol{\tau}_{\mathcal{P}}^{\mathrm{T}}\mathbf{H}_\ell\boldsymbol{\tau}_{\mathcal{P}} \\
&= \sum_{t=1}^{T} w_t\left[\ell_0 + \boldsymbol{\tau}_t^{\mathrm{T}}\mathbf{J}_\ell\right] + \frac{1}{2}\boldsymbol{\tau}_{\mathcal{P}}^{\mathrm{T}}\mathbf{H}_\ell\boldsymbol{\tau}_{\mathcal{P}}.
\end{aligned}
$$

Let us now focus on the quadratic term:

$$
\begin{aligned}
\boldsymbol{\tau}_{\mathcal{P}}^{\mathrm{T}}\mathbf{H}_\ell\boldsymbol{\tau}_{\mathcal{P}} &= (\sum_{t=1}^{T} w_t\boldsymbol{\tau}_t)^{\mathrm{T}}\mathbf{H}_\ell(\sum_{t=1}^{T} w_t\boldsymbol{\tau}_t) = (\sum_{t=1}^{T} w_t\boldsymbol{\tau}_t^{\mathrm{T}}\mathbf{H}_\ell)(\sum_{t=1}^{T} w_t\boldsymbol{\tau}_t) \\
&= (\sum_{t=1}^{T} w_t\boldsymbol{\tau}_t^{\mathrm{T}}\mathbf{H}_\ell)(w_t\boldsymbol{\tau}_t + \sum_{t'\neq t} w_{t'}\boldsymbol{\tau}_{t'}) = \sum_{t=1}^{T} w_t^2\boldsymbol{\tau}_t^{\mathrm{T}}\mathbf{H}_\ell\boldsymbol{\tau}_t + w_t\boldsymbol{\tau}_t^{\mathrm{T}}\mathbf{H}_\ell\sum_{t'\neq t}^{T} w_{t'}\boldsymbol{\tau}_{t'} \\
&= \sum_{t=1}^{T} w_t^2\boldsymbol{\tau}_t^{\mathrm{T}}\mathbf{H}_\ell\boldsymbol{\tau}_t + \sum_{t=1}^{T} w_t\boldsymbol{\tau}_t^{\mathrm{T}}\mathbf{H}_\ell\sum_{t'\neq t}^{T} w_{t'}\boldsymbol{\tau}_{t'} \\
&= \sum_{t=1}^{T} w_t^2\boldsymbol{\tau}_t^{\mathrm{T}}\mathbf{H}_\ell\boldsymbol{\tau}_t + \sum_{t=1}^{T}\sum_{t'\neq t}^{T} w_t w_{t'}\boldsymbol{\tau}_t^{\mathrm{T}}\mathbf{H}_\ell\boldsymbol{\tau}_{t'} \\
&\quad \text{Since } \boldsymbol{\tau}_t^{\mathrm{T}}\mathbf{H}_\ell\boldsymbol{\tau}_{t'} = \boldsymbol{\tau}_{t'}^{\mathrm{T}}\mathbf{H}_\ell\boldsymbol{\tau}_t \text{ (symmetry)} \\
&= \sum_{t=1}^{T} w_t^2\boldsymbol{\tau}_t^{\mathrm{T}}\mathbf{H}_\ell\boldsymbol{\tau}_t + 2\sum_{t,t'<t}^{T} w_t w_{t'}\boldsymbol{\tau}_t^{\mathrm{T}}\mathbf{H}_\ell\boldsymbol{\tau}_{t'}
\end{aligned}
$$

Given that $w_t^2 = w_t \cdot w_t = w_t(1 - \sum_{t'\neq t}^{T} w_{t'}) = w_t - w_t\sum_{t'\neq t}^{T} w_{t'}$:

$$
\begin{aligned}
&= \sum_{t=1}^{T}\left(w_t - w_t\sum_{t'\neq t}^{T} w_{t'}\right)\boldsymbol{\tau}_t^{\mathrm{T}}\mathbf{H}_\ell\boldsymbol{\tau}_t + 2\sum_{t=1,t'<t}^{T} w_t w_{t'}\boldsymbol{\tau}_t^{\mathrm{T}}\mathbf{H}_\ell\boldsymbol{\tau}_{t'} \\
&= \sum_{t=1}^{T} w_t\boldsymbol{\tau}_t^{\mathrm{T}}\mathbf{H}_\ell\boldsymbol{\tau}_t - \sum_{t=1}^{T}\left(w_t\sum_{t'\neq t}^{T} w_{t'}\right)\boldsymbol{\tau}_t^{\mathrm{T}}\mathbf{H}_\ell\boldsymbol{\tau}_t + 2\sum_{t=1,t'<t}^{T} w_t w_{t'}\boldsymbol{\tau}_t^{\mathrm{T}}\mathbf{H}_\ell\boldsymbol{\tau}_{t'} \\
&= \sum_{t=1}^{T} w_t\boldsymbol{\tau}_t^{\mathrm{T}}\mathbf{H}_\ell\boldsymbol{\tau}_t - \sum_{t=1}^{T}\sum_{t'\neq t}^{T} w_t w_{t'}\boldsymbol{\tau}_t^{\mathrm{T}}\mathbf{H}_\ell\boldsymbol{\tau}_t + 2\sum_{t=1,t'<t}^{T} w_t w_{t'}\boldsymbol{\tau}_t^{\mathrm{T}}\mathbf{H}_\ell\boldsymbol{\tau}_{t'} \\
&= \sum_{t=1}^{T} w_t\boldsymbol{\tau}_t^{\mathrm{T}}\mathbf{H}_\ell\boldsymbol{\tau}_t - \sum_{t=1,t'<t}^{T} w_t w_{t'}(\boldsymbol{\tau}_t^{\mathrm{T}}\mathbf{H}_\ell\boldsymbol{\tau}_t + \boldsymbol{\tau}_{t'}^{\mathrm{T}}\mathbf{H}_\ell\boldsymbol{\tau}_{t'}) + 2\sum_{t=1,t'<t}^{T} w_t w_{t'}\boldsymbol{\tau}_t^{\mathrm{T}}\mathbf{H}_\ell\boldsymbol{\tau}_{t'}
\end{aligned}
$$

$$= \sum_{t=1}^{T} w_t \boldsymbol{\tau}_t^{\mathrm{T}} \mathbf{H}_\ell \boldsymbol{\tau}_t - \sum_{t=1,t'<t}^{T} w_t w_{t'} (\boldsymbol{\tau}_t^{\mathrm{T}} \mathbf{H}_\ell \boldsymbol{\tau}_t + \boldsymbol{\tau}_{t'}^{\mathrm{T}} \mathbf{H}_\ell \boldsymbol{\tau}_{t'}) - 2 w_t w_{t'} \boldsymbol{\tau}_t^{\mathrm{T}} \mathbf{H}_\ell \boldsymbol{\tau}_{t'}$$

$$= \sum_{t=1}^{T} w_t \boldsymbol{\tau}_t^{\mathrm{T}} \mathbf{H}_\ell \boldsymbol{\tau}_t - \sum_{t=1,t'<t}^{T} w_t w_{t'} (\boldsymbol{\tau}_t^{\mathrm{T}} \mathbf{H}_\ell \boldsymbol{\tau}_t + \boldsymbol{\tau}_{t'}^{\mathrm{T}} \mathbf{H}_\ell \boldsymbol{\tau}_{t'} - 2 \boldsymbol{\tau}_t^{\mathrm{T}} \mathbf{H}_\ell \boldsymbol{\tau}_{t'})$$

$$= \sum_{t=1}^{T} w_t \boldsymbol{\tau}_t^{\mathrm{T}} \mathbf{H}_\ell \boldsymbol{\tau}_t - \sum_{t=1,t'<t}^{T} w_t w_{t'} (\boldsymbol{\tau}_t - \boldsymbol{\tau}_{t'})^{\mathrm{T}} \mathbf{H}_\ell (\boldsymbol{\tau}_t - \boldsymbol{\tau}_{t'})$$

Therefore:

$$\ell_{\mathrm{cur}}(\boldsymbol{\theta}_{\mathcal{P}}) = \ell_0 + (\boldsymbol{\theta}_{\mathcal{P}} - \boldsymbol{\theta}_0)^{\mathrm{T}} \mathbf{J}_\ell + \frac{1}{2} (\boldsymbol{\theta}_{\mathcal{P}} - \boldsymbol{\theta}_0)^{\mathrm{T}} \mathbf{H}_\ell (\boldsymbol{\theta} - \boldsymbol{\theta}_0)$$

$$= \sum_{t=1}^{T} w_t \left[ \ell_0 + \boldsymbol{\tau}_t^{\mathrm{T}} \mathbf{J}_\ell \right] + \frac{1}{2} \boldsymbol{\tau}_{\mathcal{P}}^{\mathrm{T}} \mathbf{H}_\ell \boldsymbol{\tau}_{\mathcal{P}}.$$

$$= \sum_{t=1}^{T} w_t \left[ \ell_0 + \boldsymbol{\tau}_t^{\mathrm{T}} \mathbf{J}_\ell \right] + \frac{1}{2} \sum_{t=1}^{T} w_t \boldsymbol{\tau}_t^{\mathrm{T}} \mathbf{H}_\ell \boldsymbol{\tau}_t - \frac{1}{2} \sum_{t=1,t'<t}^{T} w_t w_{t'} (\boldsymbol{\tau}_t - \boldsymbol{\tau}_{t'})^{\mathrm{T}} \mathbf{H}_\ell (\boldsymbol{\tau}_t - \boldsymbol{\tau}_{t'})$$

$$= \sum_{t=1}^{T} w_t \left[ \ell_0 + \boldsymbol{\tau}_t^{\mathrm{T}} \mathbf{J}_\ell + \frac{1}{2} w_t \boldsymbol{\tau}_t^{\mathrm{T}} \mathbf{H}_\ell \boldsymbol{\tau}_t \right] - \frac{1}{2} \sum_{t=1,t'<t}^{T} w_t w_{t'} (\boldsymbol{\tau}_t - \boldsymbol{\tau}_{t'})^{\mathrm{T}} \mathbf{H}_\ell (\boldsymbol{\tau}_t - \boldsymbol{\tau}_{t'})$$

$$= \sum_{t=1}^{T} w_t \ell_{\mathrm{cur}}(\boldsymbol{\theta}_t) - \frac{1}{2} \sum_{t=1,t'<t}^{T} w_t w_{t'} (\boldsymbol{\tau}_t - \boldsymbol{\tau}_{t'})^{\mathrm{T}} \mathbf{H}_\ell (\boldsymbol{\tau}_t - \boldsymbol{\tau}_{t'}),$$

which ends the proof of Theorem 1. $\qquad \square$

## B.2 PROOF OF EQ. 17

*Proof.* In the dual-learner setting we have:

$$\Omega(A, B) = \frac{1}{2} w_A w_B (\boldsymbol{\tau}_A - \boldsymbol{\tau}_B)^{\mathrm{T}} \mathbf{H}_\ell(\boldsymbol{\theta}_0)(\boldsymbol{\tau}_A - \boldsymbol{\tau}_B).$$

If we replace the Hessian matrix with the diagonal Fisher matrix $\hat{\mathrm{F}}_{\boldsymbol{\theta}_0}$, we get:

$$\Omega(A, B) = \frac{1}{2} w_A w_B (\boldsymbol{\tau}_A - \boldsymbol{\tau}_B)^{\mathrm{T}} \hat{\mathrm{F}}_{\boldsymbol{\theta}_0} (\boldsymbol{\tau}_A - \boldsymbol{\tau}_B) =$$

$$= \frac{1}{2} w_A w_B \left[ \boldsymbol{\tau}_A^{\mathrm{T}} \hat{\mathrm{F}}_{\boldsymbol{\theta}_0} \boldsymbol{\tau}_A - 2 \boldsymbol{\tau}_A^{\mathrm{T}} \hat{\mathrm{F}}_{\boldsymbol{\theta}_0} \boldsymbol{\tau}_B + \boldsymbol{\tau}_B^{\mathrm{T}} \hat{\mathrm{F}}_{\boldsymbol{\theta}_0} \boldsymbol{\tau}_B \right].$$

If we focus on the first term inside the parenthesis, we obtain:

$$\boldsymbol{\tau}_A^{\mathrm{T}} \hat{\mathrm{F}}_{\boldsymbol{\theta}_0} \boldsymbol{\tau}_A = \boldsymbol{\tau}_A^{\mathrm{T}} \hat{\mathrm{F}}_{\boldsymbol{\theta}_0}^{1/2} \hat{\mathrm{F}}_{\boldsymbol{\theta}_0}^{1/2} \boldsymbol{\tau}_A = (\hat{\mathrm{F}}_{\boldsymbol{\theta}_0}^{1/2} \boldsymbol{\tau}_A)^{\mathrm{T}} (\hat{\mathrm{F}}_{\boldsymbol{\theta}_0}^{1/2} \boldsymbol{\tau}_A) = \| \hat{\mathrm{F}}_{\boldsymbol{\theta}_0}^{1/2} \boldsymbol{\tau}_A \|_2^2$$

$$= \sum_{i=1}^{|\theta_A|} \hat{\mathrm{F}}_{\boldsymbol{\theta}_0}^{(i)} (\boldsymbol{\tau}_A^{(i)})^2 = \sum_{i=1}^{|\theta_A|} \hat{\mathrm{F}}_{\boldsymbol{\theta}_0}^{(i)} (\boldsymbol{\theta}_A^{(i)} - \boldsymbol{\theta}_0^{(i)})^2 = \mathrm{EWC}_{\boldsymbol{\theta}_0}(\boldsymbol{\theta}_A).$$

Therefore, we can rewrite $\Omega(A, B)$ as:

$$\Omega(A, B) = \frac{1}{2} w_A w_B \left[ \mathrm{EWC}_{\boldsymbol{\theta}_0}(\boldsymbol{\theta}_A) + \mathrm{EWC}_{\boldsymbol{\theta}_0}(\boldsymbol{\theta}_B) - 2 \boldsymbol{\tau}_A^{\mathrm{T}} \hat{\mathrm{F}}_{\boldsymbol{\theta}_0} \boldsymbol{\tau}_B \right].$$

We now generalize to $T \geq 2$. Starting from Eq. 14 and replacing the Hessian with the diagonal Fisher approximation $\hat{\mathrm{F}}_{\boldsymbol{\theta}_0}$, we obtain:

$$\frac{1}{2} \sum_{t=1,t'<t}^{T} w_t w_{t'} (\boldsymbol{\tau}_t - \boldsymbol{\tau}_{t'})^{\mathrm{T}} \mathbf{H}_\ell (\boldsymbol{\tau}_t - \boldsymbol{\tau}_{t'}) \approx$$

$$\approx \frac{1}{2} \sum_{t=1,t'<t}^{T} w_t w_{t'} \left[ \mathrm{EWC}_{\boldsymbol{\theta}_0}(\boldsymbol{\theta}_t) + \mathrm{EWC}_{\boldsymbol{\theta}_0}(\boldsymbol{\theta}_{t'}) - 2\boldsymbol{\tau}_t^{\mathrm{T}} \hat{\mathrm{F}}_{\boldsymbol{\theta}_0} \boldsymbol{\tau}_{t'} \right]$$

$$= \frac{1}{2} \sum_{t=1,t'<t}^{T} w_t w_{t'} \left[ \mathrm{EWC}_{\boldsymbol{\theta}_0}(\boldsymbol{\theta}_t) + \mathrm{EWC}_{\boldsymbol{\theta}_0}(\boldsymbol{\theta}_{t'}) \right] - \sum_{t=1,t'<t}^{T} w_t w_{t'} \boldsymbol{\tau}_t^{\mathrm{T}} \hat{\mathrm{F}}_{\boldsymbol{\theta}_0} \boldsymbol{\tau}_{t'}$$

$$= \frac{1}{2} \sum_{t=1,t'\neq t}^{T} w_t w_{t'} \, \mathrm{EWC}_{\boldsymbol{\theta}_0}(\boldsymbol{\theta}_t) - \sum_{t=1,t'<t}^{T} w_t w_{t'} \boldsymbol{\tau}_t^{\mathrm{T}} \hat{\mathrm{F}}_{\boldsymbol{\theta}_0} \boldsymbol{\tau}_{t'}$$

$$= \frac{1}{2} \sum_{t=1}^{T} w_t \, \mathrm{EWC}_{\boldsymbol{\theta}_0}(\boldsymbol{\theta}_t) \underbrace{\sum_{t'\neq t}^{T} w_{t'}}_{1-w_t} - \sum_{t=1,t'<t}^{T} w_t w_{t'} \boldsymbol{\tau}_t^{\mathrm{T}} \hat{\mathrm{F}}_{\boldsymbol{\theta}_0} \boldsymbol{\tau}_{t'}$$

$$= \frac{1}{2} \sum_{t=1}^{T} w_t (1 - w_t) \, \mathrm{EWC}_{\boldsymbol{\theta}_0}(\boldsymbol{\theta}_t) - \sum_{t=1,t'<t}^{T} w_t w_{t'} \boldsymbol{\tau}_t^{\mathrm{T}} \hat{\mathrm{F}}_{\boldsymbol{\theta}_0} \boldsymbol{\tau}_{t'},$$

which aligns with the result of Eq. 17. $\qquad\square$

## C CLOSED FORM GRADIENTS FOR EQ. 19

We start by deriving the gradients of $\Omega_{\hat{\mathrm{F}}}(\cdot)$ (Eq. 14) w.r.t. the generic task vector $\boldsymbol{\tau}_k$. We initially consider the case of full-fine tuning, and then generalize the results to LoRA and (IA)$^3$.

To simplify the calculations, we first rearrange $\Omega_{\hat{\mathrm{F}}}(\cdot)$ as:

$$\Omega_{\hat{\mathrm{F}}}(\boldsymbol{\theta}_1, \ldots, \boldsymbol{\theta}_T) = \frac{1}{2} \sum_{t=1}^{T} \sum_{t'<t} w_t w_{t'} (\boldsymbol{\tau}_t - \boldsymbol{\tau}_{t'})^{\mathrm{T}} \hat{\mathrm{F}}_{\boldsymbol{\theta}_0} (\boldsymbol{\tau}_t - \boldsymbol{\tau}_{t'})$$

$$\approx \frac{1}{2} \sum_{t=1}^{T} w_t (1 - w_t) \, \mathrm{EWC}_{\boldsymbol{\theta}_0}(\boldsymbol{\theta}_t) - \sum_{t=1,t'<t}^{T} w_t w_{t'} \boldsymbol{\tau}_t^{\mathrm{T}} \hat{\mathrm{F}}_{\boldsymbol{\theta}_0} \boldsymbol{\tau}_{t'} \quad \text{(see Eq. 17)},$$

Therefore, we can write:

$$\frac{\partial \Omega_{\hat{\mathrm{F}}}}{\partial \boldsymbol{\tau}_k} = \frac{\partial \left[ \frac{1}{2} \sum_{t=1}^{T} w_t (1 - w_t) \, \mathrm{EWC}_{\boldsymbol{\theta}_0}(\boldsymbol{\theta}_t) \right]}{\partial \boldsymbol{\tau}_k} - \frac{\partial \left[ \sum_{t=1,t'<t}^{T} w_t w_{t'} \boldsymbol{\tau}_t^{\mathrm{T}} \hat{\mathrm{F}}_{\boldsymbol{\theta}_0} \boldsymbol{\tau}_{t'} \right]}{\partial \boldsymbol{\tau}_k}$$

$$= \frac{\partial \left[ \frac{1}{2} w_k (1 - w_k) \, \mathrm{EWC}_{\boldsymbol{\theta}_0}(\boldsymbol{\theta}_k) \right]}{\partial \boldsymbol{\tau}_k} - \qquad\qquad (20)$$

$$- \frac{\partial \left[ \sum_{t'<k} w_k w_{t'} \boldsymbol{\tau}_k^{\mathrm{T}} \hat{\mathrm{F}}_{\boldsymbol{\theta}_0} \boldsymbol{\tau}_{t'} + \sum_{t\neq k,t'<t} w_t w_{t'} \boldsymbol{\tau}_t^{\mathrm{T}} \hat{\mathrm{F}}_{\boldsymbol{\theta}_0} \boldsymbol{\tau}_{t'} \right]}{\partial \boldsymbol{\tau}_k}$$

$$= w_k (1 - w_k) \frac{\partial \left[ \frac{1}{2} \mathrm{EWC}_{\boldsymbol{\theta}_0}(\boldsymbol{\theta}_k) \right]}{\partial \boldsymbol{\tau}_k} - \sum_{t'\neq k}^{T} w_k w_{t'} \frac{\partial \boldsymbol{\tau}_k^{\mathrm{T}} \hat{\mathrm{F}}_{\boldsymbol{\theta}_0} \boldsymbol{\tau}_{t'}}{\partial \boldsymbol{\tau}_k}$$

$$= w_k (1 - w_k) \, \hat{\mathrm{F}}_{\boldsymbol{\theta}_0} \odot \boldsymbol{\tau}_k - w_k \sum_{t'\neq k}^{T} w_{t'} \hat{\mathrm{F}}_{\boldsymbol{\theta}_0} \odot \boldsymbol{\tau}_t'$$

$$= w_k \left[ (1 - w_k) \, \hat{\mathrm{F}}_{\boldsymbol{\theta}_0} \odot \boldsymbol{\tau}_k - \sum_{t'\neq k}^{T} w_{t'} \hat{\mathrm{F}}_{\boldsymbol{\theta}_0} \odot \boldsymbol{\tau}_t' \right]$$

$$= w_k \left[ \hat{\mathrm{F}}_{\boldsymbol{\theta}_0} \odot \left[ (1 - w_k) \boldsymbol{\tau}_k - \sum_{t'\neq k}^{T} w_{t'} \boldsymbol{\tau}_t' \right] \right]. \qquad\qquad (21)$$

We now suppose that training is performed incrementally on a sequence of $1, 2, \ldots, k, \ldots, T$ tasks. During the $k$-th task, IEL optimizes only $\boldsymbol{\tau}_k$, namely the task vector instantiated at the beginning of the $k$-th task. Indeed, as discussed in Sec. 2.3 (Eq. 19), the task vectors $\boldsymbol{\tau}_1, \boldsymbol{\tau}_2, \ldots, \boldsymbol{\tau}_{k-1}$ introduced in preceding tasks are kept frozen. Moreover, at each round, we devise uniform weights, such that $w_t = \frac{1}{k}$, with $t < k$. On top of that, we can rewrite the gradients in Eq. 20 as:

$$\frac{\partial \Omega_{\hat{F}}}{\partial \boldsymbol{\tau}_k} = \frac{1}{k} \left[ \hat{F}_{\boldsymbol{\theta}_0} \odot \left[ (1 - \frac{1}{k}) \boldsymbol{\tau}_k - \frac{1}{k} \sum_{t<k} \boldsymbol{\tau}_t \right] \right] \tag{22}$$

Notably, the term $\frac{1}{k} \sum_{t<k} \boldsymbol{\tau}_t$ is a running average of the preceding task vectors. As also discussed in App. D, this reduces the memory/time computational cost for computing the gradients of $\Omega_{\hat{F}}$ to $\mathcal{O}(1)$.

As a final step, we can exploit the result in Eq. 22 to compute the gradients of LoRA ($\boldsymbol{\tau}_k = B_k A_k$) and $(\mathtt{IA})^3$. For LoRA, we have that:

$$\frac{\partial \Omega}{\partial B_k} = \frac{\partial \Omega}{\partial \boldsymbol{\tau}_k} \frac{\partial \boldsymbol{\tau}_k}{\partial B_k} = \frac{1}{k} \left[ \hat{F}_{\boldsymbol{\theta}_0} \odot \left[ (1 - \frac{1}{k}) \boldsymbol{\tau}_k - \frac{1}{k} \sum_{t<k} \boldsymbol{\tau}_t \right] \right] A^{\mathrm{T}}$$

$$\frac{\partial \Omega}{\partial A_k} = \frac{\partial \Omega}{\partial \boldsymbol{\tau}_k} \frac{\partial \boldsymbol{\tau}_k}{\partial A_k} = \frac{1}{k} B^{\mathrm{T}} \left[ \hat{F}_{\boldsymbol{\theta}_0} \odot \left[ (1 - \frac{1}{k}) \boldsymbol{\tau}_k - \frac{1}{k} \sum_{t<k} \boldsymbol{\tau}_t \right] \right]$$

In the case of $(\mathtt{IA})^3$, where $\boldsymbol{\tau}_k = \boldsymbol{\theta}_0 \odot ((l_k - \mathbf{1}_d) \otimes \mathbf{1}_d)$:

$$\frac{\partial \Omega}{\partial l_k} = \frac{\partial \Omega}{\partial \boldsymbol{\tau}_k} \frac{\partial \boldsymbol{\tau}_k}{\partial l_k} = \left[ \frac{1}{k} \left[ \hat{F}_{\boldsymbol{\theta}_0} \odot \left[ (1 - \frac{1}{k}) \boldsymbol{\tau}_k - \frac{1}{k} \sum_{t<k} \boldsymbol{\tau}_t \right] \right] \odot \boldsymbol{\theta}_0^{\mathrm{T}} \right] \mathbf{1}_d$$

where $\mathbf{1}_d$ is a $d$-dimensional column vector of ones with shape $d \times 1$.

## D  COMPUTATIONAL ANALYSIS

**Analysis of ITA.** Regarding ITA (*i.e.*, individual training), the learning phase has constant $\mathcal{O}(1)$ time and memory complexity. As each model is optimized in isolation, the training cost is similar to that of EWC (Kirkpatrick et al., 2017) and stems from the computation and storage of the Fisher Information Matrix, along with the calculations of the penalty term. During the evaluation phase, the time complexity of ITA remains $\mathcal{O}(1)$, as it involves a single forward pass on the composed model $f(\cdot; \boldsymbol{\theta}_{\mathcal{P}})$. Memory complexity, on the other hand, is $\mathcal{O}(T)$ if maintaining separate models with distinct expertise is desired (*e.g.*, for later re-composition). Otherwise, this can be avoided by considering the composed model as a cumulative average of individual models (see later), resulting in a memory cost of $\mathcal{O}(1)$.

**Analysis of IEL.** Referring to IEL (*i.e.*, ensemble training), it might initially appear expensive due to the joint training of the ensemble. However, the complexity of IEL remains constant to $\mathcal{O}(1)$ in terms of both memory and time. This reduction is intuitive when we view the ensemble as a cumulative average of individual weights. To demonstrate this, we begin by considering the following cascade of models to be learned:

$$\mathrm{Task}_{\#1} \rightarrow f(\cdot; \boldsymbol{\theta}_{\mathcal{P}} = \boldsymbol{\theta}_0 + \boldsymbol{\tau}_1)$$

$$\mathrm{Task}_{\#2} \rightarrow f(\cdot; \boldsymbol{\theta}_{\mathcal{P}} = \boldsymbol{\theta}_0 + \frac{1}{2} \boldsymbol{\tau}_1 + \frac{1}{2} \boldsymbol{\tau}_2)$$

$$\cdots$$

$$\mathrm{Task}_{\#t} \rightarrow f(\cdot; \boldsymbol{\theta}_{\mathcal{P}} = \boldsymbol{\theta}_0 + \frac{1}{t} \boldsymbol{\tau}_1 + \frac{1}{t} \boldsymbol{\tau}_2 + \cdots + \frac{1}{t} \boldsymbol{\tau}_{t-1} + \frac{1}{t} \boldsymbol{\tau}_t)$$

At each task, only the latter component of the composed model is learnable, while the preceding components are frozen:

$$\mathrm{Task}_{\#t} \rightarrow f(\cdot; \boldsymbol{\theta}_{\mathcal{P}} = \boldsymbol{\theta}_0 + \underbrace{\frac{1}{t} \boldsymbol{\tau}_1 + \frac{1}{t} \boldsymbol{\tau}_2 + \ldots \frac{1}{t} \boldsymbol{\tau}_{t-1}}_{\text{frozen components}} + \frac{1}{t} \underbrace{\boldsymbol{\tau}_t}_{\text{learnable}}).$$

Therefore, as the preceding $\#t-1$ components are kept frozen, we can incorporate them into the initialization weights $\boldsymbol{\theta}_0 \rightarrow \boldsymbol{\theta}_0^{(t)}$ and optimize:

$$\text{Task}_{\#t} \rightarrow f(\cdot; \boldsymbol{\theta}_\mathcal{P} = \boldsymbol{\theta}_0^{(t)} + \frac{1}{t}\boldsymbol{\tau}_t) \quad \text{where} \quad \boldsymbol{\theta}_0^{(t)} = \boldsymbol{\theta}_0 + \frac{1}{t}\sum_{t'=1}^{t-1}\boldsymbol{\tau}_{t'}.$$

Under this perspective, the learning of the $t$-th task is comparable to standard fine-tuning with a re-scaling factor $= 1/t$. Therefore, provided that $\boldsymbol{\theta}_0^{(t)}$ is computed once at the beginning of the $t$-th task, the learning of the $t$-th task has constant $\mathcal{O}(1)$ time complexity.

It is noted that optimizing Eq. 19 via gradient descent can be similarly simplified, resulting in a constant $\mathcal{O}(1)$ time complexity. In fact, the gradients derived in App. C involve averaging the weights learned during previous tasks, specifically $\frac{1}{t}\sum_{t'=1}^{t-1}\boldsymbol{\tau}_{t'}$. Since the weights from previous tasks are frozen during the current task, we can compute this average once and cache it.

To reduce memory complexity to $\mathcal{O}(1)$, we must avoid storing $\boldsymbol{\tau}_1, \ldots, \boldsymbol{\tau}_{t-1}$ separately. This can be accomplished by assuming an initial null displacement $\boldsymbol{\tau}_0 = \mathbf{0}$ and redefining $\boldsymbol{\theta}_0^{(t)}$ as:

$$\boldsymbol{\theta}_0^{(t)} = \boldsymbol{\theta}_0 + \frac{1}{t}\sum_{t'=1}^{t-1}\boldsymbol{\tau}_{t'} = \boldsymbol{\theta}_0 + \boldsymbol{\tau}_0 + \frac{1}{t}\sum_{t'=1}^{t-1}\boldsymbol{\tau}_{t'} = \boldsymbol{\theta}_0 + \frac{1}{t}\sum_{t'=0}^{t-1}\boldsymbol{\tau}_{t'} =$$

$$= \boldsymbol{\theta}_0 + \boldsymbol{\tau}_{\text{AVG}}^{(t)}$$

The term $\boldsymbol{\tau}_{\text{AVG}}^{(t)} = \frac{1}{t}\sum_{t'=0}^{t-1}\boldsymbol{\tau}_{t'}$ is basically the cumulative average of the displacements up to the current task (***excluded***). The cumulative average is straightforward to compute and, as is well-known, eliminates the need to store all previous values appearing in the sum, with resulting memory complexity $\mathcal{O}(1)$.

# E  IMPLEMENTATION DETAILS OF ITA AND IEL

**Task pre-consolidation – Linear Probing.**  At the beginning of each task, during the pre-consolidation phase, we train a new classification head to account for the classes introduced by the current task. While the rest of the backbone remains frozen, a new linear classification layer is then trained with standard Stochastic Gradient Descent (SGD) for a varying number of epochs, depending on the dataset (typically either 3 or 8 epochs, as detailed in App. I).

With standard linear probing, the newly trained classification head can be considered reliable only for the classes of the current task. However, it may be miscalibrated for classes from previous tasks. This misalignment could undermine the role of pre-training in regularization. To mitigate this and enforce the hypothesis of pre-training optimality for the global empirical risk **across all tasks**, we adopt a classifier alignment approach inspired by Zhang et al. (2023). For each new class, we fit a class-specific Mixture of multivariate Gaussians (MoG) model on the feature representations obtained from the frozen pre-trained model. To capture intra-class variations, we use $K = 5$ Gaussian components per MoG. In subsequent tasks, we generate $N = 256$ synthetic feature vectors using the respective MoGs. These generated feature vectors, encompassing both past and present classes, are then used to fine-tune the new classification head. This approach allows us to enforce the hypothesis of pre-training optimality without relying on input data from other tasks.

**Task pre-consolidation – Update of the FIM.** Afterward, the diagonal Fisher Information Matrix (FIM) has to be updated to incorporate new information from the current task. Similar to Schwarz et al. (2018), we consider the estimated FIM as an online cumulative average of the squared gradients of the negative log-likelihood. Unlike Schwarz et al. (2018), we do not introduce the hyperparameter $\gamma$ to down-weight the importance of the previous estimate. Moreover, following Kunstner et al. (2019), we compute the **true** FIM as defined in Eq. 10: *i.e.*, taking the expected gradient on the prediction vector $\hat{y}$. This approach differs from the majority of existing methods (Schwarz et al., 2018; Kirkpatrick et al., 2017; Chaudhry et al., 2018), which apply a further approximation by relying on the empirical FIM: *i.e.*, only considering the gradient of the ground truth label. Consequently, our estimate is more accurate but requires multiple backward passes, but the computational impact of this operation can be significantly reduced through batch-wise parallelization as in George (2021).

**Fine-tuning – Initialization.** All methods utilize the same pre-training (supervised) on ImageNet21K (`vit_base_patch16_224.augreg_in21k` from the `timm` library). Following the original works, we initialize the learnable parameters such that task vectors start from the pre-train initialization:

- Full Fine-Tuning: we apply zero-initialization.
- LoRA: we use Gaussian initialization for matrix $A$ and zero initialization for matrix $B$.
- $(IA)^3$: we initialize the vectors $l$ with ones.

**Fine-tuning – Loss function.** Importantly, while our derivations regard the second-order approximation $\ell_{cur}$, the full loss $\ell$ is instead employed in our algorithms. Indeed, as is common in frameworks similar to ours (Chaudhry et al., 2018; Mirzadeh et al., 2020), the Taylor approximation is employed to build a surrogate of the exact loss that is both accurate and mathematically tractable. However, when it comes to practice, this proxy is often relaxed, and the full target function is used instead for simplicity.

Following existing works (Smith et al., 2023; Wang et al., 2022b), we employ the **local cross-entropy loss** as the learning objective. Given an example from the current task, while the standard cross-entropy loss considers logits related to all classes, including those learned in previous tasks, the local cross-entropy focuses only on logits corresponding to the classes introduced in the current task. This approach prevents the logits of past classes from being overly penalized during the current task (Caccia et al., 2022; Boschini et al., 2022). To ensure a fair comparison in our experiments, we apply this modification to other competing methods (*e.g.*, EWC (Kirkpatrick et al., 2017)).

**Fine-tuning – Optimization.** In each experiment, we use the AdamW optimizer (Loshchilov & Hutter, 2019) with a learning rate of $3 \times 10^{-4}$ for LoRA and $(IA)^3$ fine-tuning, and $1 \times 10^{-4}$ for full fine-tuning. Importantly, in both ITA and IEL, we employ a decoupled strategy (Loshchilov & Hutter, 2019) to incorporate the gradients of the regularization term. Specifically, we apply the gradients of the regularizing objectives directly to the parameters before the gradient update step, ensuring that the regularization term does not interfere with momentum in the optimization dynamics. By adopting this approach, we observe an empirical improvement in final accuracy, attributed to a more effective minimization of the Riemannian distance relative to the pre-training initialization (see Eq. 12). Finally, we apply this decoupled gradient update exclusively to LoRA and $(IA)^3$. For full fine-tuning, we refrain from using it as we observed numerical instabilities (*i.e.*, exploding loss).

Furthermore, we decouple the regularization strength applied to the final classification layer from that applied to the rest of the learnable parameters. This introduces two additional hyper-parameters: $\alpha_{CLS}$ for ITA and $\beta_{CLS}$ for IEL. Intuitively, by decoupling the regularization weights, we can increase the regularization strength of intermediate layers without causing numerical instabilities, which often stem from the final classification layer.

## F  EXTENDED DISCUSSION ON RELATED WORKS AND COMPETING METHODS

To deliver the most comprehensive and significant comparison with the state of the art in incremental learning, we chose a combination of well-established standard approaches (such as EWC, DER++, L2P, and CODA) and recent proposals emphasizing compositionality skills (*e.g.*, TMC and APT) and ensemble learning (*i.e.*, SEED). It is important to note that these methods have been heavily influenced by the research trends prevalent at the time they were originally proposed, and therefore, they tend to employ different strategies for fine-tuning the model. To sum up:

- EWC, LwF, DER++, SEED and TMC leverage on full-fine tuning.
- L2P, CODA, APT resort instead to prompting.

Therefore, achieving a direct apple-to-apple comparison is challenging, as the approaches present in the literature are themselves prone to this issue.

We herein summarize the main aspects of the more important recent methods we compared with.

**Elastic Weight Consolidation (EWC)** As discussed in the main paper, while our approach regularizes the distance in parameter space with respect to $\theta_0$, the regularization in EWC (Kirkpatrick

et al., 2017) instead focuses on the weights learned during the preceding task. However, beyond the different regularizing strategies, there is another significant difference between ITA and EWC. While ITA fine-tunes each task starting from the same original pre-trained model $\theta_0$, EWC begins with the weights of the previous task.

It is noted that an EWC-like term that protects the last task weights could work as well in our framework based on task vectors. We chose to anchor the model to the pre-training weights to allow for more flexible decentralized learning, wherein multiple task vectors can be trained on different tasks *in parallel*, with minimal interactions. In contrast, an EWC-like term, which regularizes each successor based on its predecessor, would necessitate training the multiple task vectors *in sequence*.

**Learning to Prompt (L2P)** (Wang et al., 2022b) and **CODA-Prompt** (Smith et al., 2023) are two continual learning techniques based on prompting. They fine-tune a pre-trained model through a few learnable parameters stored in a prompt pool, which can be either shared or tied to different tasks. At inference time, the prompts are retrieved from the pool through a query-key search. In terms of memory complexity, our ITA is in line with L2P and CODA when coupled with a parameter-efficient fine-tuning technique such as IA3. Moreover, ITA does not require the additional forward pass on the frozen pre-trained model required to retrieve the prompts.

**À-la-carte Prompt Tuning (APT)** (Bowman et al., 2023) is a prompt-based strategy similar to L2P and CODA. The prompts are trained in isolation (similarly to ITA) and concatenated at inference time to create the composed predictor (no prior query-key search is required). To avoid destructive interference, a tailored masking mechanism is employed in self-attention layers. Differently, our approaches fuse parameters through linear combinations.

**Tangent Model Composition (TMC)** (Liu & Soatto, 2023) is a recent approach addressing continual learning through task arithmetic in the tangent space. TMC builds upon task vectors and is similar to ITA (full fine-tuning). They differ in two aspects: *i)* TMC applies a first-order approximation of the forward pass to support compositionality, making it two to three times slower than a non-linear forward pass; *ii)* TMC does not include auxiliary regularization during training.

**Selection of Experts for Ensemble Diversification (SEED)** (Rypeść et al., 2024) is a recent approach that trains an ensemble of models incrementally. For each incoming task, an expert model is chosen from the pool and trained. The major difference with respect to our IEL concerns the inference stage: while IEL performs an ensemble prediction in $\mathcal{O}(1)$ time (thanks to weight averaging), SEED makes inference on all models at test time and averages their predictions.

**Model merging.** While we address compositionality during training, other approaches focus on post-training techniques, as simple averaging leads to interference (Yadav et al., 2024b) when parameters are redundant or have conflicting signs. TIES (Yadav et al., 2024b) discards uninformative parameters and addresses conflicts via majority voting. Zipit! (Stoica et al., 2024) merges redundant parameters that produce similar features, while RegMean (Jin et al., 2023) exploits a closed-form solution for linear layers. Notably, (Matena & Raffel, 2022) weighs the contribution of each parameter through the Fisher matrix, computed by each individual model at its optimum. This differs from our approach that evaluates the FIM at the pre-training optimum.

# G  DATASETS

We conduct a comprehensive evaluation using a variety of benchmarks. Following the current literature on pre-trained CL models (Wang et al., 2022b;a; Smith et al., 2023), we include conventional image datasets such as Split CIFAR-100 and Split ImageNet-R. We also include Split CUB-200, Split Caltech-256 and Split MIT-67, recently used in the context of composable incremental methods (Bowman et al., 2023; Liu & Soatto, 2023). Finally, we assess the adaptability of these pre-trained methods in settings with decreasing domain similarity to the ImageNet pre-training utilized by our backbone model (Oh et al., 2022; Cui et al., 2018). Specifically, these settings include the satellite and medical domains, represented by Split RESISC45 and Split CropDiseases, respectively. In the following, we outline the due details:

- **Standard domains**: *Split CIFAR-100* (Krizhevsky et al., 2009) and *Split ImageNet-R* (Hendrycks et al., 2021), with respectively 100 and 200 classes split into 10 tasks. We

Table 4: Comparison with the SOTA across several benchmarks (Final Forgetting [↓]).

| Model | IN-R | C-100 | CUB | Caltech | MIT | RESISC | CropDis. |
|---|---|---|---|---|---|---|---|
| Joint | ✗ | ✗ | ✗ | ✗ | ✗ | ✗ | ✗ |
| Finetune | 75.5 | 85.7 | 65.79 | 58.76 | 87.89 | 97.6 | 92.04 |
| EWC (ON) | 32.43 | 25.37 | 53.04 | 23.52 | 34.65 | 43.12 | 31.58 |
| LWF-MC | 10.39 | 6.13 | 26.2 | 5.23 | 3.56 | 5.66 | 7.94 |
| DER++ | 35.81 | 15.66 | 13.58 | 11.19 | 31.51 | 43.04 | 0.76 |
| L2P | 5.6 | 6.08 | 6.63 | 1.78 | 5.55 | 26.29 | 17.81 |
| CODA | **4.97** | 5.28 | 8.82 | 2.64 | 7.49 | 22 | 15.37 |
| SEED | 8.96 | 6.39 | 5.12 | 2.67 | 4.53 | 8.66 | 1.19 |
| InfLoRA | 5.67 | 4.05 | 4.58 | 1.89 | 6.85 | 10.77 | 4.79 |
| APT | 8.99 | 6.71 | 8.82 | 3.64 | 6.33 | 27.74 | 14.01 |
| TMC | 11.66 | 9.2 | 7.37 | 6.39 | 12.43 | 16.43 | 16.94 |
| ITA-FFT | 7.72 | 3.89 | **1.39** | 2.44 | 4.80 | 6.19 | 0.95 |
| ITA-LoRA | 8.00 | 3.22 | 2.08 | 2.38 | 4.68 | 8.30 | 0.68 |
| ITA-$(IA)^3$ | 8.72 | 3.57 | 2.23 | 2.39 | 4.48 | 7.33 | 0.76 |
| IEL-FFT | 5.40 | **2.20** | 2.04 | **1.50** | **2.29** | **3.46** | 1.40 |
| IEL-LoRA | 7.63 | 5.14 | **3.46** | 2.46 | 8.34 | 8.30 | 2.14 |
| IEL-$(IA)^3$ | 8.34 | 3.27 | 1.69 | 2.20 | 5.82 | 6.86 | 1.04 |

train each task of Split ImageNet-R for 30 epochs and each task of Split CIFAR-100 for 20 epochs. In particular, IN-R is a variant of the ImageNet dataset that includes artistic renditions such as sketches, cartoons, and paintings. It is used to evaluate the robustness and generalization capabilities of models trained on the original ImageNet when tested on out-of-distribution data. Following (Liu & Soatto, 2023), we also employ Split Caltech-256 (Griffin et al., 2007) and Split MIT-67 (Quattoni & Torralba, 2009), dividing both into 10 tasks (5 epoch each).

- **Specialized domain**: We adopt *Split CUB-200* (Wah et al., 2011) to evaluate compositional capabilities in a more fine-grained classification scenario, namely recognizing 200 species of birds. The classes are split across 10 tasks, each lasting for 50 epochs.
- **Aerial domain**: we use *Split RESISC45* (Cheng et al., 2017), which comprises 30000 RGB satellite images for land use and land cover classification. The dataset contains 45 classes (*e.g.*, airport, cloud, island, and so on) divided into 9 tasks, with each task lasting 30 epochs.
- **Medical domain**: we finally explore the medical setting (*i.e.*, plant diseases) and conduct experiments on *Split CropDiseases* (Hughes et al., 2015). It regards infected leaves with 7 tasks of 5 classes each (5 epochs).

We base our code on Mammoth (Buzzega et al., 2020b;a), a widely adopted framework in the class-incremental learning literature.

# H ADDITIONAL RESULTS

## H.1 FORGETTING

Tab. 4 reports the Final Forgetting metric (Chaudhry et al., 2018) for our experiments.

## H.2 STANDARD DEVIATION OF FA

Tab. 5 reports the standard deviation of the Final Accuracy metrics reported in Tab. 1.

## H.3 ABLATION STUDY ON IEL

We herein present an ablative analysis regarding the ensemble-oriented regularization applied to the proposed IEL approach. Specifically, we evaluate the impact of Eq. 19 on the final accuracy and

Table 5: For each experiment of Tab. 1, the standard deviation of the Final Accuracy (FA).

| Model | IN-R | C-100 | CUB | Caltech | MIT | RESISC | CropDis. |
|---|---|---|---|---|---|---|---|
| Joint | 0.87 | 0.16 | 0.41 | 0.43 | 0.31 | 0.22 | 0.21 |
| Finetune | 4.53 | 2.83 | 1.65 | 2.57 | 3.49 | 1.12 | 0.43 |
| EWC | 0.73 | 1.20 | 0.87 | 1.29 | 1.45 | 0.61 | 0.99 |
| LWF-MC | 0.89 | 1.12 | 3.23 | 3.94 | 2.98 | 2.85 | 2.85 |
| DER++ | 1.12 | 1.14 | 0.82 | 1.27 | 0.95 | 1.79 | 1.29 |
| L2P | 0.56 | 1.23 | 2.12 | 0.63 | 0.81 | 3.42 | 0.45 |
| CODA | 0.63 | 0.95 | 2.87 | 0.82 | 0.69 | 4.61 | 0.98 |
| SEED | 0.18 | 0.47 | 0.65 | 0.55 | 0.41 | 0.86 | 0.23 |
| APT | 0.84 | 1.30 | 2.45 | 0.95 | 0.73 | 1.94 | 0.68 |
| InfLoRA | 0.10 | 0.21 | 0.89 | 0.68 | 2.03 | 1.16 | 0.42 |
| TMC | 0.45 | 0.85 | 1.67 | 0.71 | 0.92 | 0.82 | 0.81 |
| ITA-FFT | 0.27 | 0.27 | 0.39 | 0.18 | 0.24 | 0.56 | 2.10 |
| ITA-LoRA | 0.26 | 0.20 | 0.13 | 0.16 | 0.40 | 0.89 | 0.55 |
| ITA-$\texttt{(IA)}^3$ | 0.28 | 0.04 | 0.18 | 0.18 | 0.69 | 0.68 | 0.27 |
| IEL-FFT | 0.11 | 0.17 | 0.21 | 0.09 | 1.12 | 0.77 | 0.90 |
| IEL-LoRA | 0.36 | 0.27 | 0.37 | 0.41 | 0.93 | 0.30 | 0.43 |
| IEL-$\texttt{(IA)}^3$ | 0.23 | 0.20 | 0.32 | 0.21 | 0.55 | 0.48 | 0.41 |

Table 6: Ablation study for ITA-$\texttt{(IA)}^3$ and IEL on several benchmarks (FA [↑]).

| Model | IN-R | C-100 | CUB | Caltech | MIT | RESISC | CropDis. |
|---|---|---|---|---|---|---|---|
| **ITA-$\texttt{(IA)}^3$** *(reg)* | **77.04** | **90.66** | **85.67** | **92.67** | 84.74 | 83.73 | 95.41 |
| *without Eq. 12 reg.* | 71.82 | 88.43 | 77.61 | 90.66 | 69.14 | 69.01 | 63.72 |
| *Eq. 12 only on* CLS | 76.72 | 90.48 | 85.56 | 92.56 | **85.25** | **84.37** | **95.45** |
| **IEL-FFT** *(reg)* | **80.09** | **89.38** | 84.89 | **92.23** | **82.79** | **81.42** | 95.83 |
| *without Eq. 19 reg.* | 40.85 | 52.56 | 14.02 | 53.76 | 47.63 | 39.20 | 31.24 |
| *Eq. 19 reg. only on* CLS | 77.99 | 85.82 | **85.30** | 91.43 | 77.58 | 76.87 | **96.18** |
| **IEL-LoRA** *(reg)* | **79.93** | **89.53** | **84.95** | **92.19** | **84.49** | **82.53** | **95.88** |
| *without Eq. 19 reg.* | 51.15 | 66.01 | 60.39 | 70.71 | 55.38 | 42.72 | 45.25 |
| *Eq. 19 reg. only on* CLS | 76.14 | 86.11 | 84.43 | 91.77 | 82.50 | 70.05 | 95.54 |
| **IEL-$\texttt{(IA)}^3$** *(reg)* | **77.86** | **89.72** | 84.57 | 92.70 | **85.54** | 81.50 | 95.68 |
| *without Eq. 19 reg.* | 73.72 | 84.00 | 74.72 | 89.58 | 69.82 | 62.52 | 66.29 |
| *Eq. 19 reg. only on* CLS | 77.23 | 89.38 | **84.70** | **92.76** | 85.43 | **81.60** | **95.72** |

report the results in Tab. 6. After examining them, we can draw conclusions similar to those made for ITA. In particular, the regularization driven by the second-order formulation proves beneficial for achieving effective composition (especially the full-fine tuning), with the final classification layer playing an important role in attaining good performance.

## H.4    Additional experiments on incremental model compositionality

Tab. 7 reports the results of ITA, IEL and TMC on additional datasets, with a focus on their compositionality capabilities (*i.e.*, specialization and unlearning).

## H.5    Alignment between task vectors of ITA and IEL

To better understand the relationship between the task vectors learned by ITA and IEL, in Fig. 2 we conduct an additional evaluation by measuring the alignment – in terms of cosine similarity – between the parameters learned by the two proposed approaches. The analysis is twofold: first, the alignment is evaluated between individual task vectors $\tau_t^{\text{ITA}}$ and $\tau_t^{\text{IEL}}$ by averaging the similarity

Table 7: Analysis of compositional capabilities. In parentheses, we report the gain (or loss) in accuracy on the target task.

| Dataset | Model | zero-shot specialization | | zero-shot unlearning | |
|---|---|---|---|---|---|
| | | $\text{FA}_{\text{TGT}}$ [↑] | $\text{FA}_{\text{CTRL}}$ | $\text{FA}_{\text{TGT}}$ [↓] | $\text{FA}_{\text{CTRL}}$ |
| **IN-R** | ITA-LoRA | 80.83 $_{(+11.40)}$ | 50.52 | 22.77 $_{(-55.02)}$ | 52.72 $_{(-25.07)}$ |
| | IEL-LoRA | 73.46 $_{(-06.68)}$ | 38.46 | 18.55 $_{(-61.38)}$ | 41.97 $_{(-37.96)}$ |
| | TMC | 69.93 $_{(+08.36)}$ | 34.08 | 45.77 $_{(-14.24)}$ | 54.37 $_{(-05.64)}$ |
| **C-100** | ITA-LoRA | 92.80 $_{(+01.63)}$ | 60.06 | 28.67 $_{(-61.29)}$ | 71.96 $_{(-17.99)}$ |
| | IEL-LoRA | 77.77 $_{(-13.22)}$ | 37.90 | 19.48 $_{(-70.05)}$ | 56.52 $_{(-33.01)}$ |
| | TMC | 87.53 $_{(+06.49)}$ | 45.75 | 55.63 $_{(-22.79)}$ | 71.83 $_{(-06.59)}$ |
| **CUB** | ITA-LoRA | 90.46 $_{(+05.21)}$ | 57.87 | 68.19 $_{(-17.36)}$ | 74.63 $_{(-10.92)}$ |
| | IEL-LoRA | 74.03 $_{(-10.57)}$ | 47.95 | 22.44 $_{(-62.51)}$ | 30.99 $_{(-53.96)}$ |
| | TMC | 71.06 $_{(+09.92)}$ | 44.93 | 67.91 $_{(-03.81)}$ | 53.22 $_{(-18.50)}$ |
| **Caltech** | ITA-LoRA | 89.84 $_{(-01.21)}$ | 65.47 | 80.02 $_{(-12.63)}$ | 82.40 $_{(-10.25)}$ |
| | IEL-LoRA | 79.70 $_{(-12.99)}$ | 62.45 | 32.00 $_{(-60.19)}$ | 42.03 $_{(-50.16)}$ |
| | TMC | 89.23 $_{(+10.63)}$ | 49.46 | 64.52 $_{(-17.78)}$ | 73.33 $_{(-08.97)}$ |
| **MIT** | ITA-LoRA | 89.66 $_{(+08.17)}$ | 57.30 | 36.99 $_{(-49.61)}$ | 74.75 $_{(-11.85)}$ |
| | IEL-LoRA | 56.14 $_{(-19.38)}$ | 37.54 | 06.39 $_{(-78.10)}$ | 32.57 $_{(-52.02)}$ |
| | TMC | 88.03 $_{(+10.11)}$ | 30.56 | 29.77 $_{(-38.89)}$ | 62.03 $_{(-06.63)}$ |
| **RESISC** | ITA-LoRA | 89.47 $_{(+06.70)}$ | 49.75 | 33.38 $_{(-48.62)}$ | 64.06 $_{(-17.94)}$ |
| | IEL-LoRA | 90.13 $_{(+04.92)}$ | 48.06 | 32.77 $_{(-49.76)}$ | 65.19 $_{(-17.34)}$ |
| | TMC | 75.77 $_{(+27.63)}$ | 17.00 | 06.64 $_{(-54.02)}$ | 52.31 $_{(-08.35)}$ |
| **CropDis.** | ITA-LoRA | 97.63 $_{(+00.11)}$ | 53.05 | 65.87 $_{(-29.98)}$ | 79.24 $_{(-16.61)}$ |
| | IEL-LoRA | 90.12 $_{(-04.68)}$ | 48.31 | 34.01 $_{(-61.87)}$ | 54.43 $_{(-41.45)}$ |
| | TMC | 73.60 $_{(+09.23)}$ | 15.95 | 06.24 $_{(-60.32)}$ | 53.21 $_{(-13.35)}$ |

across tasks; second, it is assessed between composed models $\tau_{\mathcal{P}}^{\text{ITA}}$ and $\tau_{\mathcal{P}}^{\text{IEL}}$. To eliminate potential confounding effects due to PEFT strategies, we perform the experiment using full fine-tuning.

Our results reveal mostly positive alignment, even for out-of-distribution datasets such as RESISC45 and CropDisease, and low-resolution datasets like CIFAR-100. On the other hand, we observe smaller similarities between the task vectors of the individual models learned on more fine-grained and complex datasets such as CUB-200 and MIT67. Indeed, CUB-200 demands highly detailed distinctions between bird species, while MIT67 involves subtle contextual differences, such as distinguishing children's rooms from living rooms or art studios from classrooms.

Despite the lower alignment for individual task vectors in certain datasets, we observe that the similarity is consistently higher when comparing the composed task vectors to the individual ones.

### H.6 TIMING

We herein include a wall-clock time analysis of the training algorithms. Specifically, considering Split ImageNet-R, Split RESISC45, and Split CropDisease, we measured the per-task runtime of ITA and IEL (both trained with full fine-tuning), as well as that of three existing approaches to incremental learning: DER++ (*rehearsal*), TMC (*model compositionality*), and SEED (*ensemble learning*).

Among the methods compared, SEED appears to be the most efficient. However, the runtimes of both ITA and IEL are comparable to, or even better than, those of baseline methods like DER++ and TMC. Specifically, on ImageNet-R and RESISC45, the average runtime of ITA/IEL is approximately $2/3$ that of DER++. We attribute the reasonable training times of IEL/ITA to the efficient procedure we employed for estimating the FIM, which leverages batch-wise parallelization (see App. E and George (2021)), combined with the use of closed-form gradient computation (refer to App. C).

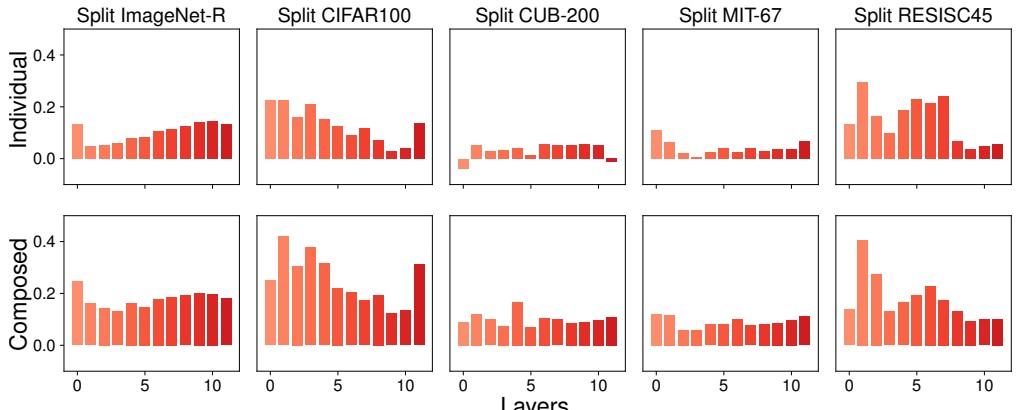

Figure 2: Alignment – *i.e.*, cosine similarity – between the task vectors produced by ITA and IEL for both the *composed* model $\boldsymbol{\theta}_{\mathcal{P}}$ and *individual* learners $\boldsymbol{\theta}_t$ (averaged across tasks $t$).

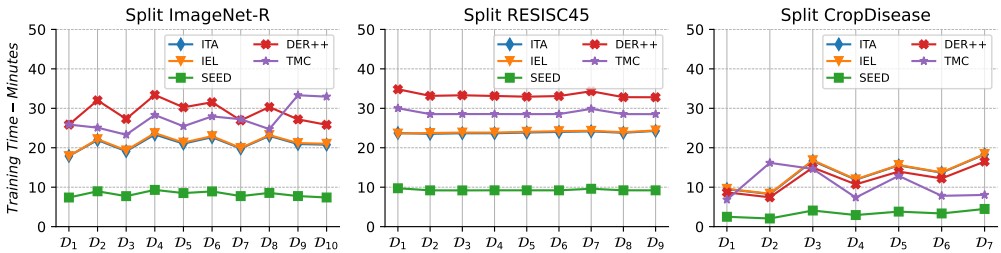

Figure 3: Comparative timing analysis (in minutes). The plot illustrates the per-task runtime of ITA and IEL, alongside baseline methods (DER++, TMC, and SEED). Runtimes include both the setup phase (*e.g.*, steps required to compute FIM statistics) and the training phase.

# I    HYPERPARAMETERS

The hyperparameters employed for each experiment are reported in the following subsections (one for each dataset).

## I.1    IMAGENET-R

**CODA-Prompt**: $\mathrm{lr} = 1.0 \times 10^{-3}$;

**DER++**: $\alpha = 3.0 \times 10^{-1}$; $\beta = 8.0 \times 10^{-1}$; $\mathrm{lr} = 1.0 \times 10^{-3}$;

**EWC**: $\epsilon = 1.0 \times 10^2$; $\gamma = 1.0$; $\mathrm{lr} = 1.0 \times 10^{-2}$;

**L2P**: $\mathrm{lr} = 2.5 \times 10^{-3}$;

**LWF-MC**: $\mathrm{wd} = 0.0$; $\mathrm{lr} = 1.0 \times 10^{-2}$;

**SEED**: $\mathrm{lr} = 3.0 \times 10^{-4}$;

**SGD**: $\mathrm{lr} = 3.0 \times 10^{-2}$;

**TMC**: $\mathrm{lr} = 1.0 \times 10^{-4}$;

**InfLoRA**: $\mathrm{lr} = 5.0 \times 10^{-4}$ $r = 10$; $\epsilon = 0.98$;

For both ITA and IEL: $\#\,\mathrm{epochs}_{\text{pre-tuning}} = 3$; $\mathrm{lr}_{\text{pre-tuning}} = 1.0 \times 10^{-2}$;

**ITA-FFT**: $\alpha = 5.0 \times 10^1$; $\alpha_{\text{CLS}} = 1.0 \times 10^{-1}$; $\alpha_{\text{CLS-prior}} = 1.0 \times 10^{-2}$; $\mathrm{lr} = 1.0 \times 10^{-4}$;

**ITA-LoRA**: $\alpha = 2.0 \times 10^{-2}$; $\alpha_{\text{CLS}} = 1.0 \times 10^{-1}$; $\alpha_{\text{CLS-prior}} = 1.0 \times 10^{-3}$; lr $= 3.0 \times 10^{-4}$;

**ITA-(IA)³**: $\alpha = 7.0 \times 10^{-1}$; $\alpha_{\text{CLS}} = 1.0 \times 10^{-1}$; $\alpha_{\text{CLS-prior}} = 1.0 \times 10^{-2}$; lr $= 3.0 \times 10^{-3}$;

**IEL-FFT**: $\beta = 5.0 \times 10^{1}$; $\beta_{\text{CLS}} = 1.0 \times 10^{-1}$; $\beta_{\text{CLS-prior}} = 3.0 \times 10^{-3}$; lr $= 1.0 \times 10^{-4}$;

**IEL-LoRA**: $\beta = 2.0 \times 10^{1}$; $\beta_{\text{CLS}} = 1.0 \times 10^{-1}$; $\beta_{\text{CLS-prior}} = 1.0 \times 10^{-3}$; lr $= 3.0 \times 10^{-4}$;

**IEL-(IA)³**: $\beta = 2.0 \times 10^{1}$; $\beta_{\text{CLS}} = 1.0 \times 10^{-1}$; $\beta_{\text{CLS-prior}} = 1.0 \times 10^{-2}$; lr $= 3.0 \times 10^{-3}$;

## I.2   CIFAR-100

**CODA-Prompt**: lr $= 1.0 \times 10^{-3}$;

**DER++**: $\alpha = 3.0 \times 10^{-1}$; $\beta = 8.0 \times 10^{-1}$; lr $= 1.0 \times 10^{-4}$;

**EWC**: $\epsilon = 1.0 \times 10^{2}$; $\gamma = 1.0$; lr $= 1.0 \times 10^{-3}$;

**L2P**: lr $= 2.5 \times 10^{-3}$;

**LWF-MC**: wd $= 0.0$; lr $= 1.0 \times 10^{-2}$;

**SEED**: lr $= 3.0 \times 10^{-4}$;

**SGD**: lr $= 1.0 \times 10^{-2}$;

**TMC**: lr $= 1.0 \times 10^{-4}$;

**InfLoRA**: lr $= 5.0 \times 10^{-4}$ $r = 10$; $\epsilon = 0.95$;

For both ITA and IEL: # epochs$_{\text{pre-tuning}} = 3$; lr$_{\text{pre-tuning}} = 1.0 \times 10^{-2}$;

**ITA-FFT**: $\alpha = 2.0 \times 10^{3}$; $\alpha_{\text{CLS}} = 1.0 \times 10^{-1}$; $\alpha_{\text{CLS-prior}} = 3.0 \times 10^{-3}$; lr $= 1.0 \times 10^{-4}$;

**ITA-LoRA**: $\alpha = 2.0 \times 10^{-2}$; $\alpha_{\text{CLS}} = 1.0 \times 10^{-1}$; $\alpha_{\text{CLS-prior}} = 1.0 \times 10^{-3}$; lr $= 3.0 \times 10^{-4}$;

**ITA-(IA)³**: $\alpha = 2.0 \times 10^{-2}$; $\alpha_{\text{CLS}} = 1.0 \times 10^{-1}$; $\alpha_{\text{CLS-prior}} = 3.0 \times 10^{-3}$; lr $= 3.0 \times 10^{-3}$;

**IEL-FFT**: $\beta = 2.0 \times 10^{3}$; $\beta_{\text{CLS}} = 2.5 \times 10^{-2}$; $\beta_{\text{CLS-prior}} = 1.0 \times 10^{-2}$; lr $= 1.0 \times 10^{-4}$;

**IEL-LoRA**: $\beta = 2.0 \times 10^{1}$; $\beta_{\text{CLS}} = 1.0 \times 10^{-1}$; $\beta_{\text{CLS-prior}} = 1.0 \times 10^{-3}$; lr $= 3.0 \times 10^{-4}$;

**IEL-(IA)³**: $\beta = 2.0 \times 10^{1}$; $\beta_{\text{CLS}} = 2.5 \times 10^{-2}$; $\beta_{\text{CLS-prior}} = 1.0 \times 10^{-3}$; lr $= 3.0 \times 10^{-4}$;

## I.3   CUB-200

**CODA-Prompt**: lr $= 1.0 \times 10^{-3}$;

**DER++**: $\alpha = 3.0 \times 10^{-1}$; $\beta = 8.0 \times 10^{-1}$; lr $= 1.0 \times 10^{-3}$;

**EWC**: $\epsilon = 1.0 \times 10^{1}$; $\gamma = 9.0 \times 10^{-1}$; lr $= 1.0 \times 10^{-2}$;

**L2P**: lr $= 2.5 \times 10^{-3}$;

**LWF-MC**: wd $= 1.0 \times 10^{-4}$; lr $= 1.0 \times 10^{-2}$;

**SEED**: lr $= 3.0 \times 10^{-4}$;

**SGD**: lr $= 3.0 \times 10^{-2}$;

**TMC**: lr $= 1.0 \times 10^{-4}$;

**InfLoRA**: lr $= 5.0 \times 10^{-4}$ $r = 10$; $\epsilon = 0.98$;

For both ITA and IEL: lr$_{\text{pre-tuning}} = 1.0 \times 10^{-2}$; # epochs$_{\text{pre-tuning}} = 8$.

**ITA-FFT**: $\alpha = 5.0 \times 10^{2}$; $\alpha_{\text{CLS}} = 1.0 \times 10^{-1}$; $\alpha_{\text{CLS-prior}} = 1.0 \times 10^{-2}$; lr $= 1.0 \times 10^{-4}$;

**ITA-LoRA**: $\alpha = 5.0$; $\alpha_{\text{CLS}} = 1.0 \times 10^{-1}$; $\alpha_{\text{CLS-prior}} = 1.0 \times 10^{-2}$; lr $= 3.0 \times 10^{-4}$;

**ITA-(IA)³**: $\alpha = 7.0 \times 10^{-1}$; $\alpha_{\text{CLS}} = 1.0 \times 10^{-1}$; $\alpha_{\text{CLS-prior}} = 1.0 \times 10^{-2}$; lr $= 3.0 \times 10^{-3}$;

**IEL-FFT**: $\beta = 5.0 \times 10^3$; $\beta_{\text{CLS}} = 1.0 \times 10^{-1}$; $\beta_{\text{CLS-prior}} = 1.0 \times 10^{-2}$; lr $= 1.0 \times 10^{-5}$;

**IEL-LoRA**: $\beta = 2.0 \times 10^1$; $\beta_{\text{CLS}} = 1.0 \times 10^{-1}$; $\beta_{\text{CLS-prior}} = 1.0 \times 10^{-2}$; lr $= 3.0 \times 10^{-4}$;

**IEL-(IA)³**: $\beta = 5.0 \times 10^2$; $\beta_{\text{CLS}} = 1.0 \times 10^{-1}$; $\beta_{\text{CLS-prior}} = 1.0 \times 10^{-2}$; lr $= 3.0 \times 10^{-4}$;

## I.4   CALTECH-256

**CODA-Prompt**: lr $= 1.0 \times 10^{-3}$;

**DER++**: $\alpha = 3.0 \times 10^{-1}$; $\beta = 8.0 \times 10^{-1}$; lr $= 1.0 \times 10^{-3}$;

**EWC**: $\epsilon = 1.0 \times 10^2$; $\gamma = 1.0$; lr $= 1.0 \times 10^{-2}$;

**L2P**: lr $= 2.5 \times 10^{-3}$;

**LWF-MC**: wd $= 0.0$; lr $= 1.0 \times 10^{-2}$;

**SEED**: lr $= 3.0 \times 10^{-4}$;

**SGD**: lr $= 1.0 \times 10^{-2}$;

**TMC**: lr $= 1.0 \times 10^{-4}$;

**InfLoRA**: lr $= 5.0 \times 10^{-4}$ $r = 10$; $\epsilon = 0.99$;

For both ITA and IEL: $\text{lr}_{\text{pre-tuning}} = 1.0 \times 10^{-2}$.

**ITA-FFT**: $\#\text{epochs}_{\text{pre-tuning}} = 3$; $\alpha = 2.0 \times 10^3$; $\alpha_{\text{CLS}} = 1.0 \times 10^{-1}$; $\alpha_{\text{CLS-prior}} = 1.0 \times 10^{-2}$; lr $= 1.0 \times 10^{-4}$;

**ITA-LoRA**: $\#\text{epochs}_{\text{pre-tuning}} = 3$; $\alpha = 2.0 \times 10^{-2}$; $\alpha_{\text{CLS}} = 1.0 \times 10^{-1}$; $\alpha_{\text{CLS-prior}} = 3.0 \times 10^{-3}$; lr $= 3.0 \times 10^{-4}$;

**ITA-(IA)³**: $\#\text{epochs}_{\text{pre-tuning}} = 8$; $\alpha = 7.0 \times 10^{-1}$; $\alpha_{\text{CLS}} = 1.0 \times 10^{-1}$; $\alpha_{\text{CLS-prior}} = 1.0 \times 10^{-2}$; lr $= 3.0 \times 10^{-3}$;

**IEL-FFT**: $\#\text{epochs}_{\text{pre-tuning}} = 3$; $\beta = 5.0 \times 10^1$; $\beta_{\text{CLS}} = 1.0 \times 10^{-1}$; $\beta_{\text{CLS-prior}} = 1.0 \times 10^{-2}$; lr $= 1.0 \times 10^{-4}$;

**IEL-LoRA**: $\#\text{epochs}_{\text{pre-tuning}} = 3$; $\beta = 2.0 \times 10^1$; $\beta_{\text{CLS}} = 5.0$; $\beta_{\text{CLS-prior}} = 3.0 \times 10^{-3}$; lr $= 3.0 \times 10^{-4}$;

**IEL-(IA)³**: $\#\text{epochs}_{\text{pre-tuning}} = 3$; $\beta = 2.0 \times 10^1$; $\beta_{\text{CLS}} = 1.0 \times 10^{-1}$; $\beta_{\text{CLS-prior}} = 3.0 \times 10^{-3}$; lr $= 3.0 \times 10^{-4}$;

## I.5   MIT-67

**CODA-Prompt**: lr $= 1.0 \times 10^{-3}$;

**DER++**: $\alpha = 3.0 \times 10^{-1}$; $\beta = 8.0 \times 10^{-1}$; lr $= 1.0 \times 10^{-3}$;

**EWC**: $\epsilon = 1.0 \times 10^2$; $\gamma = 1.0$; lr $= 1.0 \times 10^{-3}$;

**L2P**: lr $= 2.5 \times 10^{-3}$;

**LWF-MC**: wd $= 0.0$; lr $= 1.0 \times 10^{-2}$;

**SEED**: lr $= 3.0 \times 10^{-4}$;

**SGD**: lr $= 1.0 \times 10^{-2}$;

**TMC**: lr $= 1.0 \times 10^{-4}$;

**InfLoRA**: lr $= 1.0 \times 10^{-3}$ $r = 10$; $\epsilon = 0.95$;

For both ITA and IEL: $\text{lr}_{\text{pre-tuning}} = 1.0 \times 10^{-2}$.

**ITA-FFT**: $\# \text{epochs}_{\text{pre-tuning}} = 8$; $\alpha = 5.0 \times 10^3$; $\alpha_{\text{CLS}} = 1.0 \times 10^{-1}$; $\alpha_{\text{CLS-prior}} = 1.0 \times 10^{-1}$; $\text{lr} = 1.0 \times 10^{-4}$;

**ITA-LoRA**: $\# \text{epochs}_{\text{pre-tuning}} = 8$; $\alpha = 5.0$; $\alpha_{\text{CLS}} = 1.0 \times 10^{-1}$; $\alpha_{\text{CLS-prior}} = 1.0 \times 10^{-2}$; $\text{lr} = 3.0 \times 10^{-4}$;

**ITA-(IA)³**: $\# \text{epochs}_{\text{pre-tuning}} = 8$; $\alpha = 5.0$; $\alpha_{\text{CLS}} = 1.0 \times 10^{-1}$; $\alpha_{\text{CLS-prior}} = 3.0 \times 10^{-3}$; $\text{lr} = 3.0 \times 10^{-4}$;

**IEL-FFT**: $\# \text{epochs}_{\text{pre-tuning}} = 8$; $\beta = 5.0 \times 10^3$; $\beta_{\text{CLS}} = 1.0 \times 10^{-1}$; $\beta_{\text{CLS-prior}} = 1.0 \times 10^{-2}$; $\text{lr} = 1.0 \times 10^{-4}$;

**IEL-LoRA**: $\# \text{epochs}_{\text{pre-tuning}} = 3$; $\beta = 2.0 \times 10^1$; $\beta_{\text{CLS}} = 1.0 \times 10^{-1}$; $\beta_{\text{CLS-prior}} = 1.0 \times 10^{-2}$; $\text{lr} = 3.0 \times 10^{-4}$;

**IEL-(IA)³**: $\# \text{epochs}_{\text{pre-tuning}} = 3$; $\beta = 2.0 \times 10^1$; $\beta_{\text{CLS}} = 1.0 \times 10^{-1}$; $\beta_{\text{CLS-prior}} = 1.0 \times 10^{-2}$; $\text{lr} = 3.0 \times 10^{-4}$;

## I.6 RESISC

**CODA-Prompt**: $\text{lr} = 1.0 \times 10^{-3}$;

**DER++**: $\alpha = 3.0 \times 10^{-1}$; $\beta = 8.0 \times 10^{-1}$; $\text{lr} = 1.0 \times 10^{-3}$;

**EWC**: $\epsilon = 1.0 \times 10^2$; $\gamma = 9.0 \times 10^{-1}$; $\text{lr} = 1.0 \times 10^{-2}$;

**L2P**: $\text{lr} = 2.5 \times 10^{-3}$;

**LWF-MC**: $\text{wd} = 0.0$; $\text{lr} = 1.0 \times 10^{-2}$;

**SEED**: $\text{lr} = 3.0 \times 10^{-4}$;

**SGD**: $\text{lr} = 1.0 \times 10^{-2}$;

**TMC**: $\text{lr} = 3.0 \times 10^{-4}$;

**InfLoRA**: $\text{lr} = 5.0 \times 10^{-4}$ $r = 10$; $\epsilon = 0.98$;

For both ITA and IEL: $\# \text{epochs}_{\text{pre-tuning}} = 8$; $\text{lr}_{\text{pre-tuning}} = 1.0 \times 10^{-2}$.

**ITA-FFT**: $\alpha = 1.0 \times 10^4$; $\alpha_{\text{CLS}} = 1.0 \times 10^{-1}$; $\alpha_{\text{CLS-prior}} = 1.0 \times 10^{-2}$; $\text{lr} = 1.0 \times 10^{-4}$;

**ITA-LoRA**: $\alpha = 2.0 \times 10^{-2}$; $\alpha_{\text{CLS}} = 1.0 \times 10^{-1}$; $\alpha_{\text{CLS-prior}} = 1.0 \times 10^{-2}$; $\text{lr} = 3.0 \times 10^{-4}$;

**ITA-(IA)³**: $\alpha = 5.0$; $\alpha_{\text{CLS}} = 1.0 \times 10^{-1}$; $\alpha_{\text{CLS-prior}} = 3.0 \times 10^{-3}$; $\text{lr} = 3.0 \times 10^{-3}$;

**IEL-FFT**: $\beta = 5.0 \times 10^3$; $\beta_{\text{CLS}} = 1.0 \times 10^{-1}$; $\beta_{\text{CLS-prior}} = 3.0 \times 10^{-3}$; $\text{lr} = 1.0 \times 10^{-4}$;

**IEL-LoRA**: $\beta = 2.0 \times 10^2$; $\beta_{\text{CLS}} = 1.0 \times 10^{-2}$; $\beta_{\text{CLS-prior}} = 1.0 \times 10^{-3}$; $\text{lr} = 3.0 \times 10^{-4}$;

**IEL-(IA)³**: $\beta = 2.0$; $\beta_{\text{CLS}} = 1.0 \times 10^{-2}$; $\beta_{\text{CLS-prior}} = 1.0 \times 10^{-3}$; $\text{lr} = 3.0 \times 10^{-4}$;

## I.7 CROPDISEASE

**CODA-Prompt**: $\text{lr} = 1.0 \times 10^{-3}$;

**DER++**: $\alpha = 3.0 \times 10^{-1}$; $\beta = 8.0 \times 10^{-1}$; $\text{lr} = 1.0 \times 10^{-3}$;

**EWC**: $\epsilon = 1.0$; $\gamma = 1.0$; $\text{lr} = 1.0 \times 10^{-2}$;

**L2P**: $\text{lr} = 2.5 \times 10^{-3}$;

**LWF-MC**: $\text{wd} = 1.0 \times 10^{-4}$; $\text{lr} = 1.0 \times 10^{-2}$;

**SEED**: $\text{lr} = 3.0 \times 10^{-4}$;

**SGD**: $\text{lr} = 1.0 \times 10^{-2}$;

**TMC**: $\text{lr} = 1.0 \times 10^{-4}$;

**InfLoRA**: $\mathrm{lr} = 5.0 \times 10^{-4}$ $r = 10$; $\epsilon = 0.98$;

For both ITA and IEL: $\# \mathrm{epochs}_{\mathrm{pre\text{-}tuning}} = 8$; $\mathrm{lr}_{\mathrm{pre\text{-}tuning}} = 1.0 \times 10^{-2}$;

**ITA-FFT**: $\alpha = 5.0 \times 10^3$; $\alpha_{\mathrm{CLS}} = 2.5 \times 10^{-2}$; $\alpha_{\mathrm{CLS\text{-}prior}} = 1.0 \times 10^{-2}$; $\mathrm{lr} = 1.0 \times 10^{-4}$;

**ITA-LoRA**: $\alpha = 5.0$; $\alpha_{\mathrm{CLS}} = 1.0 \times 10^{-1}$; $\alpha_{\mathrm{CLS\text{-}prior}} = 1.0 \times 10^{-2}$; $\mathrm{lr} = 3.0 \times 10^{-4}$;

**ITA-`(IA)`$^3$**: $\alpha = 2.0 \times 10^{-2}$; $\alpha_{\mathrm{CLS}} = 1.0 \times 10^{-1}$; $\alpha_{\mathrm{CLS\text{-}prior}} = 1.0 \times 10^{-2}$; $\mathrm{lr} = 3.0 \times 10^{-4}$;

**IEL-FFT**: $\beta = 2.0 \times 10^2$; $\beta_{\mathrm{CLS}} = 1.0 \times 10^{-1}$; $\beta_{\mathrm{CLS\text{-}prior}} = 1.0 \times 10^{-2}$; $\mathrm{lr} = 1.0 \times 10^{-5}$;

**IEL-LoRA**: $\beta = 5.0 \times 10^1$; $\beta_{\mathrm{CLS}} = 1.0 \times 10^{-1}$; $\beta_{\mathrm{CLS\text{-}prior}} = 1.0 \times 10^{-2}$; $\mathrm{lr} = 3.0 \times 10^{-4}$;

**IEL-`(IA)`$^3$**: $\beta = 2.0$; $\beta_{\mathrm{CLS}} = 2.5 \times 10^{-2}$; $\beta_{\mathrm{CLS\text{-}prior}} = 1.0 \times 10^{-2}$; $\mathrm{lr} = 3.0 \times 10^{-4}$;

