# OpenReview forum: "A Second-Order Perspective on Model Compositionality and Incremental Learning"
_ICLR.cc/2025/Conference — ICLR 2025 Spotlight_

### Official Review · Reviewer_hFQw · 2024-10-31

**Soundness:** 3
**Presentation:** 3
**Contribution:** 3
**Rating:** 6
**Confidence:** 4

**Summary:**

The paper studies model compositionality applied under class-incremental continual learning settings, where individual models fine-tuned on disjoint tasks are composed via weight averaging. By studying the second-order Taylor expansion of deep networks about a strong initial condition, the paper motivates two modifications to existing training objectives. The first encourages local linearity in model weights by minimizing the change in weights (from the pre-trained point) resulting from fine-tuning, which is shown to penalize deviations between the pre-trained and fine-tuned model outputs. The second operates in the regime where individual model weight perturbations are fine-tuned using the loss computed from the _composed_ model. The paper proposes adding a regularizer to minimize the deviation between the loss resulting from weight composition, and the loss resulting from output composition (ensembling).

**Strengths:**

- The paper explores the model composition approach for continual learning, which has strong advantages in more general settings, for instance in federated setups where individual models can be trained in parallel on disjoint datasets and then composed at the end.
- I appreciate the idea of enforcing local convexity or linearity via regularization, since this circumvents the need to do so explicitly via model linearization techniques like in TMC, and it appears to work very well in practice (Table 1).
- The composed model training objective proposed in IEL is also intuitive, essentially "learning to compose" since each new task perturbation is conditioned on all previously seen ones.
- Experiments are comprehensive and a variety of datasets are explored at least in the class-incremental setting. Method performs well across all datasets.

**Weaknesses:**

- The regularization condition proposed in Sec 2.2 does not seem to necessitate the second-order analysis, since it simply aims to enforce a local (first-order) linear structure by making perturbation smaller.
- The wording on "incrementally train(ing) the whole composed model rather than its individual components" used throughout the paper is misleading. I only understood what this statement is trying to convey after seeing equation (15) on Page 5. It appears to me that what is being trained is still individual perturbation components ($\tau_1, \ldots, \tau_T$) , but only that the _loss_ is computed based on the composed model. I would suggest modifying the terminology for better clarity.
- L120 assumes that the pre-trained initialization $\theta_0$ is globally optimal for a quadratic model trained on the union of all tasks. This seems to be a very strong assumption that likely would never hold in practice (why train/compose when the initialization is already optimal?), contrary to what is claimed in footnote 1. This weakens the theoretical analysis.
- Lacks wall-clock time comparison ($\mathcal{O}(1)$ does not shed any light on the actual cost incurred), for instance that used to update the FIM, which is important to faithfully evaluate the method w.r.t. to other works.
- The need for the task pre-consolidation (linear probing, etc.) stage requires additional training steps, so comparisons to most existing methods that do not require this would not be completely fair. If the paper did not implement control steps to make fairer comparisons, then this should be mentioned explicitly in the limitations.

**Questions:**

- I see some replay-based methods like DER++ are included in Table 1, but I don't see any mention of the experimental setup used for these comparisons (e.g. assumptions on buffer size)
- Equation (13) suggests that by minimizing $\Omega$, IEL also seeks to encourage linearity in the loss, as done in ITA.  Can you provide some intuition (formal or informal) regarding under what regimes IEL fairs better than ITA, and vice-versa?

---

> ### Author Response · Authors · 2024-11-19
>
> We thank the reviewer for their careful review and helpful suggestions. We have provided thorough responses to each of the points raised.
>
> ## Sec 2.2 and second-order analysis
>
> > The regularization condition proposed in Sec 2.2 does not seem to necessitate the second-order analysis, since it simply aims to enforce a local (first-order) linear structure by making perturbation smaller.
>
> We would like to clarify that the regularization term in Sec. 2.2 actually stems from a second-order analysis.
> Indeed, the regularization objective based on the FIM originates from a second-order approximation of the KL divergence between the output distribution of the current model and that of the pre-trained model.
> The full derivation was omitted as it builds on well-established proofs \[a, b\] and does not constitute part of our original contributions. We referenced this connection in line 172 of the original submission, and more theoretical details are provided in the first section of the Appendix in \[a\].
>
> - \[a\] Chaudhry, Arslan, et al. "Riemannian walk for incremental learning: Understanding forgetting and intransigence." ECCV. 2018.
> - \[b\] Schwarz, Jonathan, et al. "Progress \& compress: A scalable framework for continual learning." ICML. 2018.
>
> ## Wording
>
> > The wording on "incrementally train(ing) the whole composed model rather than its individual components" used throughout the paper is misleading ... I would suggest modifying the terminology for better clarity.
>
> Based on the comment of the reviewer, we thoroughly analyzed the original submission and identified the following critical lines:
>
> - _line 70_: ...optimizing each model individually vs. optimizing the whole composed model directly
> - _line 106-107_: ... can we incrementally train the whole composed model rather than its individual components (ensemble training)
> - _line 213_: For these reasons, we conduct a theoretical analysis of the dual perspective: the incremental training of the composed model (ensemble training).
>
> To clarify the terminology, we have implemented the following modifications:
>
> - _line 71_ ... optimizing the loss of each model individually vs. optimizing the loss of the composed model
> - _line 106_ ... Instead of optimizing each model on its individual loss, could we optimize each model based on the loss of the whole composed model (ensemble training)?
> - _line 211-212_: ... For these reasons, we adopt the dual perspective, in which each model is directly optimized using the loss of the composed model (ensemble training).
>
> ## Optimality assumption
>
> > L120 assumes that the pre-trained initialization $\theta\_0$ is globally optimal for a quadratic model trained on the union of all tasks. This seems to be a very strong assumption that likely would never hold in practice (why train/compose when the initialization is already optimal?), contrary to what is claimed in footnote 1. This weakens the theoretical analysis.
>
> We acknowledge the optimality of the base model represents a strong assumption that may not hold in practice. In the future, we are committed to extending this work by exploring more relaxed conditions that may yield equivalent theoretical results.
>
> That being said, we believe that incremental training remains valuable even in a scenario where the base model is already considered "locally optimal" with respect to all tasks. Indeed, our framework offers a practical mechanism to decompose the model into multiple weight perturbations, each tailored to specialize in a specific task distribution and potentially achieve a better local minimum. By doing so, the task vectors derived from fine-tuning the base model would enable both specialization and unlearning capabilities, as demonstrated in our experiments.
>
> Moreover, while the assumption of "optimality" for the base model may be considered unrealistic, it serves as a theoretical ground to understand why model merging techniques have been shown to perform more effectively when starting from stronger base models --- e.g., those that are wider or augmented through similar forms of pre-tuning (see \[a\], a work referenced by Reviewer tsHk).
>
> We have reformulated footnote 1 to include some important references provided by Reviewer tsHk.
>
> \[a\] Yadav, P., Vu, T., Lai, J., Chronopoulou, A., Faruqui, M., Bansal, M., \& Munkhdalai, T. (2024). What Matters for Model Merging at Scale?. arXiv preprint arXiv:2410.03617.

---

> ### Author Response · Authors · 2024-11-19
>
> ## Wall-clock time
>
> > Lacks wall-clock time comparison ($\mathcal{O}(1)$ does not shed any light on the actual cost incurred), for instance that used to update the FIM, which is important to faithfully evaluate the method w.r.t. to other works.
>
> In response to the reviewer's suggestion, we have included a wall-clock time analysis in Sec. H.6 of the Appendix. Specifically, considering Split ImageNet-R, Split RESISC45, and Split CropDisease, we measured the per-task runtime of ITA and IEL (both trained with full fine-tuning), as well as that of three existing approaches to incremental learning: DER++ (_rehearsal_), TMC (_compositionality_), and SEED (_ensemble learning_). Notably, the runtime includes the time required to setup the regularization of ITA and IEL (e.g., the computation of the FIM statistics).
>
> Among the methods compared, SEED appears to be the most efficient. However, the runtimes of both ITA and IEL are comparable to, or even better than, those of baseline methods like DER++ and TMC. Specifically, on ImageNet-R and RESISC45, the average runtime of ITA/IEL is approximately $\frac{2}{3}$ that of DER++.
>
> We attribute the reasonable training times of IEL/ITA to the efficient procedure we employed for estimating the FIM, which leverages batch-wise parallelization (refer to Sec. E of the Appendix, see also George, 2021), combined with the use of closed-form gradient computation (refer to Sec. C of the Appendix).
>
> ## Comparison
>
> > The need for the task pre-consolidation (linear probing, etc.) stage requires additional training steps, so comparisons to most existing methods that do not require this would not be completely fair. If the paper did not implement control steps to make fairer comparisons, then this should be mentioned explicitly in the limitations.
>
> We understand the point raised by the reviewer. The comparisons in Tab. 1 were primarily designed to position our approaches within the leaderboard of continual learning. Hence, we selected a heterogeneous and challenging snapshot of state-of-the-art approaches, ranging from older rehearsal-based methods to more recent prompt-tuning techniques. Compared to these methods, we acknowledge that our approach introduces additional overhead due to the pre-tuning stage. On the other hand: _i)_ we avoid the double forward pass required by prompt-tuning approaches such as L2P and CODA-Prompt; _ii)_ we employ a rehearsal-free approach that does not require maintaining a memory buffer (as in DER++); _iii)_ we avoid the additional overhead introduced by TMC in computing the JVP.
>
> Hence, the fairest apple-to-apple comparison among all methods is somewhat elusive due to their heterogeneity. If the reviewer deems it appropriate, we can include this discussion in the supplementary materials.
>
> ## Questions
>
> > I see some replay-based methods like DER++ are included in Table 1, but I don't see any mention of the experimental setup used for these comparisons (e.g. assumptions on **buffer size**).
>
> We sincerely apologize to the reviewer for this oversight. In the updated version of the main paper, we have explicitly stated the buffer size (see the caption of Tab. 1). Specifically, DER++ utilizes a buffer of $1,000$ elements across all the experiments we conducted.
>
> > Equation (13) suggests that by minimizing $\Omega$, IEL also seeks to encourage linearity in the loss, as done in ITA. Can you provide some intuition (formal or informal) regarding under what regimes IEL fairs better than ITA, and vice-versa?
>
> This point is very appealing and aligns with our interests for future work. While ITA trains each model separately, IEL directly trains the ensemble as a whole, rather than each task in isolation. This implies that, with IEL, each task can benefit from the knowledge already accumulated within the ensemble, potentially enabling improved positive transfer. However, to substantiate this claim, we would need additional experimental analyses beyond those reported in this paper, such as comparing the two methods in scenarios where knowledge overlap between consecutive tasks is possible. Hence, we leave this investigation for future work.

---

> > ### Comment · Reviewer_hFQw · 2024-11-21
> >
> > Dear authors,
> >
> > Thank you for your detailed response to my questions. Incorporating the additional implementation/timings details along with discussions on the elusiveness/caveats of an apples-to-apples comparisons would significantly improve the clarity of the paper.
> >
> > My concerns regarding the weakness of the theoretical contributions, and its relative disparity with respect to the empirical evaluations, remain. I still find the assumption on the optimality of $\theta_0$ too strong, and the justification that "incremental training remains valuable even in a scenario where the base model is already considered "locally optimal" with respect to all tasks" is rather shaky. If a model is optimal for all future tasks, why train it further and incur catastrophic forgetting?
> >
> > Decomposing component models for unlearning is indeed a valuable application as explored by [1] (seems to be a missing reference), but if the base model itself already performs optimally with respect to the tasks to be unlearned, then unlearning should be performed on $\theta_0$ instead. The exception is if $\theta_0$ is optimal without having seen any of the data to unlearn, in which case no unlearning or decomposition is necessary, and we can simply keep $\theta_0$ as is.
> >
> > It would be a stronger argument if the authors can show that such situations can hold in practice. Existing experiments use $\theta_0$ as the generic ImageNet pre-training weights, which is clearly non-optimal for the downstream tasks. To justify and validate the theory, it would make sense to use as $\theta_0$ a solution optimal for the downstream tasks. As it stands, the empirical and theoretical portions of the paper seem largely disjoint. Nevertheless, I find the empirical results comprehensive and strong, hence maintain my positive score.
> >
> > [1] Tangent Transformers for Composition, Privacy and Removal. ICLR 2024.

---

> > > ### Author Response · Authors · 2024-11-25
> > >
> > > ## Response
> > >
> > > We sincerely appreciate the time and thought you have been dedicating to this discussion. Given your helpful engagement, we kindly ask for an additional moment of your time.
> > >
> > > > Incorporating the additional implementation/timings details along with discussions on the elusiveness/caveats of an apples-to-apples comparisons would significantly improve the clarity of the paper.
> > >
> > > The requested implementation and timing details have already been incorporated into the updated manuscript. Additionally, we will ensure that a discussion highlighting the caveats and challenges of apples-to-apples comparisons is included in the next version.
> > >
> > > > It would be a stronger argument if the authors can show that such situations can hold in practice. Existing experiments use $\theta\_0$ as the generic ImageNet pre-training weights, which is clearly non-optimal for the downstream tasks. To justify and validate the theory, it would make sense to use as $\theta\_0$ a solution optimal for the downstream tasks.
> > >
> > > Challenged by the valid point raised by the reviewer, we conducted additional experiments to make the assumption of optimality for $\theta\_0$ more reflective of practical conditions. Specifically, we designed a new experimental setting that incorporates a **bootstrapping phase** prior to sequential training on multiple disjoint tasks. During the bootstrapping phase, we fine-tune the ImageNet pre-trained model on a smaller subset of the overall joint training set, consisting of 30\% of the data from all tasks. The resulting model is then used as $\theta\_0$ for subsequent continual fine-tuning. This setting offers the following advantages:
> > >
> > > - The resulting initialization, $\theta\_0$, is closer to the assumption of pre-training optimality, as encouraged by the reviewer.
> > > - The bootstrapping phase simulates real-world applications such as product categorization or face recognition \[c\], where incremental learning typically begins with a model trained on a **pre-collected dataset**.
> > > - In literature, the use of pre-collected data for model bootstrapping is widely regarded as both realistic and feasible, and it is not in contrast with the principles of continual learning, as evidenced by well-established papers \[c, d\], including works that use half of the classes as a bootstrapping phase.
> > > - Despite the creation of the bootstrapped model, continual learning remains essential for integrating the remaining data shards, comprising 70\% of all tasks, into the base model, in order to further improve generalization capabilities.
> > >
> > > Considering the experimental setting herein introduced, we tested our approaches, ITA and IEL, on full fine-tuning, both with and without regularization. The results are presented in the following tables.
> > >
> > > | | **Split ImageNet-R** | **Split CIFAR-100** | **Split MIT67** | **Split RESISC45** |
> > > |--------------------------|----------------------|---------------------|-----------------|--------------------|
> > > | $\theta\_0$ (_bootstrapping_) | $71.50$ | $85.39$ | $75.69$ | $94.49$ |
> > > | **ITA** | $77.37$ | $88.56$ | $79.88$ | $95.15$ |
> > > | **ITA without reg.** | $14.53$ | $32.15$ | $22.80$ | $50.52$ |
> > > |---|---|---|---|---|
> > > | **IEL** | $\mathbf{79.36}$ | $\mathbf{89.28}$ | $\mathbf{79.96}$ | $\mathbf{95.17}$ |
> > > | **IEL without reg.** | $47.38$ | $58.12$ | $47.93$ | $58.57$ |
> > >
> > > From this preliminary analysis\*, we conclude that the performance advantage offered by our proposed regularization remains remarkable even when the assumption of pre-training optimality is brought closer to reality. Both IEL and ITA demonstrate improvements over the near-optimal base model $\theta\_0$ and their unregularized counterparts.
> > >
> > > \* Because of the time constraints of the rebuttal period, these experiments were conducted on four out of seven datasets. However, if the reviewer considers it valuable, we will ensure that the final version includes a comprehensive evaluation across all datasets.
> > >
> > > \[c\] Hou, S., Pan, X., Loy, C. C., Wang, Z., \& Lin, D. (2019). Learning a unified classifier incrementally via rebalancing. CVPR.
> > >
> > > \[d\] Douillard, A., Cord, M., Ollion, C., Robert, T., \& Valle, E. (2020). Podnet: Pooled outputs distillation for small-tasks incremental learning. ECCV.

---

> > > > ### Author Response · Authors · 2024-11-25
> > > >
> > > > ### Bootstrapping and specialization/unlearning capabilities
> > > >
> > > > We also examine the skills in terms of specialization and unlearning in the context of model bootstrapping. In doing so, we follow the same experimental protocol described in the main paper, distinguishing between target and control tasks. As suggested by the reviewer, we compare the edited models with the (bootstrapped) base model $\theta\_0$.
> > > >
> > > > ## **zero-shot specialization**
> > > >
> > > > |Dataset|Method|$FA\_{\text{TGT}}$|$FA\_{\text{CTRL}}$|
> > > > |--|--|--|--|
> > > > |Split ImageNet-R|$\theta\_0$ (_bootstrapping_)|$74.28$|$70.30$|
> > > > ||ITA|$81.53$|$48.79$|
> > > > ||IEL|$41.11$|$5.96$|
> > > > |---|---|---|---|---|
> > > > |Split CIFAR-100|$\theta\_0$ (_bootstrapping_)|$85.43$|$85.37$|
> > > > ||ITA|$90.83$|$60.61$|
> > > > ||IEL|$87.43$|$49.94$|
> > > > |---|---|---|---|---|
> > > > |Split MIT67|$\theta\_0$ (_bootstrapping_)|$70.98$|$77.71$|
> > > > ||ITA|$81.34$|$54.45$|
> > > > ||IEL|$78.51$|$47.37$|
> > > > |---|---|---|---|---|
> > > > |Split RESISC45|$\theta\_0$ (_bootstrapping_)|$94.54$|$94.46$|
> > > > ||ITA|$95.79$|$62.34$|
> > > > ||IEL|$95.94$|$62.72$|
> > > >
> > > > We emphasize the accuracy gap of ITA relative to the base model on the target tasks (see column $FA_{\text{TGT}}$). This suggests that coupling further fine-tuning with ITA enables the model to specialize in a zero-shot manner on a desired selection of tasks. For example, on Split ImageNet-R, ITA achieves an accuracy of $81.53$, which is approximately $+7$ points higher than the base model's $74.28$.
> > > >
> > > > We believe this new evidence further validates our framework, as it is obtained in a setting that bridges the gap between theoretical assumptions (pre-training optimality) and practical implementations (bootstrapping).
> > > >
> > > > ## **zero-shot unlearning**
> > > >
> > > > |Dataset|Method|$FA\_{\text{TGT}}$|$FA\_{\text{CTRL}}$|
> > > > |--|--|--|--|
> > > > |Split ImageNet-R|$\theta\_0$ (_bootstrapping_)|$71.47$|
> > > > ||ITA|$25.02$|$56.79$|
> > > > ||IEL|$4.52$|$25.07$|
> > > > |---|---|---|---|---|
> > > > |Split CIFAR-100|$\theta\_0$ (_bootstrapping_)|$88.56$|
> > > > ||ITA|$63.49$|$76.38$|
> > > > ||IEL|$25.38$|$62.60$|
> > > > |---|---|---|---|---|
> > > > |Split MIT67|$\theta\_0$ (_bootstrapping_)|$79.88$|
> > > > ||ITA|$54.29$|$65.72$|
> > > > ||IEL|$29.65$|$53.14$|
> > > > |---|---|---|---|---|
> > > > |Split RESISC45|$\theta\_0$ (_bootstrapping_)|$94.49$|
> > > > ||ITA|$80.50$|$82.59$|
> > > > ||IEL|$87.84$|$83.55$|
> > > >
> > > > When aiming to remove specific tasks from the base model, the results align with those presented in the main paper, indicating that subtracting task vectors trained with ITA strikes a balance between unlearning the target task and preserving performance on the control task.
> > > >
> > > > > I still find the assumption on the optimality of $\theta\_0$ too strong.
> > > >
> > > > While we understand their concerns, we view the optimality of $\theta_0$ as an idealized scenario, which can be effectively approached when powerful, large-scale foundational models are employed as the base initialization for $\theta\_0$. Indeed, we believe that this assumption essentially stretches the increasingly predominant practice of continual fine-tuning from pre-trained foundation models, making the exploration of near-optimality base condition both relevant and valuable.
> > > >
> > > > Crucially, this idealized perspective has directly inspired the design of our regularization methods, which have demonstrated their effectiveness in a comprehensive experimental analysis, as recognized by all reviewers. In future work, we plan to further relax this assumption or focus on applications where the gap between theoretical assumptions and practical initial conditions can be bridged.
> > > >
> > > > If the reviewer agrees, we will revise the limitations section to explicitly highlight this point and ensure greater clarity.
> > > >
> > > > > Nevertheless, I find the empirical results comprehensive and strong, hence maintain my positive score.
> > > >
> > > > Once again, we express our sincere gratitude to the reviewer for their insightful feedback. Their thoughtful suggestions have offered valuable guidance, enabling us to refine and enhance the quality of this work further.

---

> > > > > ### Author Response · Authors · 2024-11-29
> > > > > **Gentle reminder**
> > > > >
> > > > > As the rebuttal deadline approaches, we wanted to follow up on our response to your most recent comment. We hope the new experiments involving base model bootstrapping have addressed your concerns or, at the very least, clarified our perspective. We deeply appreciate your thoughtful feedback and fully understand the demands on your time.
> > > > >
> > > > > Thank you once again for your time and consideration.

---

> > > > > > ### Comment · Reviewer_hFQw · 2024-12-03
> > > > > >
> > > > > > Thank you for the additional experiments and the response.
> > > > > >
> > > > > > While the new experiments are great and indeed further demonstrate the empirical advantages of the method, $\theta_0$ is remains sub-optimal in the experiments above (except maybe for in Split RESISC45), since after training with ITA the model clearly gets much better. As such, I do not think that these experiments have faithfully reflected what is assumed in the theoretical analysis of the paper. Also, there are no baselines to compare these methods to (understandably since they were done in the rebuttal).
> > > > > >
> > > > > > Regarding the unlearning experiments, my concern was more conceptual and seems to still remain unaddressed
> > > > > > >  if the base model itself already performs optimally with respect to the tasks to be unlearned, then unlearning should be performed on $\theta_0$ instead. The exception is if $\theta_0$ is optimal without having seen any of the data to unlearn, in which case no unlearning or decomposition is necessary, and we can simply keep $\theta_0$ as is.
> > > > > >
> > > > > > Due to the significant weaknesses in the theoretical section of the paper which remain unaddressed, I maintain my current score.  I believe that motivating the algorithm in different manner that does not rely on the unreasonable assumption of optimal $\theta_0$ would greatly strengthen the paper, given its already strong and diverse empirical results.

---

> > > > > > > ### Author Response · Authors · 2024-12-03
> > > > > > >
> > > > > > > Thank you for your valuable feedback. Regarding the point about unlearning, we are not sure we fully understand the reviewer’s concern, but we greatly appreciate the comment and will carefully reflect on it. Considering pre-train optimality, we will revise the final section 'Discussion of limitations and future directions' to explicitly acknowledge this current limitation, in order to provide the more balanced and transparent summary of our contributions.

---

### Official Review · Reviewer_YXTt · 2024-11-02

**Soundness:** 2
**Presentation:** 2
**Contribution:** 2
**Rating:** 8
**Confidence:** 3

**Summary:**

In this work, the author(s) examines the compositional properties of fine-tuning deep pre-trained models, focusing on how specialized modules within these models can be combined to create a single, multi-task model. To be specific, the author leverage a second-order Taylor approximation to understand what promotes compositionality in non-linear networks.

**Strengths:**

This approach reveals the importance of staying within the pre-training 'basin' to create composable modules. To achieve this, the authors introduce two incremental training algorithms: 1) training multiple models individually, and 2) optimizing the entire composed model.

The theoretical analysis section is well-developed, and the experiments are clearly presented.

Tested on incremental classification tasks, this approach enables the creation of an effective multi-task model, with capabilities for unlearning and specialization in specific tasks.

**Weaknesses:**

Major:

The motivation for using the second-order linear approximation is unclear. To my understanding, this work employs a similar approach to neural tangent kernel (NTK) analyses. However, in the linear approximation, NTK typically uses the first-order Taylor expansion, here the author employs the second order. Why not use the first-order Taylor expansion or perhaps a third-order expansion? So, it would be great the author add a section and comparing their work with [1-2], especially the Eq.3-4 with Lemma 3.1 in [2].

Also, there are NTK-based class incremental learning works [3-5]. The authors could include a section in the introduction or discussion to clearly point out the differences and benefits of using their approach compared to these works.

Minor issue:

Some notation is hard to follow and not clearly defined, such as $ L_{\text{cur}}$. The definition of in incremental learning $t,T,\tau$ might also be confusing. The dataset $D_t$ contains $n_t$ training samples $x, y) \sim p_t(x, y)$, drawn from a distribution that varies across tasks. The authors use $\tau$ for $ T $ from $1 \ldots t $, so it is likely that $D_T$ would be more appropriate.


[1] Neural tangent kernel: Convergence and generalization in neural networks.

[2] On Exact Computation with an Infinitely Wide Neural Net

[3] Tkil: Tangent kernel optimization for class balanced incremental learning

[4] Parameter-Efficient Fine-Tuning for Continual Learning: A Neural Tangent Kernel Perspective

[5] A Continual Learning Algorithm Based on Orthogonal Gradient Descent Beyond Neural Tangent Kernel Regime

**Questions:**

see weakness.

---

> ### Author Response · Authors · 2024-11-19
>
> We are grateful to the reviewer for their constructive feedback and valuable insights. Please find our detailed responses to each comment below.
>
> ## Connection with the NTK
>
> > The motivation for using the second-order linear approximation is unclear. To my understanding, this work employs a similar approach to neural tangent kernel (NTK) analyses. However, in the linear approximation, NTK typically uses the first-order Taylor expansion, here the author employs the second order. ... So, it would be great the author add a section and comparing their work with \[1-2\], especially the Eq. 3-4 with Lemma 3.1 in \[2\].
>
> Despite the common use of Taylor approximations, we highlight that there are some notable differences between our approach and those leveraging NTK analysis:
>
> - Our approach employs the second-order Taylor approximation of the **loss function**, aligning with the spirit of works such as Mirzadeh et al. (2020) and Chaudhry et al. (2018). Our focus is on the geometry of the loss landscape and understanding how the loss changes (i.e., potential performance degradation) with respect to weight perturbations.
> - To the best of our knowledge, works dealing with NTK analyses typically apply the first-order Taylor approximation to the **output space**, i.e., the activations of the final layer. Their approach primarily investigates the learning dynamics of the output, whereas our method focuses on approximating the loss function.
>
> We believe that these two frameworks are not mutually exclusive, as they serve different aims. Consequently, they could potentially be integrated to derive alternative solutions. We leave this intriguing, albeit complex, exploration for future work.
>
> > Why not use the first-order Taylor expansion or perhaps a third-order expansion?
>
> With the NTK analyses, the output of the model can be approximated during training by its first-order Taylor expansion around its initialization. However, this approximation has often been shown to be invalid at finite widths, with the training dynamics not being adequately captured by the first-order approximation. Indeed, in their 2024 study, Ortiz-Jimenez et al. observed that models like CLIP do not fine-tune within a linear regime and lack kernel behavior. To address this, both Ortiz-Jimenez et al. (2024) and TMC by Liu \& Soatto (2023) proposed training linearized versions of these models directly, resulting in more disentangled weights.
>
> Instead, we consider the Taylor expansion of the loss function and include a second-order term, with the hope that this could lead to a better approximation. Hence, our approach is not explicitly grounded in NTK or kernel-based behavior. Although it could be extended to even higher-order terms, we limit our focus to the second order due to practical constraints, such as the challenges associated with estimating the third-order derivative of a deep neural network.
>
> ## NTK and Continual Learning
>
> > Also, there are NTK-based class incremental learning works \[3-5\]. The authors could include a section in the introduction or discussion to clearly point out the differences and benefits of using their approach compared to these works.
>
> We thank the reviewer for highlighting these references. We have updated Sec. 4 (Relation to Existing Work, see ll. 369-377) to explicitly address NTK-based approaches for class-incremental learning.
>
> However, we would like to point out that the original submission already includes a comparison with **Tangent Model Composition** (**TMC**) by Liu \& Soatto (2023). This work leverages _model linearization_ to address the shortcomings of standard non-linear fine-tuning, which has been shown to not present kernel behaviour. In this respect, _model linearization_ is essentially equivalent as training a **kernel predictor** with neural tangent kernel defined as $k\_{\rm NTK}(x,x') = \nabla\_{\theta} f(x;\theta\_0)^\top \nabla\_{\theta} f(x';\theta\_0)$. In this respect, we established a theoretical connection between two inequalities (see Eq. 6 and Eq. 7), as well as an experimental comparison with TMC (see Tab. 1 and Tab. 3). Specifically, we demonstrate that our method significantly enhances the task arithmetic capabilities of the merged model, outperforming NTK-based methods such as TMC.

---

> > ### Author Response · Authors · 2024-11-19
> >
> > ## Notation
> >
> > > Some notation is hard to follow and not clearly defined, such as ${L}\_{\text{cur}}$. The definition of in incremental learning $t,T,\tau$ might also be confusing. The dataset $D\_t$ contains $n\_t$ training samples $x, y) \sim p\_t(x, y)$, drawn from a distribution that varies across tasks. The authors use $\tau$ for $T$ from $1 \ldots t $, so it is likely that $D\_T$ would be more appropriate.
> >
> > To address these concerns, we have explicitly introduced the symbol $\mathcal{L}\_{\text{cur}}$ before the formula in line 110 of the revised manuscript. However, we are uncertain about some of the remaining points raised by the reviewer.
> >
> > To clarify, the symbols $t$, $T$, and $\tau$ represent, respectively, a generic task, the total number of tasks, and the task vector (i.e., the shift in parameter space after fine-tuning). The symbol $\tau$ is used consistently with recent works about task arithmetic, like (Ilharco et al., 2022) and (Ortiz-Jimenez et al. (2024)). Furthermore, it is important to maintain a distinction between the training distribution $p\_t(x, y)$ and the dataset $D\_t$, which consists of a finite set of samples used for training.
> >
> > That said, we are open to further engagement with the reviewer and are happy to address specific concerns or fix any issues they identify.

---

> > > ### Comment · Reviewer_YXTt · 2024-11-20
> > >
> > > I appreciate the author's rebuttal; it is clear. I will maintain my positive score for this paper.

---

> > > > ### Author Response · Authors · 2024-11-21
> > > >
> > > > Thank you for your thoughtful feedback and for recognizing the clarity of our rebuttal. If there are any remaining concerns or suggestions for improvement, we would greatly value your insights. Otherwise, if you feel we have sufficiently addressed your concerns, we kindly ask you to consider raising your score to support the acceptance of our manuscript.

---

> > > > > ### Author Response · Authors · 2024-12-01
> > > > > **Further theoretical relations with the NTK theory**
> > > > >
> > > > > The additional days of the rebuttal period provided us with the opportunity to deepen our analysis and establish a **formal connection** between our work and a new potential approach grounded in NTK theory.
> > > > >
> > > > > **TL;DR** We conjecture an alternative approach grounded in linearization and NTK theory. This method leverages regularization to address representation drifts caused by model merging. We demonstrate that the regularized loss function encourages alignment across task vectors, **similar to our IEL**. The key difference lies in the choice of space metrics: IEL employs the Fisher Information Matrix (loss space), whereas the conjectured approach relies on the empirical NTK matrix (output space).
> > > > >
> > > > > For simplicity, we focus on a sequence of just two tasks. We consider first-order linearized models (Ortiz-Jimenez et al. (2024)):
> > > > >
> > > > > $f\_{\rm lin}(x;\theta\_0+\tau) = f(x;\theta\_0) +  (\theta^{\rm FT} - \theta\_0)^\top \nabla\_{\theta} f(x;\theta\_0)=f(x; \theta\_0) + \tau^\top \nabla\_{\theta} f(x;\theta\_0) \quad \text{(Eq. A)}$
> > > > >
> > > > > where $\tau = \theta^{\rm FT} - \theta\_0 \in \mathbb{R}^{|\theta|}$ is the task vector computed from the fine-tuned weights $\theta^{\rm FT}$. $\nabla_{\theta} f(x;\theta\_0) \in \mathbb{R}^{|\theta| \times d}$ indicates the Jacobian matrix, where $d$ and $|\theta|$ stands for the output dimension and the number of parameters respectively. The class-wise prediction can be obtained by feeding the output of the linearized model through a final projection layer.
> > > > >
> > > > > Next, we herein present an alternative training process designed to suit this family of models.
> > > > >
> > > > > **Task 1** Using the data from the first task $\mathcal{D}\_1$, we fine-tune the pre-trained model to minimize the cross-entropy loss on $\mathcal{D}\_1$, resulting in the task vector $\tau\_1$.
> > > > >
> > > > > **Task 2** Using the data from the second task $\mathcal{D}\_2$, we optimize $\tau\_2$ with two objectives:
> > > > >
> > > > > - **Obj 1)** Minimize the cross-entropy loss $\ell(\tau\_2; \mathcal{D}\_2)$ on the second task.
> > > > > - **Obj 2)** Mitigate potential _interference issues_ with the data of the first task.
> > > > >
> > > > > Regarding the second objective, interference arises if merging the two task vectors $\tau\_1$ and $\tau\_2$ causes representation drifts for the data of the first task. This occurs when corresponding output embeddings shift, resulting in a misalignment with the classifier of the first task. Thanks to the linearization property, we can quantify the representation drift that occurs when using the merged task vector $\tau\_\mathcal{P} = \frac{1}{2}(\tau\_1 + \tau\_2)$ as follows:
> > > > >
> > > > > $\delta\_x(\tau\_2)=f\_{\rm lin}(x; \theta\_0+\frac{1}{2}(\tau\_1+\tau\_2)) - f\_{\rm lin}(x; \theta\_0+\tau\_1) \quad \text{s.t.} \quad x \in \mathcal{D}_1 \quad\quad\quad \text{(Eq. B)}$
> > > > >
> > > > > $\quad\quad\quad=\frac{1}{2}(\tau\_2-\tau\_1)^\top \nabla\_{\theta} f(x;\theta\_0) \quad (\text{by using Eq. A}) \quad\quad\quad\quad\quad\quad\quad\quad \text{(Eq. C)}$
> > > > >
> > > > > As stated in Obj 2), we could apply regularization on $\tau\_2$ to minimize the representation drift (i.e., its $L^2$-norm) for examples of the first task.
> > > > >
> > > > > In formulas:
> > > > >
> > > > > $\operatorname{min}\_{\tau\_2} \underbrace{\ell(\tau\_2; \mathcal{D}\_2)}\_{\text{Obj 1)}}+\beta\underbrace{\ell\_{drift}(\tau\_2; \mathcal{D}\_1)}\_{\text{Obj 2) (reg.)}}\quad\text{(Eq. D)}$
> > > > >
> > > > > $\text{where}\quad \ell\_{drift}(\tau\_2; \mathcal{D}\_1)=\frac{1}{n\_1} \sum\_{x \in \mathcal{D}\_1} ||\delta\_x(\tau\_2)||^2 \quad \text{(Eq. E)}$
> > > > >
> > > > > In an incremental scenario, we have to compute $\ell\_{drift}(\tau\_2; \mathcal{D}\_1)$ without relying on examples from the first task. To do so, let us rearrange $\ell\_{drift}$:
> > > > >
> > > > > $\ell\_{drift}(\tau\_2; \mathcal{D}\_1)=\frac{1}{n\_1} \sum\_{x \in \mathcal{D}\_1} ||\delta\_x(\tau\_2)||^2 \quad (\text{let us use Eq. C})$
> > > > >
> > > > > $\quad\quad\quad\quad\quad\quad=\frac{1}{4 n\_1} \sum\_{x \in \mathcal{D}\_1} (\tau\_2-\tau\_1)^\top(\nabla\_{\theta}f(x;\theta\_0)\nabla\_{\theta}f(x;\theta\_0)^\top)(\tau\_2-\tau\_1) \quad \text{(Eq. F)}$
> > > > >
> > > > > The term $\nabla\_{\theta}f(x;\theta\_0) \nabla\_{\theta}f(x;\theta\_0)^\top$ is a squared matrix $|\theta| \times |\theta|$ that can be understood as a **NTK-based sensitivity matrix**, capturing the sensitivity of each parameter within the neural tangent space of the base model. Note that this matrix is not the commonly referenced kernel matrix $\nabla\_{\theta}f(x;\theta\_0)^\top \nabla\_{\theta}f(x;\theta\_0)$ in the NTK literature, which typically has dimensions $d \times d$ (number of output dimension).
> > > > >
> > > > > Thanks to the sum property of quadratic forms, we can bring the expectation over $\mathcal{D}\_1$ inside the matrix product as:
> > > > >
> > > > > $\ell\_{drift}(\tau\_2;\mathcal{D}\_1)=\frac{1}{4} (\tau\_2-\tau\_1)^\top \mathcal{K}\_{\rm NTK}(\mathcal{D}\_1)(\tau\_2-\tau\_1)\quad \text{(Eq. G)}$
> > > > >
> > > > > where $\mathcal{K}\_{\rm NTK}(\mathcal{D}\_1)=\frac{1}{n\_1} \sum\_{x \in \mathcal{D}\_1} \nabla\_{\theta}f(x;\theta\_0) \nabla\_{\theta}f(x;\theta\_0)^\top$ is the empirical NTK-based per-parameter sensitivity matrix.

---

> > > > > > ### Author Response · Authors · 2024-12-01
> > > > > > **Further theoretical relations with the NTK theory**
> > > > > >
> > > > > > **Conclusion** In this conjectured approach, we could reduce representation shifts after model merging by _i)_ approximating and retaining the empirical self-NTK matrix at the end of the first task; _ii)_ using it during the subsequent second task to encourage alignment between $\tau\_1$ and $\tau\_2$. Importantly, the structure of this regularizer closely mirrors that of our proposed approach, IEL (see Eq. 14 and Appendix B.2). The key distinction lies in the choice of the space metric used to parameterize vector alignment: our approach utilizes the Fisher Information Matrix, emphasizing the sensitivity of the loss function, while an NTK-based formulation operates within the neural tangent space, highlighting sensitivity in the output space.
> > > > > >
> > > > > > We appreciate the reviewer's insightful suggestion, which has inspired us to explore this direction further. As a result, we will include this discussion in the supplementary materials and plan to focus our next work on this approach.

---

### Official Review · Reviewer_tsHk · 2024-11-04

**Soundness:** 3
**Presentation:** 3
**Contribution:** 3
**Rating:** 8
**Confidence:** 4

**Summary:**

The paper explores how incremental learning can be addressed through incremental model merging. The theoretical framework is based on the following intuition(s):
- if the base model is already well suited for the downstream asks (\theta_0 is a local min.), fine-tuned versions of this base model will merge better. This connects well to the recent work [5], which shows that stronger base models (instruction tuned, or more params.) lead to better merging.
This intuition serves as a starting point for their theoretical analysis and the derivation of the **ITA** method, which is similar in spirit to ColD Fusion (which does not include the EWC term) [1]. ITA ensures that each new fine-tuned model lies in the viscidity of the base model (as measured by Riemannian distance), which intuitively results in better merging.

It has to be noted, that because this work addressed image classification setting where each task has a different output space, in order to make sure that the base model is a local minimum of the downstream tasks, authors propose the so-called "task pre-consolidation" step, which involves training a task-specific on a fixed backbone before the backbone's params. are learned. The importance of this type of alignment has been recently explored in [2].

**ITA method**: in essence, this method is similar to EWC, with the difference that for each new task we start from the base model (\theta_0) and not from the model learned for the previous task (I have some questions regarding the motivation of this approach, see questions part)

While ITA enables completely independent model training in the spirit of "MoErging" methods [3], one problem with ITA is the fact that each new task's model only transfers knowledge from the base model and not from other tasks. Motivated by this, the paper proposes **IEL** method. IEL optimizes each new task vector assuming access to the merged model on tasks seen so far (somewhat standard in CL literature). This amounts to using both EWC regularization w.r.t to the base model and task alignment term ensuring the new task vector is similar to the previous ones. Importantly, the gradients of the regularization term in IEL can be computed efficiently.

This paper presents a solid empirical section with several interesting insights. All the theoretical insights are substantiated with empirical validations.


[1] Don-Yehiya, Shachar, et al. "Cold fusion: Collaborative descent for distributed multitask finetuning." _arXiv preprint arXiv:2212.01378_ (2022).

[2] Schmidt, Fabian David, Ivan Vulić, and Goran Glavaš. "Free lunch: Robust cross-lingual transfer via model checkpoint averaging." _arXiv preprint arXiv:2305.16834_ (2023).

[3] Yadav, Prateek, et al. "A survey on model moerging: Recycling and routing among specialized experts for collaborative learning." _arXiv preprint arXiv:2408.07057_ (2024).

[4] Ostapenko, Oleksiy, et al. "Continual learning with foundation models: An empirical study of latent replay." _Conference on lifelong learning agents_. PMLR, 2022.

[5] Yadav, Prateek, et al. "What Matters for Model Merging at Scale?." _arXiv preprint arXiv:2410.03617_ (2024).

**Strengths:**

Overall, while the idea of using model merging for continual learning has been explored in previous works, see e.g. [1], this paper does a good job of first formalizing intuitive ideas into an appealing theoretical framework, and subsequently validating these intuitions (and theoretical insights) empirically.  While I have several questions regarding several formulations made in the paper, the overall quality of the work is commendable.

Another strength of this work is its empirical section. Even though experiments are performed only on the image classification domain, a broad range of datasets is covered including also datasets which are OOD w.r.t the backbone.

**Weaknesses:**

- This works connects to several existing works in the realm of model merging, see the "summary" section of my review. These works are not cited in the current version, maybe because these connections were not obvious to the authors. I suggest the authors revisit this potential connection and cite the related works.

**Questions:**

- ll. 028: "Recent AI technologies are predominantly being viewed as monoliths": while I understand the point authors are making here, it must be acknowledged that modular architectures, such as MoE have become increasingly widely used in recent years (maybe as wide as monolithic models)
- ll. 152-157: the paragraph seems to argue that the right-hand side of equation 5 will increase as new tasks are learned. But in equation 5's right-hand side, each individual loss starts from the base model and does not depend on any form of continual learning. So I am not sure I understand the argument being made here.
- would the pre-consolidation step be necessary in language modelling, where the output space is usually shared across tasks?
Empirical section:
- are the different classification heads of the task-specific models also merged together? if so, does it mean that all tasks must have the same number of classes (i.e. share the output space)?
- since IN-R, C-100 and CUB are very much id-distribution w.r.t pre-training on imagenet, I wonder whether simple fine-tuning of the final classification layer, which can be a metric-based classifier with no forgetting, can be sufficient to achieve good performance? (e.g. see [4])
- Did the authors investigate the effect of pre-consolidation on the CL performance?

---

> ### Author Response · Authors · 2024-11-19
>
> Thank you for your thoughtful feedback and recommendations. We address each point in detail in our responses below.
>
> ## Connection to related works
>
> > These works are not cited in the current version, maybe because these connections were not obvious to the authors. I suggest the authors revisit this potential connection and cite the related works.
>
> We sincerely thank the reviewer for their contribution in connecting our work to other fields and highlighting some of the most recent, closely related studies. We have incorporated all five works mentioned in their summary throughout the paper.
>
> Given its importance, we had already included a section discussing related works on model merging in the original submission. However, as our work spans multiple fields and explores their potential intersections, it is challenging to comprehensively address all relevant literature while adhering to the page limit set by ICLR. For this reason, that section was moved to the supplementary materials before submitting the paper, specifically to Sec. F.
>
> ## Questions and clarifications
>
> > ... it must be acknowledged that modular architectures, such as MoE have become increasingly widely used in recent years (maybe as wide as monolithic models).
>
> We agree with this observation and have rephrased the first paragraph of the Introduction to more accurately reflect the feedback (see ll. 028-032).
>
> > ll. 152-157: the paragraph seems to argue that the right-hand side of equation 5 will increase as new tasks are learned. But in equation 5's right-hand side, each individual loss starts from the base model and does not depend on any form of continual learning. So I am not sure I understand the argument being made here.
>
> On the right-hand side of Eq. 5, we observe a sum of the empirical risks achieved by each individual model across **all** tasks. Since each model is trained on only one of these tasks, the upper bound is likely to increase when the evaluation is expanded to include a new task -- particularly if the new task is poorly aligned with the training distributions learned by the individual models in previous tasks.
>
> If any part of this explanation remains unclear, please do not hesitate to contact us for further clarification.
>
> > ll. 152-157: would the pre-consolidation step be necessary in language modelling, where the output space is usually shared across tasks?
>
> In language modeling, where tasks generally share an output space, a pre-consolidation step may not be strictly necessary. However, we believe that strategies to enhance the base model, in the same spirit as pre-tuning, can still be advantageous. For instance, domain adaptation could be employed to strengthen the base model, possibly by leveraging a reference unlabeled dataset with semantic similarities to the downstream tasks. Alternatively, data from the first task could be used for this purpose, following an approach similar to "First Session Adaptation" proposed by \[a\]. We intend to explore these strategies, as well as tasks involving CLIP and Large Language Models (LLMs), in future work.
>
> \[a\] Panos, A., Kobe, Y., Reino, D. O., Aljundi, R., \& Turner, R. E. (2023). First Session Adaptation: A Strong Replay-Free Baseline for Class-Incremental Learning. ICCV.

---

> > ### Author Response · Authors · 2024-11-19
> >
> > ## Questions on the experiments
> >
> > > Are the different classification heads of the task-specific models also merged together? if so, does it mean that all tasks must have the same number of classes (i.e. share the output space)?
> >
> > The classification heads are not merged together; instead, they are concatenated along the class axis to create a unified head across tasks. This approach is suited to our experiments, which involve disjoint output spaces. However, our method can also be effectively applied to scenarios with shared classes but varying input domains (Domain Incremental Learning). As discussed in the previous response, pre-tuning may not be strictly necessary in such cases; however, we believe that methods to strengthen the base model could still provide benefits.
> >
> > > since IN-R, C-100 and CUB are very much id-distribution w.r.t pre-training on imagenet, I wonder whether simple fine-tuning of the final classification layer, which can be a metric-based classifier with no forgetting, can be sufficient to achieve good performance? (e.g. see \[4\])
> >
> > In response, we kindly refer the reviewer to Fig. 1 (Section 5 of the main paper). As shown, in some datasets (e.g., C-100, Caltech, and MIT), the test loss for the approach that fine-tunes only the last layer (yellow line) remains consistently low, suggesting that a metric-based classifier is sometimes sufficient to achieve good performance. However, fine-tuning also the network achieves lower loss values across all datasets, even for those with data distributions aligned with the pretraining distribution.
> >
> > > Did the authors investigate the effect of pre-consolidation on the CL performance?
> >
> > We investigated this during the preliminary phases, and empirically, it appears to help in reducing forgetting. To further address your suggestion, we conducted a comparison with EWC equipped with pre-tuning initialization of the last classification heads. While this approach shows improved performance over naive EWC, it does not reach the accuracy of ITA.
> >
> > \
> >
> > |Method|IN-R|C-100|CUB|Caltech|MIT|RESISC|CropDis.|
> > |--|--|--|--|--|--|--|--|
> > |EWC|$58.64$|$73.49$|$40.33$|$73.23$|$64.44$|$58.80$|$70.33$|
> > |EWC (w. pre-tuning)|$74.71$|$86.04$|$79.99$|$87.59$|$74.57$|$65.75$|$75.70$|
> > |ITA|$76.43$|$89.38$|$84.80$|$92.32$|$85.35$|$80.50$|$91.81$|

---

> > > ### Comment · Reviewer_tsHk · 2024-11-24
> > >
> > > I thank the authors for their detailed response. I will keep my high score.

---

> > > > ### Author Response · Authors · 2024-11-25
> > > >
> > > > Thank you for your thoughtful feedback and appreciation of our work. We truly value your time and effort in reviewing it and remain available to address any further questions or concerns.

---

### Official Review · Reviewer_yT3o · 2024-11-05

**Soundness:** 4
**Presentation:** 3
**Contribution:** 3
**Rating:** 8
**Confidence:** 3

**Summary:**

The paper conducts a theoretical analysis of model fusion in parameter space through the analysis of the second-order Taylor approximation of the multitask loss function.

The authors begin by showing that the 2nd order approximation of the multitask (MT) loss, when evaluated on the $w$-weighted averaged of individual models ($\theta_P = w_1\theta_1 + ... + w_t\theta_t$), is upper bounded by $w$-weighted sum of individual models evaluated on the same multitask data. From this observation, the authors argue that, if each individual model could perform well on all tasks, this would drive the MT loss on $\theta_P$ down.

Given that individual models cannot be optimized on data outside their training distribution, the authors instead propose to regularize individual models towards the base pretrained model $\theta_0$, for such samples. This leads the authors to propose a EWC-like regularizer between $\theta_0$ and $\theta_t$ (when optimizing $\theta_t$) where the Empirical Fisher Information Matrix can be computed iteratively, accumulating gradients from $\theta_0$ on data from task $t$.

Furthermore, the authors proceed to quantify the exact gap between the previous inequality, showing that term is a function of the weighted alignment across task vectors. The authors then proceed to propose a new learning algorithm, which extends the previous EWC regularizer to also penalize not only differences between $\theta_0$ and $\theta_t$, but also difference between $\theta_t$ and subsequent  learned task vectors $\theta_{<t}$.

The authors proceed to analyze their methods on class-incremental learning benchmarks, showing favorable results. An ablation showing the importance of the EWC regularizer is also provided. Finally, akin to Task Arithmetic manipulations, the authors show that task vectors can be used to enhance or unlearn specific task knowledge.

**Strengths:**

1. The paper proposes a novel theoretical analysis of the multitask loss through a second-order Taylor expansion, leading to interesting insights (namely equation 5 & 14)
2. The authors leverage this insight to propose new methods for incremental learning with weight-space merging, showing favorable results.
3. The paper is well-structured, and the experimental section contains relevant ablations showing the effectiveness of the proposed components

**Weaknesses:**

I have some reservations as to the motivation of the proposed methods.

1. Regarding equation 5, I agree that should all individual models perform well on all tasks, then  $\theta_P$ would likely perform quite well. That being said, I find that approximating this condition by regularizing the task vectors to stay close to the base model somewhat of a big conceptual leap. By construction, the base model performs sub-optimally on the tasks for which we optimize a task vector, otherwise the task-specific optimization does not serve much purpose. Therefore, in Eq. 5, wouldn't replacing $\theta_t$ with $\theta_0$ for any $x \sim p_{t'}, t' \neq t$, still make the upper bound loose, and therefore not useful ?

2. The authors motivate ITA vs IEL as an approach that can be executed in a decentralized fashion. However, I am not sure this is the case; given that the FIM is updated sequentially, how would one train several tasks in a decentralized manner without communicating FIM statistics ?

**Questions:**

1. How different are the task vectors learned with ITA vs IEL ? Do you find them to be well-aligned ?

---

> ### Author Response · Authors · 2024-11-19
>
> We appreciate the reviewer's insightful feedback and suggestions. Below, we provide a comprehensive response to each of the comments.
>
> ## Optimality of pre-training model
>
> > ... That being said, I find that approximating this condition by regularizing the task vectors to stay close to the base model somewhat of a big conceptual leap. By construction, the base model performs sub-optimally on the tasks for which we optimize a task vector, otherwise the task-specific optimization does not serve much purpose. Therefore, in Eq. 5, wouldn't replacing ${\theta}\_{t}$ with ${\theta}\_0$ for any $x \sim p_{t'}, t' \neq t$, still make the upper bound loose, and therefore not useful?
>
> We understand the concern expressed by the reviewer and agree that, by construction, the base model likely performs sub-optimally on the downstream tasks. As rightly pointed out, replacing $\theta\_t$ with $\theta\_0$ in Eq. 5 would indeed loosen the upper bound, but not to the same extent as the right-hand side of Eq. 5. As shown in Fig. 1 of the main paper, the upper bound yielded by the base model $\theta\_0$ (yellow line) is consistently better than the original upper bound in Eq. 5 (without regularization, see dark gray rectangles). This occurs because the base model remains frozen and, by construction, maintains broader capabilities (although not optimal) across all tasks. These capabilities are instead lost when training models individually without regularization, hence rendering the upper bound in Eq. 5 excessively loose.
>
> The core intuition, therefore, is that it is better to regularize using a (although not optimal) base model rather than leaving individual models to move arbitrarily in parameter space.
>
> ## ITA and decentralized learning
>
> > ... given that the FIM is updated sequentially, how would one train several tasks in a decentralized manner without communicating FIM statistics?
>
> We would like to discuss further our original statement. In our scenario, the factor that either enables (ITA) or prevents (EWC) decentralized learning is the choice of weight distribution used for regularization. Specifically, while EWC’s FIM pertains to the previous task, our approach computes the FIM statistics with respect to the base model. In our opinion, this approach allows for a decentralized learning setup, which can be summarized as follows:
>
> 1. Multiple disjoint tasks are distributed across several ITA clients, with an ITA server responsible for handling communications.
> 2. Each ITA client performs pre-tuning and computes its local FIM for its training task.
> 3. Each ITA client communicates its FIM to the ITA server.
> 4. The ITA server merges the local FIMs and returns the result to the clients, as is common in Federated Averaging \[a\].
> 5. Each client learns its task with additional regularization provided by the merged FIM.
>
> In this way, multiple task vectors can be trained on different tasks in parallel; the only requirement is to set up a communication phase where clients send their FIMs to the server node. In contrast, an EWC-like term, which regularizes each task based on its predecessor, would require training the multiple task vectors (or clients) in sequence.
>
> \[a\] McMahan, B., Moore, E., Ramage, D., Hampson, S., & y Arcas, B. A. (2017, April). Communication-efficient learning of deep networks from decentralized data. AISTATS.
>
> ## Question: ITA vs IEL and differences in their task vectors
>
> > ... How different are the task vectors learned with ITA vs IEL? Do you find them to be well-aligned?
>
> Thank you for the suggestion. We have included an additional evaluation in Sec. H.5 of the Appendix, where we compute the cosine similarity between the task vectors learned with ITA and IEL across each layer of the ViT-B/16 backbone. For a complete analysis, we evaluate both the average cosine similarity between individual task vectors $\tau\_t^{\text{ITA}}$ and $\tau\_t^{\text{IEL}}$, and the cosine similarity between the composed task vectors $\tau\_{\mathcal{P}}^{\text{ITA}}$ and $\tau\_{\mathcal{P}}^{\text{IEL}}$.
>
> Our analysis indicates that the task vectors are largely similar, particularly for datasets that require significant adaptation. This is mostly evident in out-of-distribution datasets ( _i.e._, RESISC45 and CropDisease) as well as in low-resolution ones  (_i.e._, CIFAR-100).
>
> However, when focusing on the similarity between individual task vectors we observe minimal alignment for CUB-200 and MIT67. We attribute this to the higher complexity of these datasets, with CUB-200 involving fine-grained classification and MIT67 involving capturing nuanced contextual differences, such as those between $\texttt{art studio}$ and $\texttt{classroom}$.
>
> Regardless of this distinction, we observe that the similarity of the composed task vectors consistently exceeds the (average) similarity between individual task vectors.

---

> > ### Comment · Reviewer_yT3o · 2024-11-22
> > **Re: rebuttal**
> >
> > The authors have addressed my initial concerns on the paper. I will update my score accordingly

---

> > > ### Author Response · Authors · 2024-11-25
> > >
> > > Thank you for your thoughtful feedback and for updating your score. We greatly appreciate your time and consideration in reviewing our work. We remain available to address any further questions or concerns.

---

### Author Response · Authors · 2024-11-19

We sincerely thank the reviewers for their valuable feedback. We are glad that all of the reviewers appreciated the well-structured experimental section, which was noted for its comprehensive evaluations, including the tested datasets (tsHk, hFQw), ablation studies (yT3o) and its editing capabilities (YXTt). The two proposed algorithms (yT3o, YXTt), along with the concept of enforcing local convexity or linearity through regularization (hFQw), have been recognized as **significant contributions**. Additionally, reviewers yT3o and tsHk highlighted the **novelty** and valuable insights offered by our theoretical framework.

# Summary of changes

To address the concerns of the reviewers, we revised the paper and uploaded a new draft with the following changes:

- **yT3o**: We computed and analyzed the alignment between the task vectors of ITA and IEL (Sec. H.5 of the Appendix).
- **tsHk**: We reviewed and incorporated the suggested papers as new references in the main paper.
- **YXTt**: We provided a discussion on the relationship between our work and those leveraging the NTK regime (Sec. 4 of the main paper, ll. 369-377).
- **hFQw**: We added a wall-clock time comparison across various datasets and methods (Sec. H.6 of the Appendix) and clarified some previously missing details, such as the size of the buffer used for DER++.

---

### Meta-Review · Area_Chair_oa5f · 2024-12-16

**Metareview:**

This work presents a theoretical analysis on model merging. Through the second-order Taylor decomposition of the multi-task loss, they find that it is important that the weights of the models to be composed lie within the pre-training basin. This motivates introducing a new incremental learning algorithm that keeps the weights of each task close to the original pre-trained weights in a similar way to EWC, achieving promising results.

All reviewers find this is a good work and there is consensus towards acceptance.

**Additional Comments On Reviewer Discussion:**

All reviewers find this is a good work and there is consensus towards acceptance.

---

### Decision · Program_Chairs · 2025-01-22

Accept (Spotlight)